**ARTICLES**

## OPEN

# Genetic control of RNA splicing and its distinct role in complex trait variation

Ting Qi[1,2,3], Yang Wu [ID][3], Hailing Fang[1,2], Futao Zhang[3,4,5], Shouye Liu[1,3], Jian Zeng [ID][3] and Jian Yang [ID][1,2,3] ✉

**Most genetic variants identified from genome-wide association studies (GWAS) in humans are noncoding, indicating their role in gene regulation. Previous studies have shown considerable links of GWAS signals to expression quantitative trait loci (eQTLs) but the links to other genetic regulatory mechanisms, such as splicing QTLs (sQTLs), are underexplored. Here, we introduce an sQTL mapping method, testing for heterogeneity between isoform-eQTL effects (THISTLE), with improved power over competing methods. Applying THISTLE together with a complementary sQTL mapping strategy to brain transcriptomic data ($n = 2,865$) and genotype data, we identified 12,794 genes with *cis*-sQTLs at $P < 5 \times 10^{-8}$, approximately 61% of which were distinct from eQTLs. Integrating the sQTL data into GWAS for 12 brain-related complex traits (including diseases), we identified 244 genes associated with the traits through *cis*-sQTLs, approximately 61% of which could not be discovered using the corresponding eQTL data. Our study demonstrates the distinct role of most sQTLs in the genetic regulation of transcription and complex trait variation.**

GWAS have led to the discovery of tens of thousands of genetic variants associated with human complex traits (including diseases)[1,2]. However, most of the trait-associated variants are of uncharacterized function and the mechanisms through which genetic variants exert their effects on traits are largely elusive. Considering that most of the GWAS signals are located in noncoding regions of the genome, one hypothesis is that genetic variants affect traits through genetic regulation of gene expression[3,4]. The effects of genetic variation on messenger RNA abundance (also known as eQTLs) have been studied for more than a decade[5–7] and nearly all genes have been found with one or more genetic variants associated with their mRNA abundance[6,8]. These advances have propelled the development of methods[9–13] to integrate eQTL data with GWAS data to prioritize the genes responsible for GWAS signals. However, only a moderate proportion of GWAS signals have been attributed to *cis*-eQTLs[14–18], likely because of various reasons, including limited power, spatiotemporal eQTL effects that occur in specific tissues or cell types at specific developmental stages, focus on genomic regions in *cis* and mechanisms beyond the genetic control of mRNA abundance.

Genetic control of pre-mRNA splicing (also called sQTLs) is another fundamental mechanism of gene regulation but is heavily underexplored compared to eQTLs. Currently, there is no consensus in the literature regarding the relationship between eQTLs and sQTLs. For example, there are observations showing that sQTLs are largely independent of eQTLs[6,19,20] and hypothesized to be one of the major contributors to genetic risk of disease[20], whereas a recent study showed that the contribution of sQTLs to trait heritability is not statistically significant conditional on eQTLs[21]. These results motivated us to investigate the role of sQTLs in complex traits using a larger dataset. Depending on how alternative splicing variation is quantified, sQTL mapping strategies can be broadly classified into two categories[22], that is, transcript-level[23–27] or event-level[28–31], each favoring different types of splicing events (Section 1 of the Supplementary Note).

In this study, we aimed to investigate the genetic control of RNA splicing by generating the largest collection of sQTLs to date and describing their role in complex trait variation. We focused our study on the brain because of data availability and the considerable links of sQTLs to neurodegenerative diseases such as Alzheimer's disease (AD)[32], schizophrenia[33] and Parkinson's disease (PD)[34,35] reported recently. Recognizing the differences between transcript- and event-level sQTL mapping strategies (Section 1 of the Supplementary Note), we intended to combine the two strategies with state-of-the-art tools, that is, RSEM[36] and sQTLseekeR[27,37] for transcript-level analysis and LeafCutter[31] and QTLtools[38] for event-level analysis, to increase the yield of sQTLs. Nevertheless, the limited number of sQTLs detected by sQTLseekeR motivated us to develop a more powerful transcript-level sQTL method, THISTLE. We applied THISTLE together with LeafCutter and QTLtools to the largest publicly available brain transcriptomic data ($n = 2,865$) with genotype data to detect sQTLs and integrated the sQTL summary statistics into GWAS for 12 brain-related traits (including diseases) of large sample sizes ($n = 51,710–766,345$) to prioritize genes associated with the traits through genetic regulation of splicing. We benchmarked the role of sQTLs in complex trait variation by the eQTLs identified using the same data.

### Results

**Calibration of THISTLE.** Details of the THISTLE method can be found in the Methods, with the schematics of the method illustrated in Fig. 1. We calibrated THISTLE using simulations in comparison with three existing sQTL methods in the same category, namely sQTLseekeR v.2 (ref. [37]), DRIMSeq v.1.18 (ref. [39]) and multivariate analysis of variance (MANOVA) (implemented in rrcov v.1.5.5). We first performed simulations with mRNA abundances generated from multivariate normal or Poisson distributions (Section 2 of the Supplementary Note) and focused on the comparison with sQTLseekeR. There was no inflation in false positive rate (FPR)

[1]School of Life Sciences, Westlake University, Hangzhou, China. [2]Westlake Laboratory of Life Sciences and Biomedicine, Hangzhou, China. [3]Institute for Molecular Bioscience, The University of Queensland, Brisbane, Queensland, Australia. [4]Neuroscience Research Australia, Sydney, New South Wales, Australia. [5]Clinical Genetics and Genomics, New South Wales Health Pathology Randwick, Sydney, New South Wales, Australia. ✉e-mail: jian.yang@westlake.edu.cn

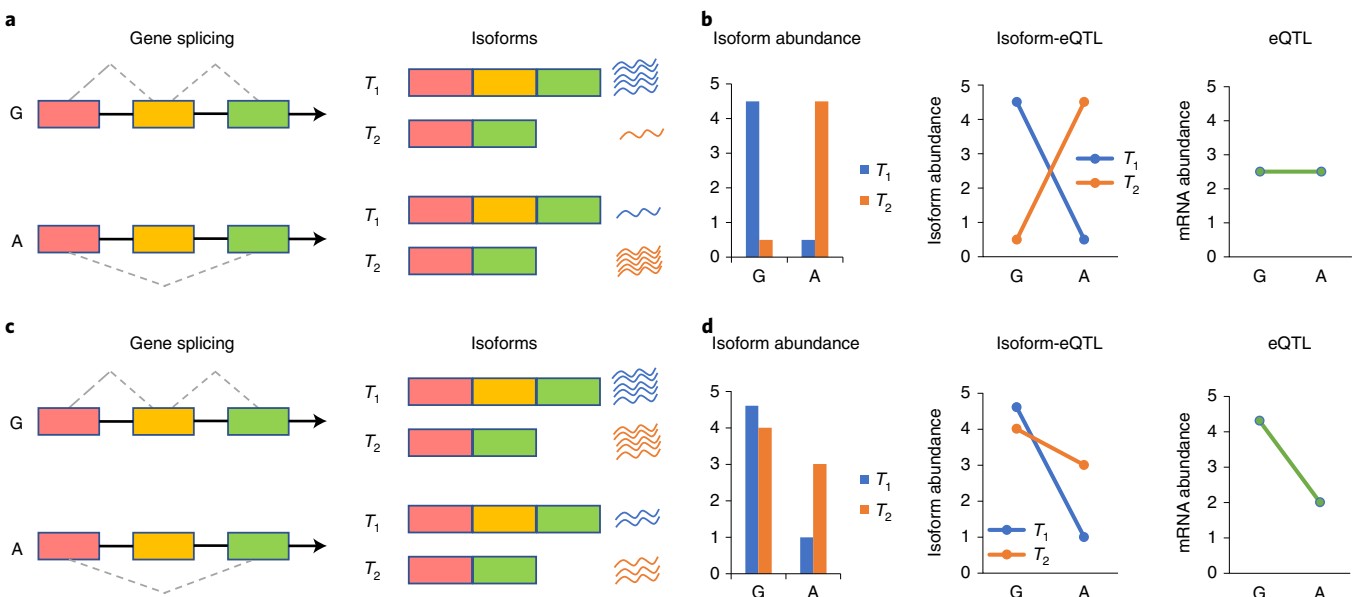

**Fig. 1 | Schematics of the THISTLE sQTL analysis.** In this toy example, a genetic variant with two alleles, G and A, is associated with a splicing event (for example, exon skipping) in a gene with two transcript isoforms, $T_1$ and $T_2$. **a,b**, Schematics of the THISTLE sQTL analysis in the absence of an eQTL effect. In this scenario, individuals with the G allele show higher mean abundance of $T_1$ than $T_2$ and individuals with the A allele show higher mean abundance of $T_2$ than $T_1$ (**a**), meaning that the genetic variant is associated with the difference in abundance between isoforms. In other words, there is a difference in the isoform-eQTL effect between $T_1$ and $T_2$ (**b**). However, there is no difference in overall gene expression between individuals with different alleles, meaning that this genetic variant is not an eQTL. **c,d**, Schematics of the THISTLE sQTL analysis in the presence of an eQTL effect. In this scenario, individuals with the G allele show similar abundance between $T_1$ and $T_2$ and individuals with the A allele show lower abundance of $T_1$ than $T_2$ (**c**). The isoform-eQTL effect for $T_1$ is different from that for $T_2$ albeit in the same direction (**d**). In this case, there is a difference in overall gene expression between alleles G and A, meaning that this genetic variant is also an eQTL.

under the null hypothesis of no sQTL effect for both THISTLE and sQTLseekeR, in either the absence or presence of an eQTL effect (Supplementary Fig. 1). The statistical power of a method is often measured by the true positive rate (TPR). We also used the area under the receiver operating characteristic curve (AUC) to measure power to account for potential inflation in FPR. Although the overall AUC for THISTLE was only slightly (4.7% on average) larger than that for sQTLseekeR, the difference in TPR between the methods increased with the $-\log_{10}(P)$ threshold (Extended Data Fig. 1). Next, we compared all four methods using more comprehensive simulations, with mRNA abundances generated by sampling RNA sequencing (RNA-seq) reads[40] in a broader range of scenarios with varying sample sizes, sQTL effect sizes, degree of overdispersion of transcription abundance and number of isoforms per gene (Section 2 of the Supplementary Note). All methods except DRIMSeq showed well-calibrated test statistics under the null hypothesis (Supplementary Fig. 2); THISTLE was more powerful than sQTLseekeR, DRIMSeq and MANOVA in most scenarios, including those in which genes were simulated with a large number of isoforms (Extended Data Fig. 2). In line with the observation above, the difference in TPR between THISTLE and sQTLseekeR increased with a more stringent $P$ threshold (Extended Data Fig. 2e).

An increase of difference in power between THISTLE and sQTLseekeR with the increased $-\log_{10}(P)$ threshold was also observed (even more prominently) in the real-data analysis. For instance, in the analysis of the Religious Orders Study and Memory and Aging Project (ROSMAP) data ($n = 832$), 6,358 genes with 795,592 unique sQTL SNPs (no linkage disequilibrium (LD) clumping) were discovered by THISTLE versus 3,077 genes with 390,497 sQTL SNPs by sQTLseekeR using a $P$ threshold of $5 \times 10^{-8}$ (Methods, Extended Data Fig. 3b and Section 3 of the Supplementary Note), a 2.1- and 2.0-fold difference in the number of sGenes and sQTL SNPs,

respectively. In this article, we refer to genes with a significant sQTL as sGenes. The ratio decreased to 1.9 at $P < 1 \times 10^{-6}$, to 1.2 at $P < 1 \times 10^{-4}$ and eventually to nearly 1 at $P < 1 \times 10^{-3}$ (Extended Data Fig. 3d). Analysis of the ROSMAP data without covariate adjustment led to a decreased number of sGenes for both methods but the ratio of THISTLE to sQTLseekeR was 1.7 at $P < 5 \times 10^{-8}$. Despite the differences, there was a strong overlap between the sQTLseekeR and THISTLE sQTL results (Extended Data Fig. 3e), the splicing events captured by THISTLE were similar to those by sQTLseekeR (Supplementary Fig. 3) and the THISTLE $P$ values computed from the saddlepoint approximation[41] were remarkably consistent with those from the Davies method[42] used in the latest version of sQTLseekeR (Supplementary Fig. 4). Moreover, benchmarked on a computing platform with 16 GB memory and 16 central processing unit cores, the overall runtime of THISTLE (including the time to estimate the isoform-eQTL effects) in the analysis of the ROSMAP data with 382 genes and 109,853 SNPs on chromosome 22 was 1.05 min (averaged from 10 repeats), approximately 7.6 times faster than sQTLseekeR (Extended Data Fig. 4). In addition, the performance of THISTLE using individual-level SNP genotype and RNA-seq data was similar to that using summary-level isoform-eQTL data (Supplementary Fig. 5).

**Identifying *cis*-sQTLs in the brain.** We applied THISTLE to ten brain transcriptomic datasets from seven cohorts. After quality control (Methods, Extended Data Fig. 5, Supplementary Figs. 6 and 7 and Section 4 of the Supplementary Note), we included in the analysis RNA-seq data of 2,865 samples from 2,443 unrelated individuals of European ancestry and genetic data of approximately 12 million variants with a minor allele frequency (MAF) > 0.01 (Supplementary Table 1). In total, we identified 1,342,073 unique *cis*-sQTL SNPs with $P_{sQTL} < 5 \times 10^{-8}$ for 9,305 genes (Supplementary

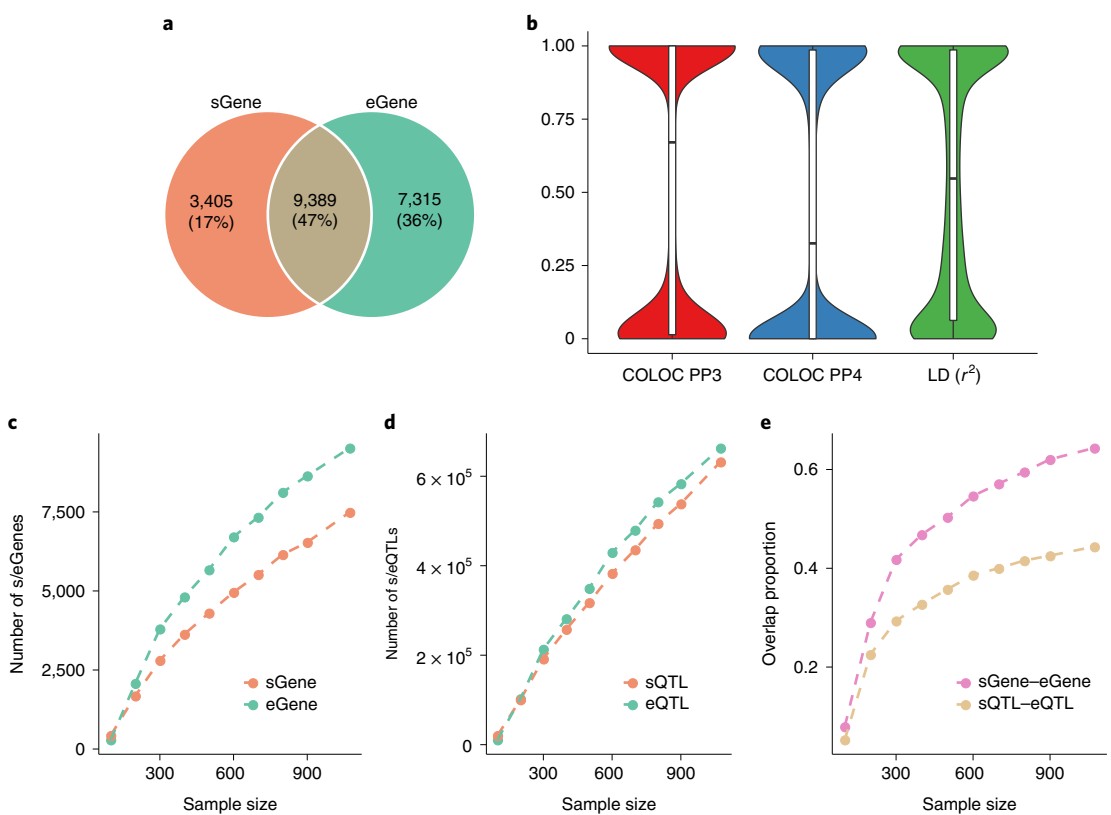

**Fig. 2 | Relationship between sQTLs and eQTLs. a**, Overlap between sGenes and eGenes. **b**, COLOC PP3 and PP4 values between the *cis*-sQTL and *cis*-eQTL signals and LD $r^2$ between the lead *cis*-sQTL and *cis*-eQTL SNPs for the 9,389 overlapping genes. The line inside each box indicates the median value, the notches indicate the 95% confidence interval (CI), the central box indicates the interquartile range (IQR) and the whiskers indicate data up to 1.5 times the IQR. **c**, The number of sGenes (or eGenes) discovered as a function of sample size. **d**, The number of sQTLs (or eQTLs) discovered as a function of sample size. **e**, The overlap between sGenes and eGenes (or between sQTLs and eQTLs) as a function of sample size, where sQTL–eQTL overlap is defined as the proportion of sGenes for which the lead sQTL SNP is a significant eQTL SNP for the same gene.

Table 2). We focused most of the subsequent analyses on the sQTLs with $P_{sQTL} < 5 \times 10^{-8}$ for two reasons: (1) it is the most commonly used genome-wide significance threshold in GWAS and the default threshold used in summary-data-based Mendelian randomization (SMR) to select instrument SNPs[11] (see below); (2) it was more stringent than the permutation-based $P$ threshold in all the 10 datasets and the sQTLs with $P_{sQTL} < 5 \times 10^{-8}$ represented a large proportion of the sQTLs with a false discovery rate (FDR) < 0.05 (Section 5 of the Supplementary Note and Supplementary Table 3). Of the 9,305 sGenes detected at $P_{sQTL} < 5 \times 10^{-8}$, 220 (2.4%) were lowly expressed in the brain with median transcripts per million (TPM) <0.1 (Supplementary Fig. 8b). Moreover, by comparing the number of sGenes identified above (based on GENCODE v.37) with those based on three other transcriptome references (RefSeq, GENCODE v.37 Basic, which includes only a subset of representative transcripts for each gene, and de novo assembly), we showed that GENCODE v.37 substantially outperformed RefSeq and GENCODE v.37 Basic and that de novo assembly gave rise to only approximately 4% more sGenes (Extended Data Fig. 6). Considering the small gain and potential errors in de novo assembled transcripts, we opted to use the GENCODE v.37 results in the following analyses.

Next, we used LeafCutter to detect splicing events that might not be well captured by transcript-level analysis (Methods, Extended Data Fig. 5 and Supplementary Fig. 7). Overall, we identified 1,371,483 unique *cis*-sQTL SNPs for 15,136 intron clusters in 8,602 genes at $P_{sQTL} < 5 \times 10^{-8}$ (Supplementary Table 2) and 203,889 unique *cis*-sQTL SNPs for 1,148 intron clusters with unknown associated genes. Of the 8,602 sGenes detected at $P_{sQTL} < 5 \times 10^{-8}$, 174

genes (2.1%) were lowly expressed in the brain (Supplementary Fig. 8c). As above, the $P$ threshold of $5 \times 10^{-8}$ was more stringent than the permutation-based $P$ threshold at an FDR < 0.05 in all 10 datasets (Supplementary Table 3).

Combining the sQTL results from THISTLE and LeafCutter and QTLtools, there were 1,864,200 unique *cis*-sQTL SNPs for 12,794 sGenes at $P_{sQTL} < 5 \times 10^{-8}$ compared with 462,722 unique sQTL SNPs for 7,296 sGenes from the largest previous study[43,44] (Section 6 of the Supplementary Note, Supplementary Fig. 9 and Supplementary Table 4). There were 4,192 and 3,489 sGenes unique to THISTLE and LeafCutter and QTLtools, respectively, and 5,113 common sGenes for both (Extended Data Fig. 7a). For 2,858 of the 5,113 common sGenes, the THISTLE sQTL signal was distinct from the LeafCutter and QTLtools sQTL signal as indicated by a COLOC[9] PP3 value >0.8 (Supplementary Fig. 10), in line with many common sGenes for which the lead sQTL SNPs from the 2 methods were in low-to-moderate LD (Extended Data Fig. 7b). Together with the large proportions of method-specific sGenes, this result suggests that most sQTL signals detected by THISTLE and LeafCutter and QTLtools were distinct, demonstrating the benefit of using a combination of the two sQTL mapping strategies.

**Quantifying the relationship between sQTLs and eQTLs.** To assess the relationship between sQTLs and eQTLs, we performed an eQTL analysis using the same data as above (Methods and Extended Data Fig. 5) and identified 1,962,048 unique *cis*-eQTL SNPs with $P_{eQTL} < 5 \times 10^{-8}$ for 16,704 genes (Supplementary Table 2). Similarly, we refer to genes with a significant eQTL as eGenes. We found that

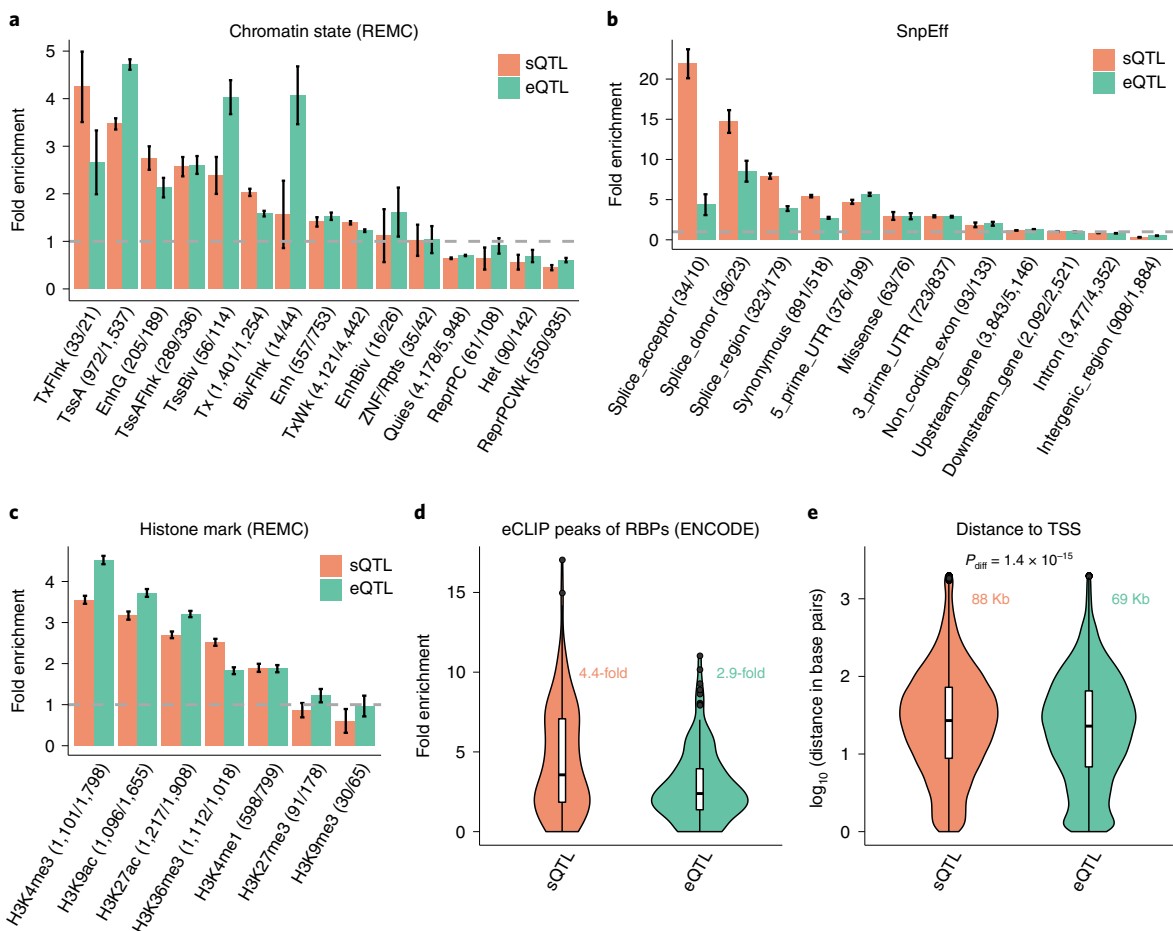

**Fig. 3 | Enrichment of the lead *cis*-sQTL or *cis*-eQTL SNPs in functional annotation categories. a–d**, The annotation categories were defined by the chromatin state annotation data from REMC (**a**), predicted variant functions by SnpEff (**b**), histone marks from REMC (**c**) and eCLIP peaks of 113 RBP binding sites from ENCODE (**d**). **a–c**, The fold enrichment was computed by dividing the percentage of lead *cis*-sQTL (or *cis*-eQTL) SNPs in a category by the mean percentage observed in 1,000 sets of control SNPs sampled repeatedly at random (Methods). Each column represents an estimate of fold enrichment with an error bar indicating the 95% CI of the estimate. The gray dashed line represents no enrichment. The numbers in parentheses are the number of sQTL/eQTL SNPs in each functional category. **d**, The text in color represents the median across 113 RBP binding sites. **e**, Distance of the lead *cis*-sQTL (or *cis*-eQTL) SNP to the TSS of the gene. The texts in color represent the median across 12,794 sGenes and 16,704 eGenes, respectively; $P_{diff}$ was computed from a two-sided *t*-test for a mean difference between the two groups. **d,e**, The violin plots show the distributions of fold enrichment estimates of sQTL and eQTL SNPs across the 113 RBP biding sites (**d**) or distances to the TSS across 12,794 sGenes and 16,704 eGenes (**e**), respectively. The line inside each box indicates the median value, the notches indicate the 95% CI, the central box indicates the IQR, the whiskers indicate data up to 1.5 times the IQR and the outliers are shown as separate dots.

73% (9,389 out of 12,794) of the sGenes were also eGenes (Fig. 2a), a proportion much higher than that (50%) reported in a recent study using transcriptomic data from the fetal brain[19]. We hypothesized that the difference is due to the larger sample size (*n*) used in our study than that in the fetal brain study (*n* = 233). To test this hypothesis, we used a downsampling strategy to assess the sGene–eGene overlap in several subsets of data with *n* ranging from 100 to 1,073. The result showed that the power of either sGene or eGene discovery was proportional to *n* and that the difference in discovery power between sGene and eGene increased with increasing *n* (Fig. 2c,d), in line with the observation from previous work[6]. We also found that the sGene–eGene overlap was positively correlated with *n* (Fig. 2e), which is expected if most genes have both eQTLs and sQTLs. Of the 9,389 overlapping genes, there were 4,377 genes for which the sQTL signal was distinct from the eQTL signal as indicated by a COLOC PP3 value >0.8, in line with a large proportion of overlapping genes for which the lead sQTL SNP was in low-to-moderate LD with the lead eQTL SNP (Fig. 2b). The result was largely unchanged when we performed the colocalization analysis with eCAVIAR[12] that accounts

for multiple causal variants at a locus (Supplementary Fig. 11). In summary, although a large proportion of sGenes are expected to be eGenes with large *n* and this proportion increases with increasing *n*, most sQTLs are distinct from eQTLs (an estimate of approximately 61%, (4,377 + 3,405)/12,794 in this study, with 3,405 being the number of genes that are sGenes only).

**sQTLs are enriched for splicing and RNA-binding protein binding sites.** Having shown that sQTLs were largely distinct from eQTLs (Fig. 2), we then asked whether sQTLs and eQTLs show different patterns of functional enrichment. To do this, we annotated the lead SNP for each of the 12,794 sGenes and 16,704 eGenes using SnpEff[45], functional annotation data of 113 RNA-binding protein (RBP) binding sites[46], 7 histone marks[47] and 15 chromatin states[47] (Methods). The fold enrichment was computed as the proportion of sQTLs or eQTLs in a functional category divided by the mean of a null distribution generated by resampling 'control' SNPs with MAF and distance to the transcription start site (TSS) matched to the SNPs in question (Methods and Section 7 of the Supplementary

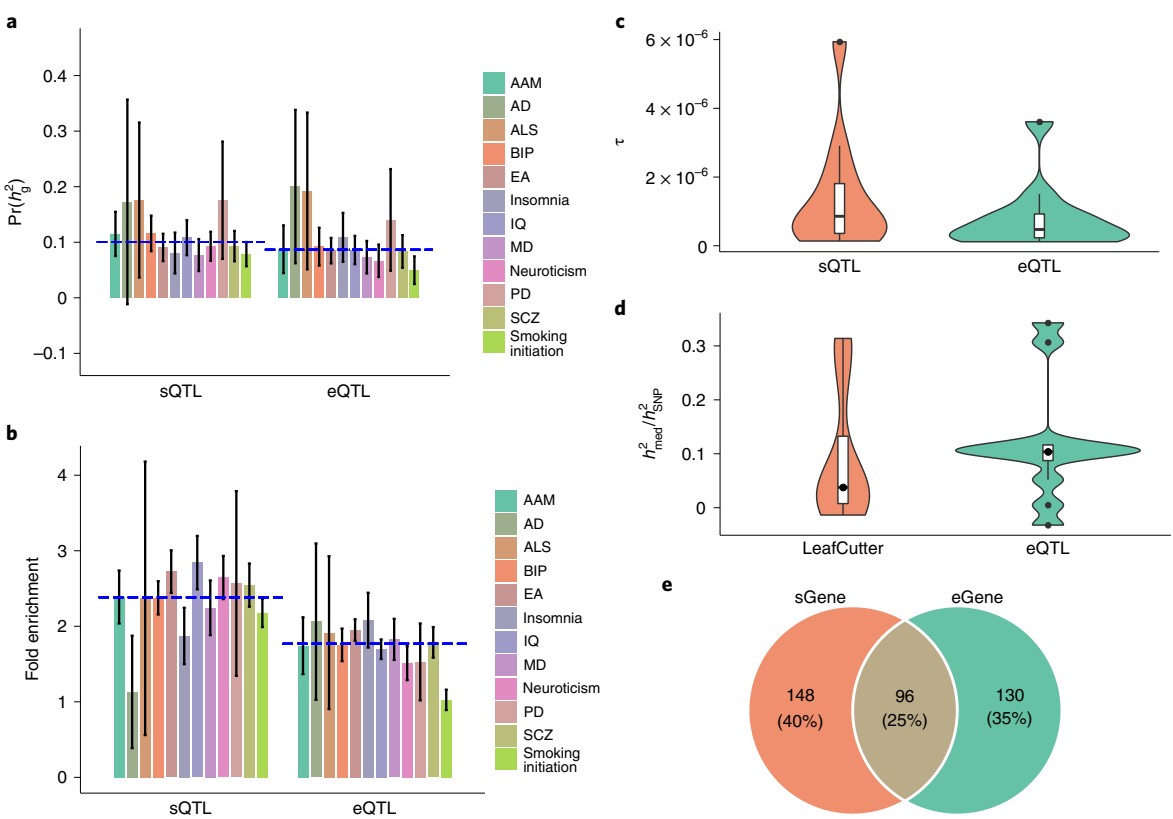

**Fig. 4 | Enrichment of the lead *cis*-sQTL or *cis*-eQTL SNPs for heritability of the 12 brain-related traits. a**, Pr $\left(h_g^2\right)$ is a ratio of heritability attributable to the SNPs in query to the overall SNP-based heritability ($h_{xQTL}^2/h_{SNP}^2$). **b**, Heritability enrichment is defined as a ratio of the proportion of heritability explained by the SNPs in query to the mean proportion observed in 1,000 sets of control SNPs sampled repeatedly at random (Methods). **a,b**, The blue dashed line represents the median value across traits; each column represents a point estimate with an error bar indicating the 95% CI of the estimate. **c**, The stratified LD score regression parameter $\tau$ was used to assess the contribution of the lead *cis*-sQTL SNPs to heritability when fitted jointly with the lead *cis*-eQTL SNPs. **d**, Proportion of heritability mediated by the *cis*-sQTL (or *cis*-eQTL) SNPs ($h_{med}^2/h_{SNP}^2$), estimated by the mediated expression score regression. Only the sQTLs discovered by LeafCutter and QTLtools were included in the mediated expression score regression analysis. **c,d**, The violin plots show the distributions of the estimates of $\tau$ and $h_{med}^2/h_{SNP}^2$, respectively, across the 12 traits. The line inside each box indicates the median value, the notches indicate the 95% CI, the central box indicates the IQR, the whiskers indicate data up to 1.5 times the IQR and the outliers are shown as separate dots. **e**, Overlap between the trait-associated sGenes and eGenes identified by SMR and COLOC PP4.

Note). The results showed that sQTLs were more enriched in splicing sites (for example, splice acceptors and splice donors) and RBP binding sites than eQTLs (Fig. 3b,d). eQTLs were more enriched in the TSS than sQTLs (for example, active TSS and bivalent TSS; Fig. 3a), which is consistent with our observation that eQTLs were located much closer to the TSS than sQTLs (Fig. 3e). Both sQTLs and eQTLs tended to be enriched at both ends of gene bodies, exons and introns (Supplementary Fig. 12) and were significantly enriched in the histone marks H3K4me3, H3K9ac, H3K27ac, H3K36me3 and H3K4me1 (Fig. 3c) but depleted in noncoding regions (Fig. 3b). Compared to that of eQTLs, the enrichment of the sQTLs was higher in H3K36me3 but lower in H3K4me3, H3K9ac or H3K27ac (Fig. 3c). The functional enrichment patterns were largely unchanged whether we stratified the sQTLs by mapping strategy (Extended Data Fig. 8) or performed the enrichment analysis with TORUS[48] using the full summary statistics without SNP selection (Supplementary Fig. 13).

**Enrichment of sQTLs for trait heritability.** We next tested whether the brain *cis*-sQTLs are enriched for genetic variants associated with complex traits and disorders related to the brain. We acquired GWAS summary statistics for 12 brain-related traits from previous work[49–60] (Methods and Supplementary Table 5). Both sQTLs and eQTLs showed more inflated GWAS test statistics compared

to the other SNPs for all 12 traits (Supplementary Fig. 14) and the levels of inflation were indistinguishable between sQTLs and eQTLs (Supplementary Fig. 15). We then performed a stratified LD score regression[61,62] analysis to quantify the enrichment of the lead *cis*-sQTL SNPs for heritability in comparison with that of the lead *cis*-eQTL SNPs when fitted together with 53 other functional categories in the 'baseline-LD model'[61] (Methods). The results showed that the sQTLs were significantly enriched for heritability for most traits and the fold enrichment of sQTLs was comparable to (even appeared to be higher than) that of eQTLs on average across traits (Supplementary Fig. 16 and Fig. 4a). Considering that the lead SNPs were ascertained in the *cis*-regions known to explain disproportionately more trait variation than intergenic regions[63], we adjusted the heritability enrichment by the control SNPs mentioned above (Methods). Under this stringent definition, the overall levels of enrichment decreased but the fold enrichment of sQTLs was comparable to that of eQTLs on average across traits (Fig. 4b). We further performed sensitivity analyses to quantify the heritability enrichment at all the significant, LD-clumped or fine-mapped *cis*-sQTL (or *cis*-eQTL) SNPs[21]; the results consistently showed comparable levels of heritability enrichment between sQTLs and eQTLs (Supplementary Figs. 17b, 18b and 19b). We showed that the $\tau$ estimates for the *cis*-sQTLs and *cis*-eQTLs were significant for most traits (Methods, Fig. 4c and Supplementary Figs. 17c, 18c

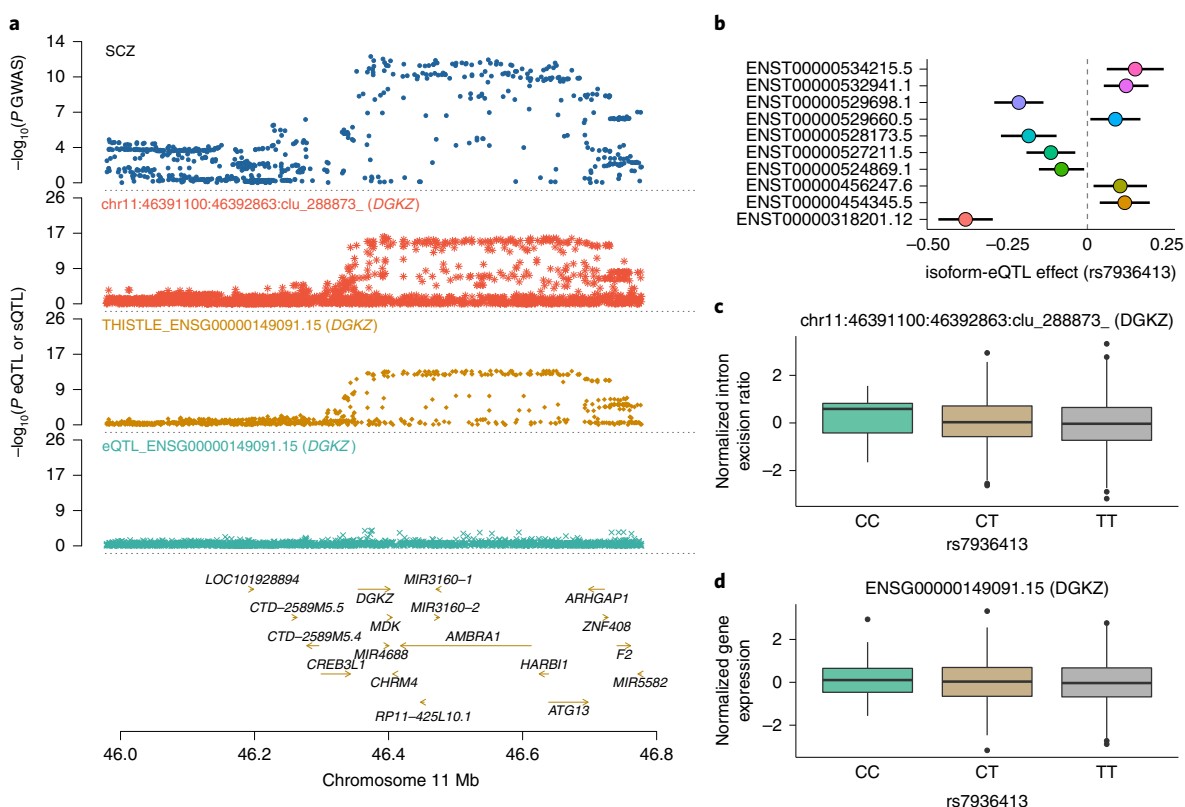

**Fig. 5 | Association of *DGKZ* with SCZ through alternative splicing rather than overall mRNA abundance. a**, GWAS, sQTL and eQTL *P* values. The top track shows −log10(*P*) of SNPs from the SCZ GWAS. The second, third and fourth tracks show −log10(*P*) from the LeafCutter and QTLtools sQTL (intron 11:46391100:46392863:clu_288873_), THISTLE sQTL and eQTL analyses, respectively, for *DGKZ*. The THISTLE sQTL *P* values were computed using a one-sided sum of chi-squared test and the eQTL and LeafCutter and QTLtools sQTL *P* values were computed using a one-sided chi-squared test. **b**, Isoform-eQTL effects for *DGKZ* in the whole dataset (*n* = 2,865), with ENST00000318201.12 and ENST00000534125.5 at the 2 extremes with opposite isoform-eQTL effects. Each dot represents an estimate of the isoform-eQTL effect with an error bar indicating the 95% CI of the estimate. **c,d**, Association of rs7936413 (the lead LeafCutter and QTLtools sQTL SNP) with an excision ratio of 11:46391100:46392863:clu_288873_ and overall mRNA abundance of *DGKZ*, respectively, in the ROSMAP data. Each box plot shows the distribution of intron excision ratios (**c**) or mRNA abundances (**d**) in a genotype class, that is, CC (*n* = 21), CT (*n* = 218) or TT (*n* = 593). The line inside each box indicates the median value, the notches indicate the 95% CI, the central box indicates the IQR, the whiskers indicate data up to 1.5 times the IQR and the outliers are shown as separate dots.

and 19c), indicating their distinct contributions to trait heritability. We also attempted to estimate the proportion of heritability mediated by the *cis*-sQTLs and *cis*-eQTLs ($h^2_{med}/h^2_{SNP}$) by the mediated expression score regression[18] method and observed a median estimate of $h^2_{med}/h^2_{SNP}$ of 0.09 and 0.11 across traits for the *cis*-sQTLs and *cis*-eQTLs, respectively (Fig. 4d), where the estimate for the *cis*-eQTLs was similar to that reported previously[18]. Overall, the analyses above suggest a unique role of sQTLs in complex trait variation, at a level comparable to that of eQTLs (see below for further discussion).

**Identifying complex trait genes using *cis*-sQTL data.** To leverage the *cis*-sQTLs to prioritize functional genes for the 12 brain-related traits, we applied the SMR approach[11] to test if an sGene is associated with a trait through sQTL and the COLOC[9] PP4 statistic to assess whether the sGene-trait association is driven by the same set of causal variants. SMR testing is typically followed by the heterogeneity in dependent instruments (HEIDI) test to distinguish whether the gene-trait association is because of shared or distinct causal variants[11]. However, the HEIDI test requires signed SNP effect estimates, which are unavailable for the THISTLE sQTLs. Instead, we used COLOC PP4, which requires only GWAS and xQTL *P* values as a replacement (Supplementary Fig. 20). We identified 773 sGene-trait associations (585 unique genes) in total for

the 12 traits at a genome-wide significance level ($P_{SMR} < 1.1 \times 10^{-6}$ for the LeafCutter and QTLtools sQTLs and $P_{SMR} < 5.7 \times 10^{-6}$ for the THISTLE sQTLs), 270 (244 unique genes) of which showed a COLOC PP4 value >0.8 (Fig. 4e and Supplementary Table 6), which is consistent with a plausible mechanism that genetic variants affect a trait through genetic control of splicing. We also included the eQTL data in the SMR analysis and identified 805 eGene-trait associations (577 unique genes; $P_{SMR} < 3.2 \times 10^{-6}$), 246 (226 unique genes) of which reached PP4 >0.8 in COLOC (Fig. 4e and Supplementary Table 6). Between the two sets of trait-associated genes discovered through sQTLs and eQTLs, respectively, 96 genes were in common, meaning that we identified 148 more genes through sQTLs on top of the 226 genes identified through eQTLs (Fig. 4e), an approximate 65% increase. One of the examples is *DGKZ* (Fig. 5), the functional relevance of which to schizophrenia (SCZ) has been implicated in previous work[64,65]. In this study, *DGKZ* was associated with SCZ using the sQTL data, implying a possible mechanism that the SNP effects on SCZ are mediated by genetic regulation of RNA splicing of *DGKZ*. Notably, no genome-wide-significant eQTL was associated with the overall expression level of *DGKZ* in our dataset (Fig. 5d), meaning that *DGKZ* would have been missed if we had analyzed the eQTL data only. In addition, the analysis with FOCUS[66] prioritized 298 sGenes for the traits, approximately 80% of which were not identified through eQTLs (Supplementary Fig. 21

and Supplementary Table 7). In summary, approximately 61% (148 out of 244) of the trait-associated sGenes were not detected using eQTL data, which again suggests the distinct role of sQTLs in complex traits. This proportion was similar (approximately 62%) in the analysis with 7 additional traits[67–69] (not limited to the brain) for which summary statistics from large GWAS ($n > 600,000$) were available (Supplementary Fig. 22).

For each of the 96 trait-associated genes that could be identified by using either the sQTL or eQTL data (Fig. 4e), we further performed a COLOC analysis to test whether the sQTL and eQTL signals were driven by the same or distinct causal variants. We found 80 genes with PP4 > 0.80. One typical example is *SETD6*, for which the COLOC analysis (PP4 = 0.98) suggested that the sQTL, eQTL and SCZ GWAS signals are all driven by the same causal variant(s) (Extended Data Fig. 9). On the other hand, there were 8 genes with PP3 > 0.8, indicating multiple GWAS signals at a locus mediated through distinctive genetic regulatory mechanisms (see Supplementary Fig. 23 for a typical example). Note that fewer genes with distinctive sQTL and eQTL signals were linked to the traits than those with a shared sQTL/eQTL signal because it was less likely to link the distinct signals consistently to multiple GWAS signals at a locus. Taken together, our results show the distinctiveness of most sQTLs in mediating the polygenic effects for complex traits and demonstrate the substantial gain of power in gene discovery for complex traits by integrating sQTL data into GWAS.

**Identifying complex trait genes using *trans*-sQTL data.** We further performed *trans*-sQTL and *trans*-eQTL analyses, focusing on SNPs > 5 megabases (Mb) apart or on a different chromosome. After cross-mapping filtering[70], we identified 1,161 unique *trans*-sQTL SNPs with $P < 1.72 \times 10^{-10}$ for 53 *trans*-sGenes by THISTLE and 2,716 *trans*-sQTL SNPs with $P < 2.75 \times 10^{-11}$ for 186 *trans*-sGenes by LeafCutter and QTLtools at 5% FDR, with an overlap of 16 genes (Supplementary Fig. 24a). Of the 223 *trans*-sGenes, 164 were also *cis*-sGenes (Extended Data Fig. 10a). We also identified 15,799 *trans*-eQTL SNPs with $P < 1.72 \times 10^{-10}$ for 230 *trans*-eGenes at 5% FDR (Supplementary Fig. 24b), 33 of which were *trans*-sGenes (Extended Data Fig. 10a). Integrating the *trans*-sQTL/eQTL data with the GWAS data as above, we prioritized 6 sGenes and 11 eGenes, with only 2 genes in common (see Extended Data Fig. 10c,d for one example).

## Discussion

In this study, we generated a comprehensive catalog of genetic variants associated with a broad spectrum of alternative splicing events in the human brain, significantly expanding our understanding of genetic control of RNA splicing. We demonstrated the benefit of using transcript- and event-level sQTL mapping strategies in combination for sQTL detection (Section 8 of the Supplementary Note). By comparing sQTLs with the eQTLs identified in this study, we showed that approximately 61% of sQTLs are distinct from eQTLs, suggesting that sQTL mapping warrants more attention in future research. By integrating sQTLs with GWAS data for 12 brain-related traits, approximately 61% of the trait-associated genes identified through sQTLs could not be discovered through eQTLs, demonstrating the distinct contribution of sQTLs to the genetic architecture underpinning complex trait variation. Moreover, the trait-associated genes identified through sQTLs in this study provide important leads for further mechanistic work to elucidate their functions in the development of the brain-related traits and disorders.

We developed an online tool (https://yanglab.westlake.edu.cn/data/brainmeta) to visualize or download the sQTL and eQTL summary statistics generated in this study. These datasets may be helpful for future studies to understand the molecular mechanisms underpinning the genetic regulation of splicing in the brain,

identify functional genes and variants for other brain-related phenotypes and improve genomic risk prediction. Our study also informs future analyses to quantify the relationship between sGenes and eGenes, between sQTLs and eQTLs or more generally between any two types of molecular QTLs. We showed that the low-to-moderate sGene–eGene overlap as observed in previous studies[6,20] is due to small $n$ because the overlap is a function of $n$ (Fig. 2). For example, only approximately 8% of sGenes are eGenes when $n = 100$ and this proportion increases to 64% as $n$ increases to 1,073. Nevertheless, even for overlapping genes, the underlying sQTL and eQTL causal variants can be distinct. In this study, we estimated that the sQTL and eQTL causal variants were shared (PP4 > 0.8) for only approximately 42% (= 3980/9389) of the overlapping genes (Fig. 2b). This 42% may even be an overestimation because of the limited power of the COLOC PP4 statistic in distinguishing close linkage from sharing, especially when $n$ is not sufficiently large. In an extreme scenario where sQTL and eQTL causal variants are in perfect LD, there is no power to distinguish linkage from sharing.

By integrating the sQTLs with GWAS data, we confirmed that the sQTLs were enriched for genetic variants associated with complex traits, as in previous studies[19,20,32–34]. We note that a previous study by Hormozdiari et al.[21]. showed that sQTLs do not have a significant contribution to disease heritability in a joint analysis of five BLUEPRINT molecular QTLs, namely eQTL, sQTL, H3K27ac histone QTL, H3K4me1 histone QTL and DNA methylation QTL, which is not consistent with our result. The discrepancy is likely due to the small sample sizes of the molecular QTL data in Hormozdiari et al. ($n$ = approximately 200 per cell type). In the present study, we provided multiple lines of evidence that the role of sQTLs in complex trait variation is largely distinct from that of eQTLs. First, approximately 61% of the sQTL and eQTL signals were distinct (Fig. 2). Second, sQTLs contributed significantly to trait heritability conditional on eQTLs (Fig. 4c). Third, approximately 61% of the trait-associated genes detected by integrating GWAS data with sQTLs were not detected by using eQTLs (Fig. 4e). Our results also imply that the contribution of sQTLs in mediating polygenic effects was comparable to that of eQTLs. For example, we observed that the inflation in GWAS test statistics at the sQTLs was indistinguishable from that at the eQTLs (Supplementary Fig. 15), that heritability enrichment of sQTLs was similar to that of eQTLs (median fold enrichment of 2.4 for sQTLs versus 1.8 for eQTLs across traits), that the proportion of mediated heritability for the traits through *cis*-sQTLs was on a par with that through *cis*-eQTLs (median $h^2_{med}/h^2_g$ of 0.09 for sQTLs versus 0.11 for eQTLs across traits) and that the number of trait-associated genes identified through sQTLs was also similar to that through eQTLs (244 versus 226). Hence, large-scale sQTL studies in blood and other tissues or specific cell types are urgently needed to discover more sQTLs to improve our understanding of genetic regulation of RNA splicing and facilitate the translation of GWAS signals into mechanisms.

Despite the potential limitations (Section 9 of the Supplementary Note), our study developed a powerful and flexible sQTL mapping method, generated a comprehensive set of sQTL summary data (with an online tool for data query), demonstrated an analysis paradigm to assess the relationship between two types of molecular QTLs and provided multiple lines of evidence that most sQTLs are distinct from eQTLs, including their roles in complex trait variation.

## Online content

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

## Methods

**Ethical approval.** This study was approved by the Ethics Committee of Westlake University (approval no. 20200722YJ001) and the University of Queensland Human Research Ethics Committee B (approval no. 2011001173).

**THISTLE.** For ease of understanding, let us take a gene with two transcript isoforms as an example. If a genetic variant is associated with a splicing event, the variant is expected to be associated with the difference in mature mRNA abundance between the two transcript isoforms (Fig. 1). Hence, an sQTL test can be performed by assessing the association of the variant with the difference in mRNA abundance between the isoforms, which is equivalent to a test of heterogeneity in isoform-eQTL effect between isoforms. Without loss of generality, let $m$ be the number of transcript isoforms for a gene, $\mathbf{y}_j$ be a vector of mRNA abundances across $n$ individuals for isoform $j$ and $\mathbf{x}$ be a vector of genotypes of a variant. If we define $\hat{\mathbf{b}} = \{\hat{b}_1, \dots \hat{b}_j, \dots, \hat{b}_m\}$ with $\hat{b}_j$ being the estimated isoform-eQTL effect for isoform $j$, we have $\hat{\mathbf{b}} \sim MVN(\mathbf{b}, S)$ with $S$ being the variance-covariance matrix of $\hat{\mathbf{b}}$. The difference between $\hat{b}_j$ and $\hat{b}_k$ ($j \neq k$) can be estimated as:

$$\hat{d}_{jk} = \hat{b}_j - \hat{b}_k$$

If we define $\hat{\mathbf{d}} = \{\hat{d}_{jk}\}_{j,k \in \{1,\dots,m\}, j<k}$, we have $\hat{\mathbf{d}} \sim MVN(\mathbf{d}, V)$ with $V$ being the variance-covariance matrix of $\hat{\mathbf{d}}$. The variance of $\hat{d}_{jk}$ over repeated experiments (that is, a diagonal element of $V$) can be written as:

$$\mathrm{var}\left(\hat{d}_{jk}\right) = \mathrm{var}\left(\hat{b}_j - \hat{b}_k\right) = \mathrm{var}\left(\hat{b}_j\right) + \mathrm{var}\left(\hat{b}_k\right) - 2\mathrm{cov}\left(\hat{b}_j, \hat{b}_k\right)$$

The covariance between $\hat{d}_{jk}$ and $\hat{d}_{gh}$ over repeated experiments (that is, an off-diagonal element of $V$) can be written as:

$$\mathrm{cov}\left(\hat{d}_{jk}, \hat{d}_{gh}\right) = \mathrm{cov}\left(\hat{b}_j - \hat{b}_k, \hat{b}_g - \hat{b}_h\right)$$
$$= \mathrm{cov}\left(\hat{b}_j, \hat{b}_g\right) - \mathrm{cov}\left(\hat{b}_j, \hat{b}_h\right) - \mathrm{cov}\left(\hat{b}_k, \hat{b}_g\right) + \mathrm{cov}\left(\hat{b}_k, \hat{b}_h\right)$$

The covariance between $\hat{b}_j$ and $\hat{b}_k$ can be estimated as $\mathrm{cov}\left(\hat{b}_j, \hat{b}_k\right) = \theta_{jk} S_j S_k$, where $S_j^2$ and $S_k^2$ are the variances of $\hat{b}_j$ and $\hat{b}_k$, respectively and $\theta_{jk}$ is the correlation between $\hat{b}_j$ and $\hat{b}_k$. Since the isoform abundances are measured on the same set of individuals, $\hat{b}_j$ and $\hat{b}_k$ are likely to be correlated (that is, $\theta_{jk} \neq 0$). We know from previous studies[11,71] that $\theta_{jk} \approx r_p \rho$, where $\rho = \frac{n_s}{\sqrt{n_j n_k}}$ measures the sample overlap with $n_j$ and $n_k$ is the sample sizes of isoforms $j$ and $k$, respectively, $n_s$ is the number of overlapping individuals between two isoforms and $r_p$ is the Pearson correlation of mRNA abundances between two isoforms in the overlapping sample. If the individual-level data are unavailable, $\theta_{jk}$ can be approximated by the Pearson correlation of the estimated isoform-eQTL effects between two isoforms across the 'null' SNPs (for example, $P_{\text{isoform-eQTL}} > 0.01$)[11,71]. Under the null hypothesis of no sQTL effect (that is, $\mathbf{d} = 0$), there is no heterogeneity in isoform-eQTL effect between isoforms. In this case, we have a vector of standard normal variables $\mathbf{z}_d = \{z_{d(jk)}\}_{j,k \in \{1,\dots,m\}, j<k}$ with $z_{d(jk)} = \hat{d}_{jk}/\sqrt{\mathrm{var}\left(\hat{d}_{jk}\right)}$ and $\mathbf{z}_d \sim MVN(0, R)$, where $R$ is the correlation matrix with $\mathrm{cor}(z_{d(jk)}, z_{d(gh)}) = \mathrm{cov}\left(\hat{d}_{jk}, \hat{d}_{gh}\right)/\sqrt{\mathrm{var}\left(\hat{d}_{jk}\right)\mathrm{var}\left(\hat{d}_{gh}\right)}$. To test against the null hypothesis ($\mathbf{d} = 0$), we constructed a test statistic $T_{\text{THISTLE}} = \mathbf{z}_d I \mathbf{z}_d^T$ with $I$ being an identity matrix. This statistic is a quadratic form in multivariate normal variables with no explicit distribution, which, however, can be well approximated by the saddlepoint method[41] as in the R4.0.3 function pchisqsum(). We implemented the THISTLE analysis pipeline in the OSCA v.0.45 (ref. [72]) software (https://yanglab.westlake.edu.cn/software/osca/#THISTLE).

**Data used in this study.** We used the genotype and RNA-seq data in brain cortex tissue from seven cohorts, namely BrainGVEX, the Lieber Institute for Brain Development, the CommonMind Consortium and the CommonMind Consortium's National Institute of Mental Health Human Brain Collection Core, Mount Sinai Brain Bank (including four cortex regions: BM10, BM22, BM36 and BM44), Mayo Clinic and ROSMAP. Data generation has been detailed elsewhere[43,44,73]. RNA-seq data from the Mount Sinai Brain Bank cohort were from four brain cortex regions: BM10 (Brodmann area 10; part of the frontopolar prefrontal cortex); BM22 (Brodmann area 22; part of the superior temporal gyrus); BM36 (Brodmann area 36; part of the fusiform gyrus); and BM44 (Brodmann area 44; opercular part of the inferior frontal gyrus). Generation of the genotype and RNA-seq data and imputation of the genotype data have been detailed elsewhere[43,44]. In each cohort, RNA-seq FASTQ data were cleaned using FASTQC and then aligned to the GRCh37 genome assembly by STAR v.2.7.8a[74]. All the transcripts including those of long noncoding RNAs were included in the analysis. Gene-level transcriptional abundances (as measured by read counts) were quantified using RNA-SeQC v.2.3.5 (ref. [75]) and isoform-level transcriptional

abundances (as measured by TPM) were quantified using RSEM v.1.3.1 (ref. [36]) using transcript annotation from GENCODE v.37. Next, RNA-seq data were filtered with a standard quality control process, for example, retaining individuals with more than 10 million total reads and RNA integrity number > 5.5. Genotyped and imputed SNP data were filtered with standard quality control criteria in each cohort using PLINK2 (ref. [76]), that is, genotyping rate > 0.95, Hardy–Weinberg equilibrium test $P > 1 \times 10^{-6}$, MAF > 0.01 and imputation information score > 0.3. We excluded individuals with non-European ancestry as inferred from principal component analysis (Section 4 of the Supplementary Note and Supplementary Fig. 6) because their sample sizes were too small to conduct a cross-ancestry meta-analysis and removed one of each pair of individuals with SNP-derived genetic relatedness > 0.05. After quality control, RNA-seq data from 2,865 brain cortex samples from 2,443 individuals of European ancestry with genetic data of approximately 12 million genotyped or imputed common SNPs were retained for further analysis (Extended Data Fig. 5).

We also included in this study summary statistics from the latest GWAS in samples of European ancestry for 12 brain-related phenotypes, namely intelligence (IQ) ($n = 269,867$)[49], educational attainment (EA) ($n = 766,345$)[50], smoking initiation (311,629 cases and 321,173 controls)[51], neuroticism ($n = 449,484$)[59], age at menarche (AAM) ($n = 370,000$)[52], schizophrenia (SCZ) (69,369 cases and 236,642 controls)[53], AD (71,880 cases and 315,120 controls)[54], PD (33,674 cases and 449,056 controls)[55], insomnia (109,402 cases and 277,131 controls)[56], bipolar disorder (BIP) (41,917 cases and 371,549 controls)[57], amyotrophic lateral sclerosis (ALS) (27,205 cases and 110,881 controls)[60] and major depression (MD) (170,756 cases and 329,443 controls)[58]. IQ was assessed using various neurocognitive tests, primarily gauging fluid domains of cognitive functioning[49]. EA was measured as the number of years of schooling that individuals completed[50]. AAM is a female-specific trait, referring to the age when periods start. Neuroticism was measured with 12 dichotomous items of the Eysenck Personality Questionnaire Revised-Short Form[77].

**Identification of *cis*-sQTLs using THISTLE and LeafCutter.** The workflow of the sQTL and eQTL analyses is illustrated in Extended Data Fig. 5, which largely follows the standard pipeline for cohort-based RNA-seq data analysis in the literature[6]. To identify sQTLs using THISTLE, we filtered out isoforms with low expression (that is, isoform-level TPM < 0.1 in more than 80% of the samples) and performed quantile normalization of TPM values across all transcripts in each brain cortex sample. VariancePartition[78] was employed to decompose the variation in isoform abundance into components attributable to multiple known biological and technical factors such as study, RNA quality (RNA integrity number) and age at death; probabilistic estimation of expression residuals (PEER)[79] was used to generate a set of latent covariates (also known as PEER factors) that can capture variation due to hidden factors. The isoform-level transcriptional abundance after adjusting for the factors identified by VariancePartition and the PEER factors was standardized by a rank-based inverse normal transformation (RINT). Note that as in the GTEx study[6], the number of PEER factors used for the adjustment was determined based on the sample size ($n$) of each dataset: 15 for $n < 150$; 30 for $150 \leq n < 250$; 45 for $250 \leq n < 350$; and 60 for $n \geq 350$. Isoform abundance after adjustment for selected biological/ technical factors and PEER factors was used for a linear regression analysis to detect isoform-eQTLs in each RNA-seq dataset, with the first five genetic principal components fitted as covariates. The isoform-eQTL summary statistics from the ten datasets were meta-analyzed by MeCS[71], which can account for correlations of estimation errors of the isoform-eQTL effects between datasets, followed by an sQTL analysis with THISTLE. We excluded genes with only one isoform and limited the *cis*-sQTL test to SNPs within 2 Mb of each gene on either side. To identify eQTLs, we applied a similar quality control and covariate adjustment pipeline as above to gene-level expression data, that is, excluding genes with TPM < 0.1 or read count < 6 in more than 80% of the samples, trimmed mean of the M-values normalization, preadjusting for covariates identified by VariancePartition and PEER factors, and RINT (Extended Data Fig. 5). A linear regression model was applied to the standardized gene-level expression data to test for eQTLs, with the first five genetic principal components fitted as covariates, followed by a meta-analysis of the eQTL summary statistics across the ten datasets by MeCS.

To identify sQTLs with LeafCutter v.0.2.9 (ref. [31]), we aligned the RNA-seq reads of each sample to GRCh37 by STAR v.2.7.8a[74], with the wasp flag to leverage SNP genotype data to remove mapping biases caused by allele-specific reads[80]. The alignment results from all the samples across datasets were used as input for LeafCutter to identify excised intron clusters, with default parameters. In each dataset, the intron excision ratio (the ratio of the reads defining an excised intron to the total number of reads of an intron cluster) was quantile-normalized within each sample and then standardized across samples. In total, 273,051 excised introns in 47,600 intronic excision clusters were identified with 43,774 clusters (92.0%) uniquely mapped to 14,085 genes (using the R function map_clusters_to_genes() provided by LeafCutter based on transcript annotation from GENCODE v.37), 2,060 clusters (4.3%) mapped to more than 1 gene and 1,766 unmapped clusters (3.7%). The proportions of variance in intron excision ratio explained by the known biological/

technical factors were much smaller than those for isoform abundance above (Supplementary Fig. 7), probably because the biological/technical factors affected both the numerator and denominator of the intron excision ratio so that their effects largely canceled out each other. As above, the intron excision ratio after adjustment for known biological and technical factors identified by VariancePartition and 15 PEER factors and RINT was tested for associations with SNPs within 2 Mb of each intron in each RNA-seq dataset using linear regression models implemented in QTLtools, with the first 5 genetic principal components fitted as covariates, followed by a meta-analysis of the sQTL summary statistics across the 10 RNA-seq datasets by MeCS.

**Enrichment of sQTLs and eQTLs for functional annotations.** To test if sQTLs and eQTLs are functionally enriched, we annotated the lead *cis*-sQTL or *cis*-eQTL SNPs using annotation data from SnpEff[45] (for example, splice region, intronic and upstream), eCLIP peaks of 113 RBP[46] binding sites from the HepG2 and K562 cell lines from the ENCODE project[81], chromatin immunoprecipitation followed by sequencing peaks for 7 histone modifications (that is, H3K4me1, H3K4me3, H3K9ac, H3K9me3, H3K27ac, H3K27me3 and H3K36me3) and 15 chromatin states (for example, active TSS and enhancer) from the brain cortex sample (E073) of the Roadmap Epigenomics Mapping Consortium (REMC)[47]. More specifically, we annotated 12,578 and 16,086 unique lead *cis*-sQTL and *cis*-eQTL SNPs, respectively in different functional annotation categories based on their physical positions and quantified the proportion of sQTL or eQTL SNPs in each category. To ameliorate ascertainment bias, we sampled at random the same number of *cis*-SNPs (that is, SNPs included in the sQTL or eQTL analysis) as 'control SNPs', with their MAF and distance to TSS matched with the SNPs in query. This sampling procedure was repeated 1,000 times. We computed the fold enrichment in each functional annotation category as the ratio of the proportion of sQTL (or eQTL) SNPs in a functional category over the mean proportion of the control SNPs in the category across 1,000 replicates. The sampling variance of the fold enrichment can be calculated approximately by the Delta method[82] (Section 7 of the Supplementary Note).

**Enrichment of sQTLs and eQTLs for trait heritability.** The stratified LD score regression[61,62] was used to quantify the enrichment of heritability attributable to sQTLs and eQTLs (when fitted together with 53 other functional categories in the 'baseline-LD model') for the 12 brain-related traits. Details of the baseline-LD model can be found elsewhere[61]. We created a binary annotation for sQTLs and eQTLs, respectively. In brief, we assigned an annotation value of 1 for the most significant sQTL (or eQTL) SNP for each gene and a zero value for the remaining SNPs, resulting in an sQTL annotation category with 10,416 SNPs and an eQTL annotation category with 14,118 SNPs with an overlap of 1,455 SNPs. The LD scores of the SNPs were computed using SNP genotype data of the individuals of European ancestry from the 1000 Genomes Project (phase 3)[83] with a window size of 1 cM. Heritability enrichment of a category was computed as the proportion of heritability explained by the category divided by the proportion of SNPs in the category. Considering that SNPs in or near genes explain disproportionately more trait variation than intergenic SNPs[63], we also computed the fold enrichment of heritability as the per-SNP heritability for the lead *cis*-sQTL (or *cis*-eQTL) SNPs divided by a mean of a distribution generated by resampling MAF- and TSS-matched *cis*-SNPs. Sampling variance of the fold enrichment of heritability can be calculated approximately by the Delta method[82] (Section 7 of the Supplementary Note). Note that both the per-SNP heritability and parameter τ reported by stratified LD score regression were used to quantify the relevance of a functional category to the trait variation[62] and the main difference between the two metrics lies in how the overlapping annotations were dealt with. More specifically, τ is the partial regression coefficient for an annotation category when fitted jointly with the other annotation categories in an LD score regression model. If all the annotation categories are disjoint (no overlapping SNP among categories), τ can be interpreted as the per-SNP heritability of the corresponding annotation category. In the presence of overlaps among the annotation categories, the interpretation of τ is complicated. However, it can still be used to quantify the contribution of an annotation category to the overall SNP-based heritability, conditioning on the other categories.

**Statistics and reproducibility.** The THISTLE and sQTLseekeR *P* values were computed using a one-sided sum of chi-squared test (approximated by the saddlepoint algorithm) and pseudo-*F* test (approximated by the Davies algorithm), respectively. The eQTL and LeafCutter and QTLtools sQTL *P* values were computed using a one-sided chi-squared test. We used 2,443 unrelated individuals of European ancestry for real-data analysis. The sample size was determined by the maximum number of unrelated individuals of European ancestry with both SNP genotype and RNA-seq data. We excluded individuals of non-European ancestry, with <10 million total reads or with an RNA integrity number <5.5 and 1 of each pair of individuals with genetic relatedness >0.05. Standard quality control criteria were applied to clean genetic variants to avoid the inclusion of low-quality variants in the association analyses. We did not use any study design that required randomization or blinding. The scripts to reproduce the main results of this paper are available at Zenodo[84].

**Reporting summary.** Further information on research design is available in the Nature Research Reporting Summary linked to this article.

## Data availability

The PsychENCODE data are available at https://www.synapse.org/#!Synapse:syn4921369. The AMP-AD data are available at https://www.synapse.org/#!Synapse:syn5550382. The online tool for querying the sQTL and eQTL summary statistics is available at https://yanglab.westlake.edu.cn/data/brainmeta. The full summary statistics from the sQTL, eQTL, SMR and COLOC analyses are available at https://yanglab.westlake.edu.cn/pub_data.html. The GRCh37 genome assembly is available at https://www.ncbi.nlm.nih.gov/genome/guide/human. The GENCODE-v37 transcriptome reference is available at https://www.gencodegenes.org/human/release_37lift37.html. Source data are provided with this paper.

## Code availability

The computer code and documentation of THISTLE are available at https://yanglab.westlake.edu.cn/software/osca/#THISTLE, with source code available at https://github.com/jianyangqt/osca. All custom codes used to perform the data analysis relevant to this paper, including RNA-seq data cleaning, sQTL and eQTL mapping, functional enrichment analyses and SMR and COLOC analyses, are available at Zenodo[84].

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

## Acknowledgements

This research was partly supported by the Pioneer and Leading Goose R&D Program of Zhejiang (no. 022SDXHDX0001, J.Y.), the Leading Innovative and Entrepreneur Team Introduction Program of Zhejiang (no. 2021R01013, J.Y.), the National Natural Science Foundation of China (no. 32100493, T.Q.), the Westlake Education Foundation (no. 101566022001, J.Y.), the Australian National Health and Medical Research Council (nos. 1107258 and 1113400, J.Y.) and the Australian Research Council (nos. DP160101343 and FT180100186, J.Y.). We thank the High-Performance Computing Center and the Research Center for Industries of the Future at Westlake University for assistance in computing. This study uses data from the PsychENCODE (Synapse accession no. syn4921369) and Accelerating Medicines Partnership Program for Alzheimer's Disease (AMP-AD) (Synapse accession: syn5550382) consortia. A full list of acknowledgements to the data can be found in Section 10 of the Supplementary Note.

## Author contributions

J.Y. conceived and supervised the study. T.Q. and J.Y. designed the experiment and developed the method. T.Q. performed all the simulations and data analyses under assistance or guidance from Y.W., J.Z. and J.Y. T.Q., H.F., F.Z. and S.L. developed the application tools. T.Q. and J.Y. wrote the manuscript with the participation of all authors.

## Competing interests

The authors declare no competing interests.

## Additional information

**Extended data** is available for this paper at https://doi.org/10.1038/s41588-022-01154-4.

**Correspondence and requests for materials** should be addressed to Jian Yang.

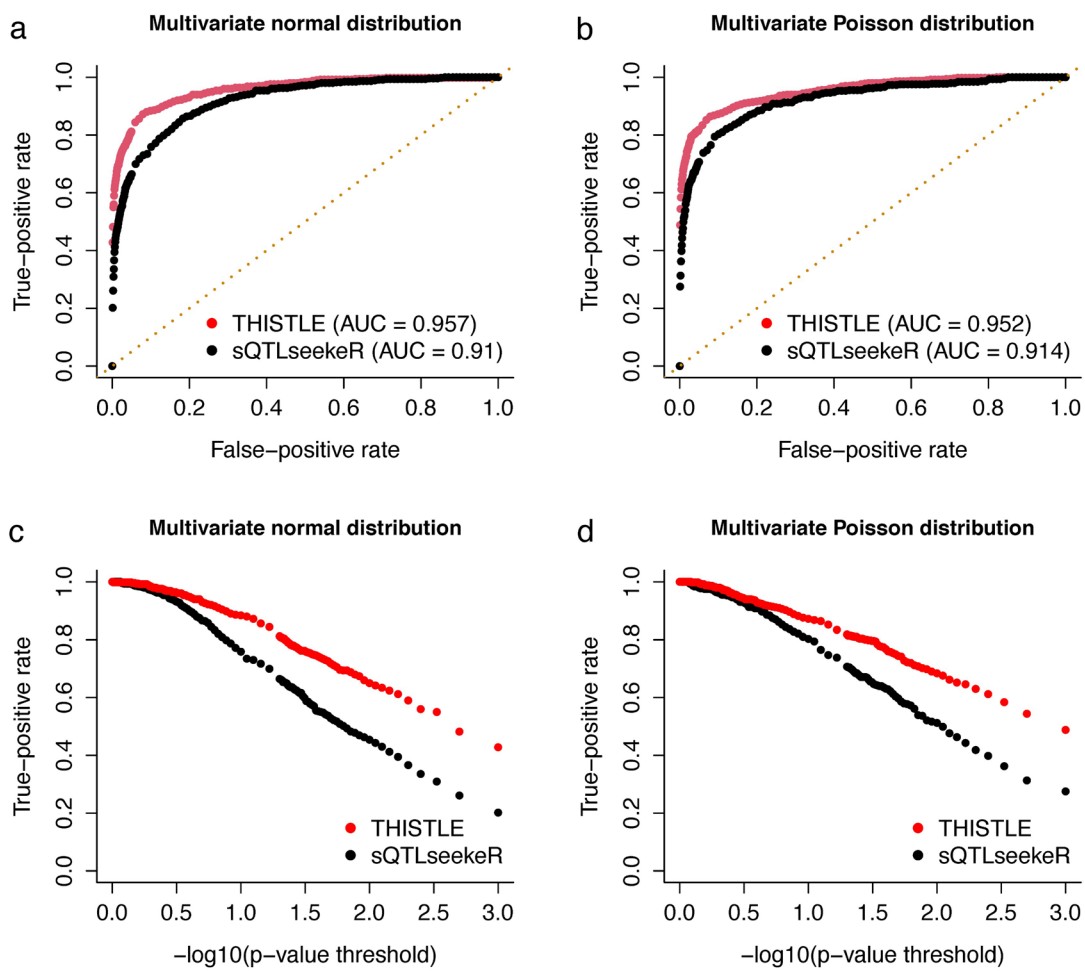

**Extended Data Fig. 1 | Comparison of statistical power between THISTLE and sQTLseekeR by simulation with transcription abundances generated from a multivariate distribution.** Transcription abundance was simulated from either a multivariate normal distribution (panels **a** and **c**) or a multivariate Poisson distribution (panels **b** and **d**) with (scenario 4) or without an eQTL effect (scenario 3) in a sample of 500 unrelated individuals (Section 2 of the Supplementary Note). The THISTLE and sQTLseekeR p-values were computed from a one-sided sum of chi-squared test (approximated by the Saddlepoint algorithm) and pseudo-$F$ test (approximated by the Davies algorithm), respectively.

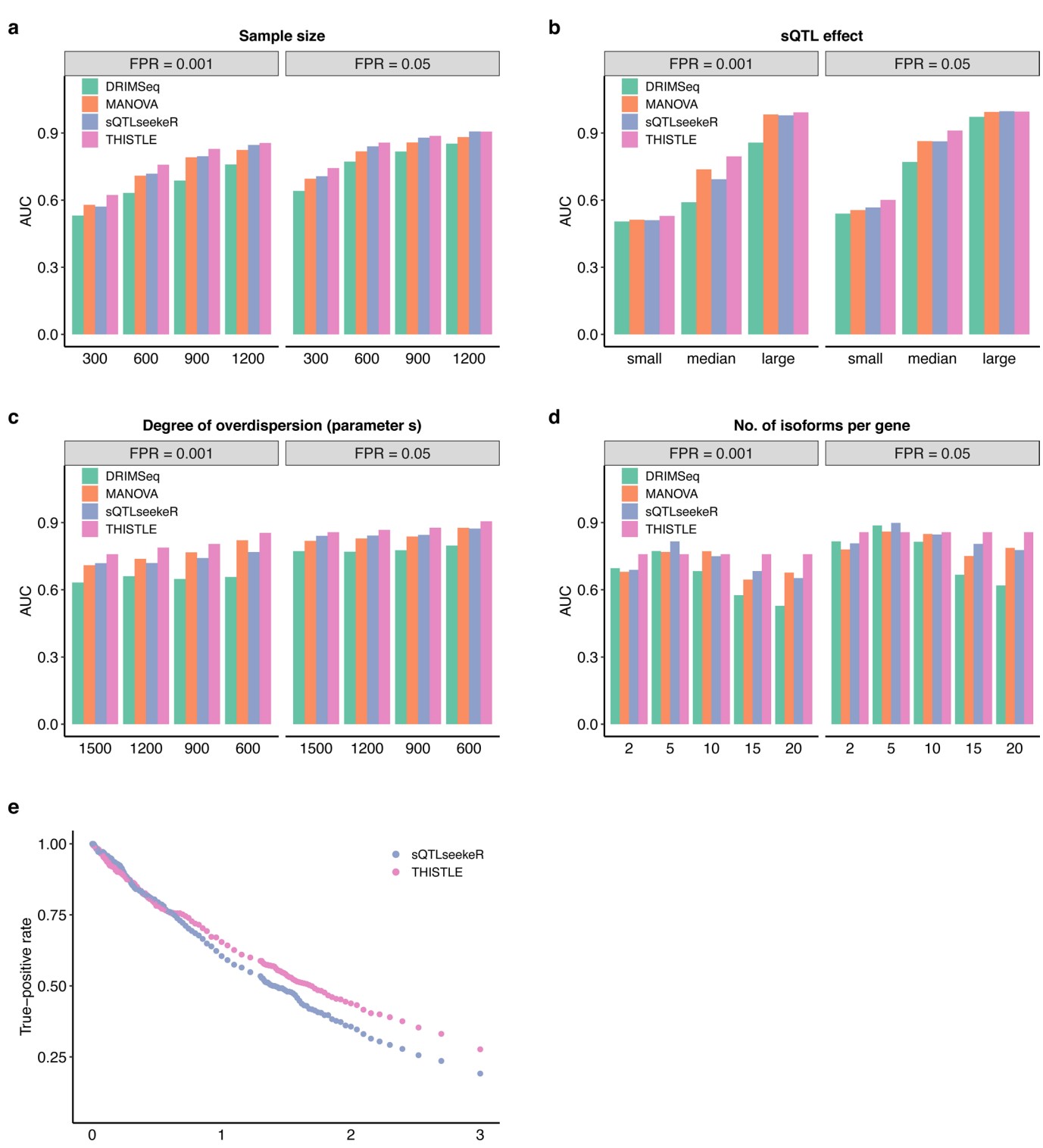

**Extended Data Fig. 2 | Comparison of statistical power between DRIMSeq, MANOVA, sQTLseekeR, and THISTLE by simulation with transcription abundances generated from RNA-seq reads.** Panels **a**, **b**, **c** and **d** show the results from 500 simulation replicates to quantify the power, measured by AUC truncated at FPR = 0.05 or 0.001, for DRIMSeq, and MANOVA, sQTLseekeR, and THISTLE. Panel **e** shows the result from 500 simulation replicates to quantify the power, measured by TPR at different levels of p-value threshold, for sQTLseekeR and THISTLE. Transcriptional abundance data of 500 unrelated individuals were generated by sampling RNA-seq reads to mimic real RNA-seq data using Polyester (Section 2 of the Supplementary Note). The simulations were performed with varying **a**) sample size, **b**) sQTL effect size, **c**) the degree of overdispersion of transcription abundances, and **d**) the number of isoforms per gene. When one simulation parameter was fixed (for example, $n = 300$ in panel **a**, all the other parameters were randomly sampled from the specified categories, for example, the sQTL effect size from {small, median, or large}, the number of isoforms per gene from {2, 5, 10, 15, or 20}, and the degree of over-dispersion of transcriptional abundance from {600, 900, 1200, or 1500}. The DRIMSeq, MANOVA, sQTLseekeR, and THISTLE p-values were computed using a one-sided likelihood ratio test, $F$ test, pseudo-$F$ test, and sum of chi-squared test, respectively.

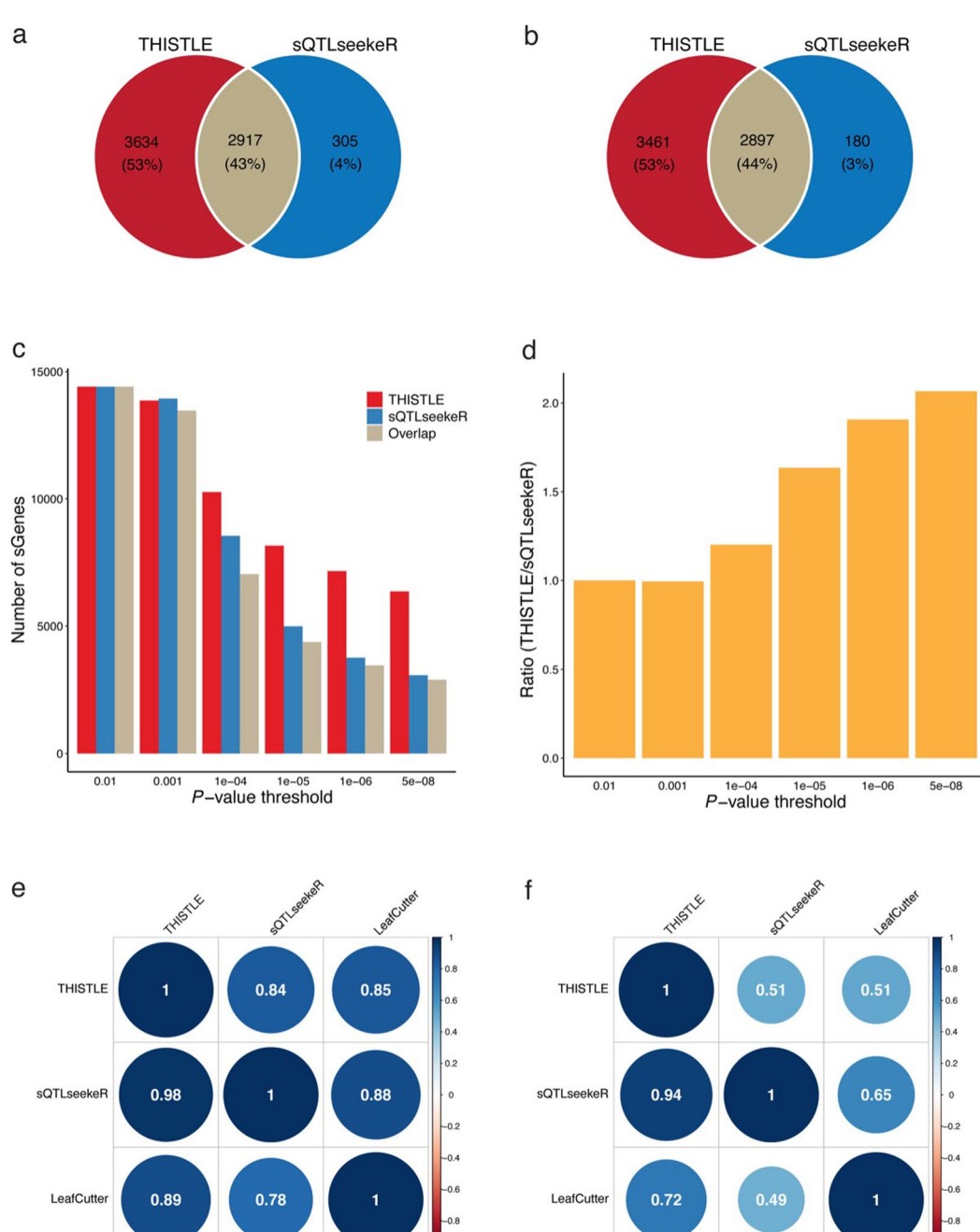

**Extended Data Fig. 3 | Comparison between sQTL results from the sQTLseekeR, THISTLE, and LeafCutter & QTLtools analyses of real data.** The analyses were performed with the ROSMAP data ($n = 832$). Panel **a** shows the numbers of overlapping sGenes between THISTLE and sQTLseekeR. Panel **b** shows the numbers of overlapping sGenes considering only the SNP-gene pairs tested in common between THISTLE and sQTLseekeR. Panel **c** shows the number of sGenes discovered at different p-value thresholds for THISTLE and sQTLseekeR. Panel **d** shows the ratio of the number of sGenes identified by THISTLE to that by sQTLseekeR at different p-value thresholds. Panels **e** and **f** show the replication rates of the lead cis-sQTL SNPs at $P_{sQTL} < 0.05$ and $P_{sQTL} < 0.05/m$ (where $m$ is the number of SNPs taken forward for replication for each method), respectively. Each row represents an analysis from which the lead cis-sQTL SNPs (one SNP per gene) were identified ($P < 5 \times 10^{-8}$), and each column represents an analysis in which the SNPs were replicated. The THISTLE, sQTLseekeR, and LeafCutter & QTLtool p-values were computed using a one-sided sum of chi-squared test, pseudo-$F$ test, and chi-squared test, respectively.

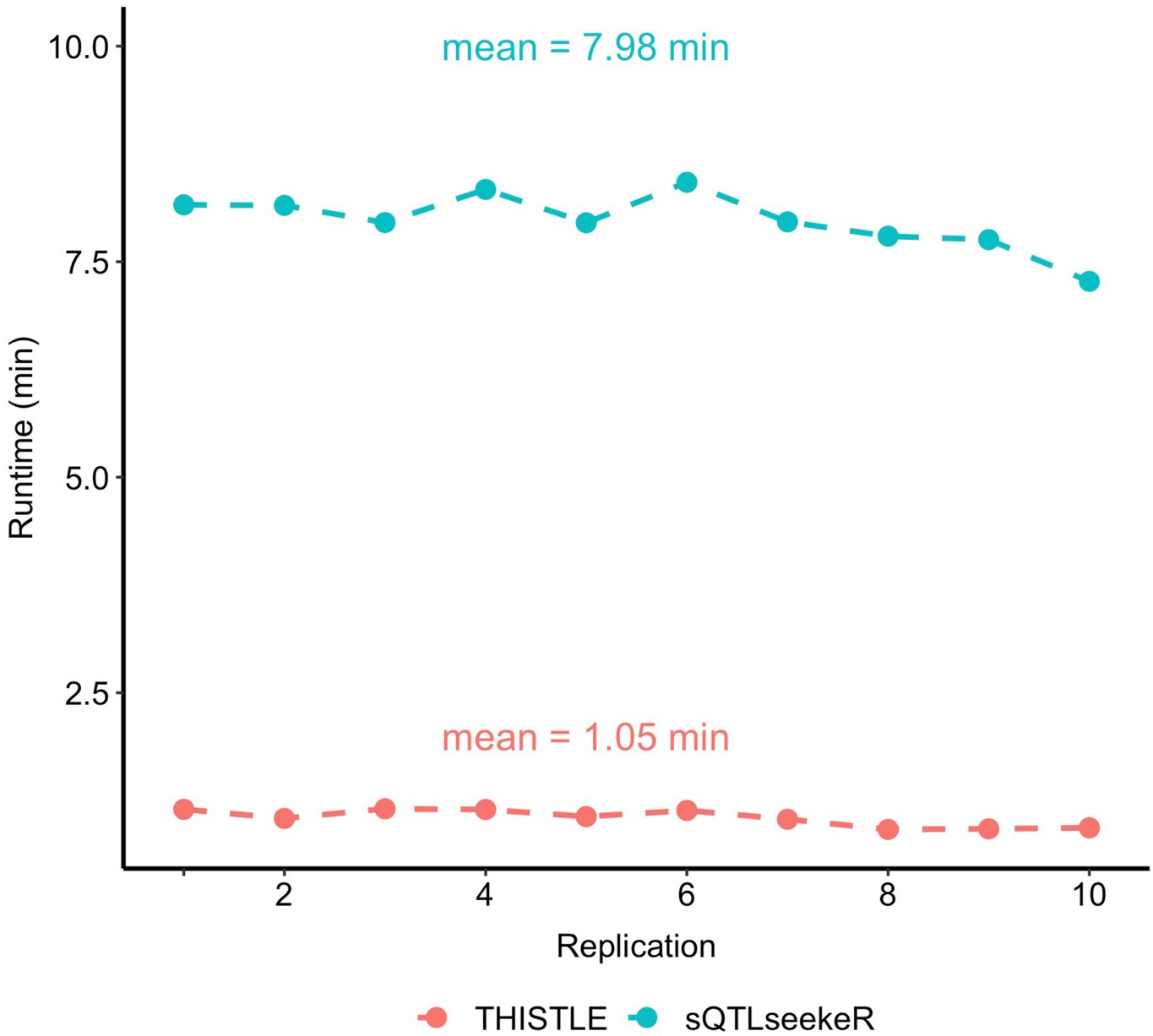

**Extended Data Fig. 4 | Comparison of runtime between THISTLE and sQTLseekeR.** We benchmarked the runtime of THISTLE (implemented in OSCA) on a computing platform with 16 GB memory and 16 CPU cores, in comparison with sQTLseekeR2-nf, using the ROSMAP data ($n$ = 832) with 382 genes and 109,853 SNPs on chromosome 22. We repeated each analysis with 10 replicates.

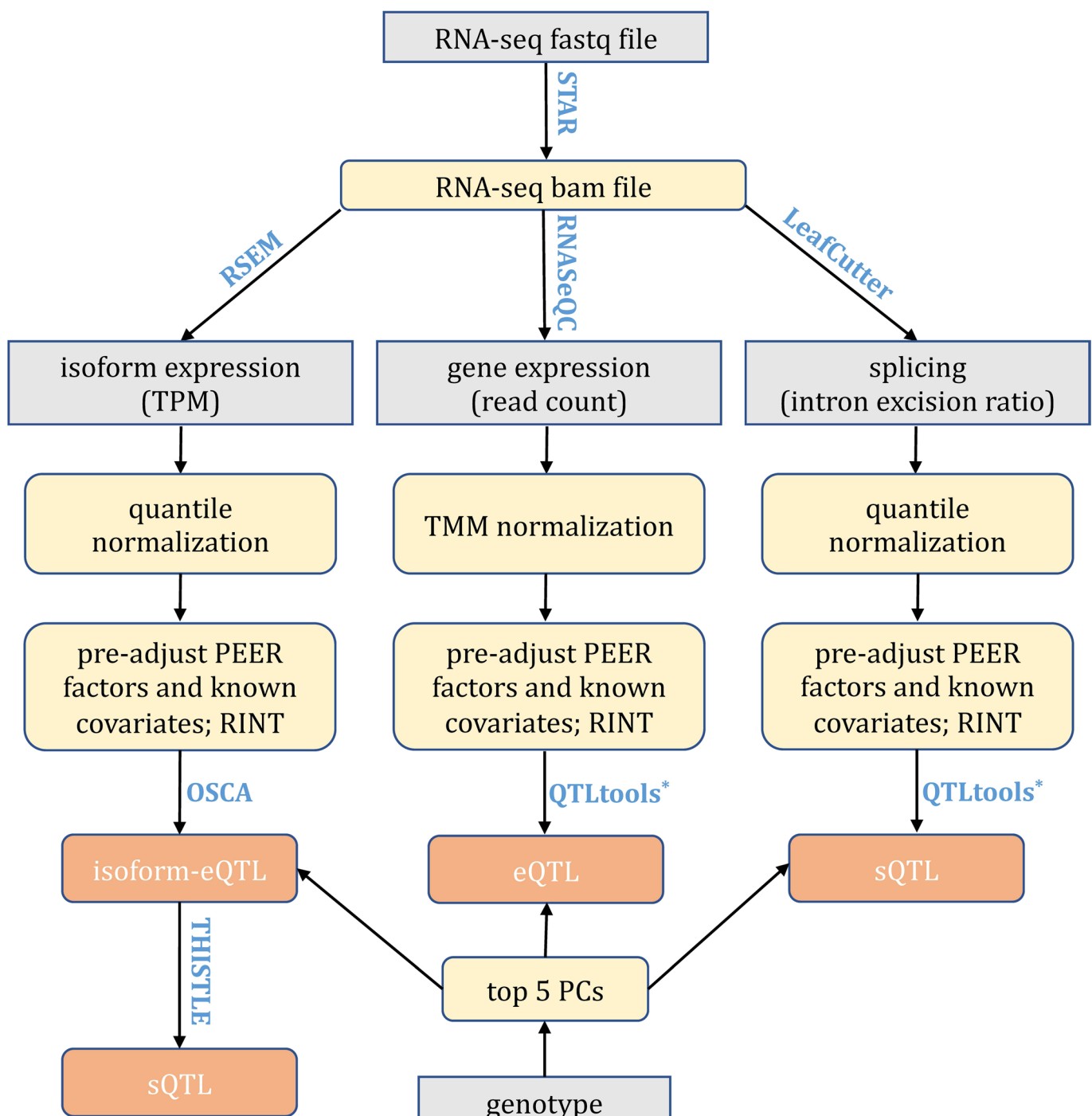

**Extended Data Fig. 5 | Workflow of the sQTL and eQTL analyses in this study.** RINT: rank-based inverse normal transformation. * Note: we have implemented the QTLtools QTL mapping method (linear regression) in OSCA and used the two tools interchangeably, for example, OSCA for real data analysis to save the sQTL or eQTL mapping results in SMR BESD format and QTLtools for eQTL or Leafcutter sQTL permutation analysis.

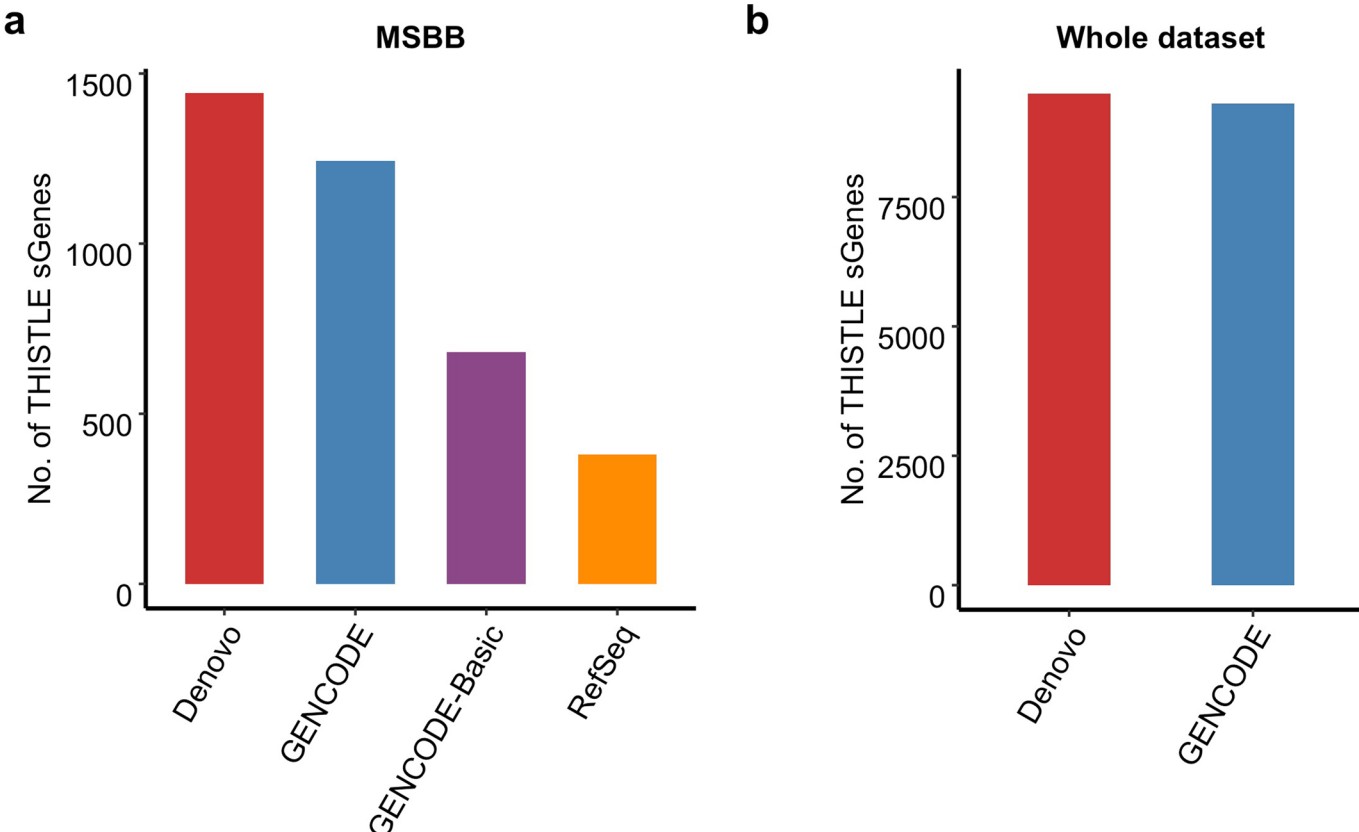

**Extended Data Fig. 6 | Number of sGenes identified by THISTLE based on four different transcriptome references.** The four transcriptome references are RefSeq, GENCODE-Basic (v37), GENCODE (v37), and de novo assembly (constructed from the RNA-seq data using StringTie based on GENCODE). Panel **a** shows the comparison in the number of sGenes in the MSBB cohort ($n = 183$), and panel **b** shows the comparison in the whole dataset ($n = 2,865$).

a

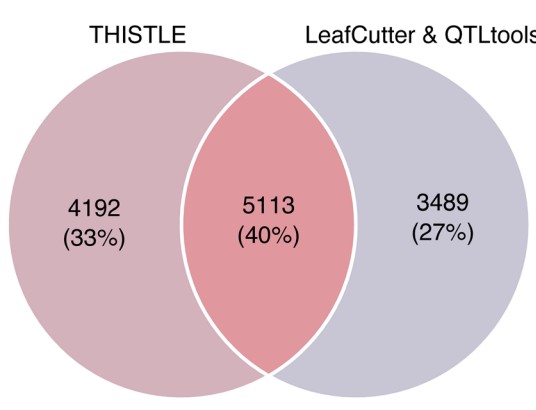

b

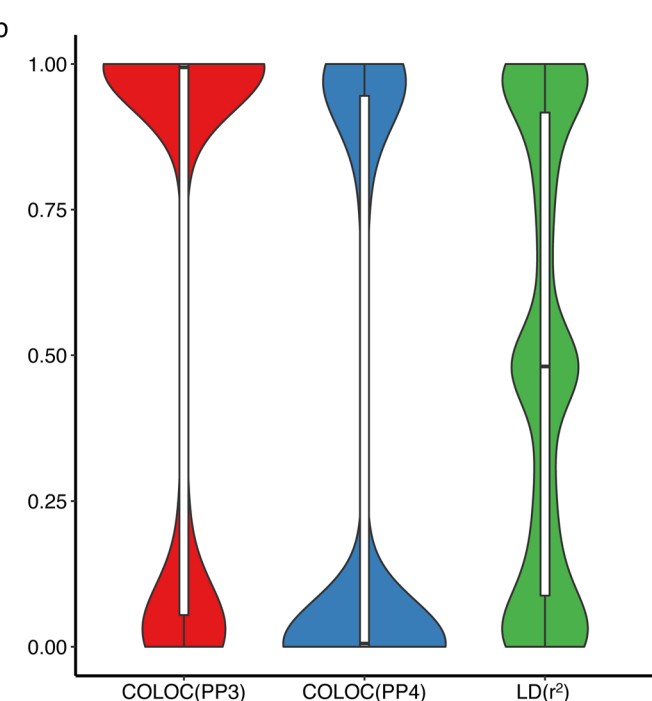

**Extended Data Fig. 7 | Comparison between the sQTLs results from THISTLE and LeafCutter & QTLtools. a**) Comparison between the sGenes identified by THISTLE and LeafCutter & QTLtools. **b**) COLOC PP3 and PP4 values between the THISTLE and LeafCutter & QTLtools sQTL signals, and LD $r^2$ between the lead THISTLE and LeafCutter & QTLtools cis-sQTL SNPs for 5,113 overlapping sGenes. In panel **b**, the bold line inside each box indicates the median value, notches indicate the 95% confidence interval (CI), the central box indicates the interquartile range (IQR), and whiskers indicate data up to 1.5 times the IQR.

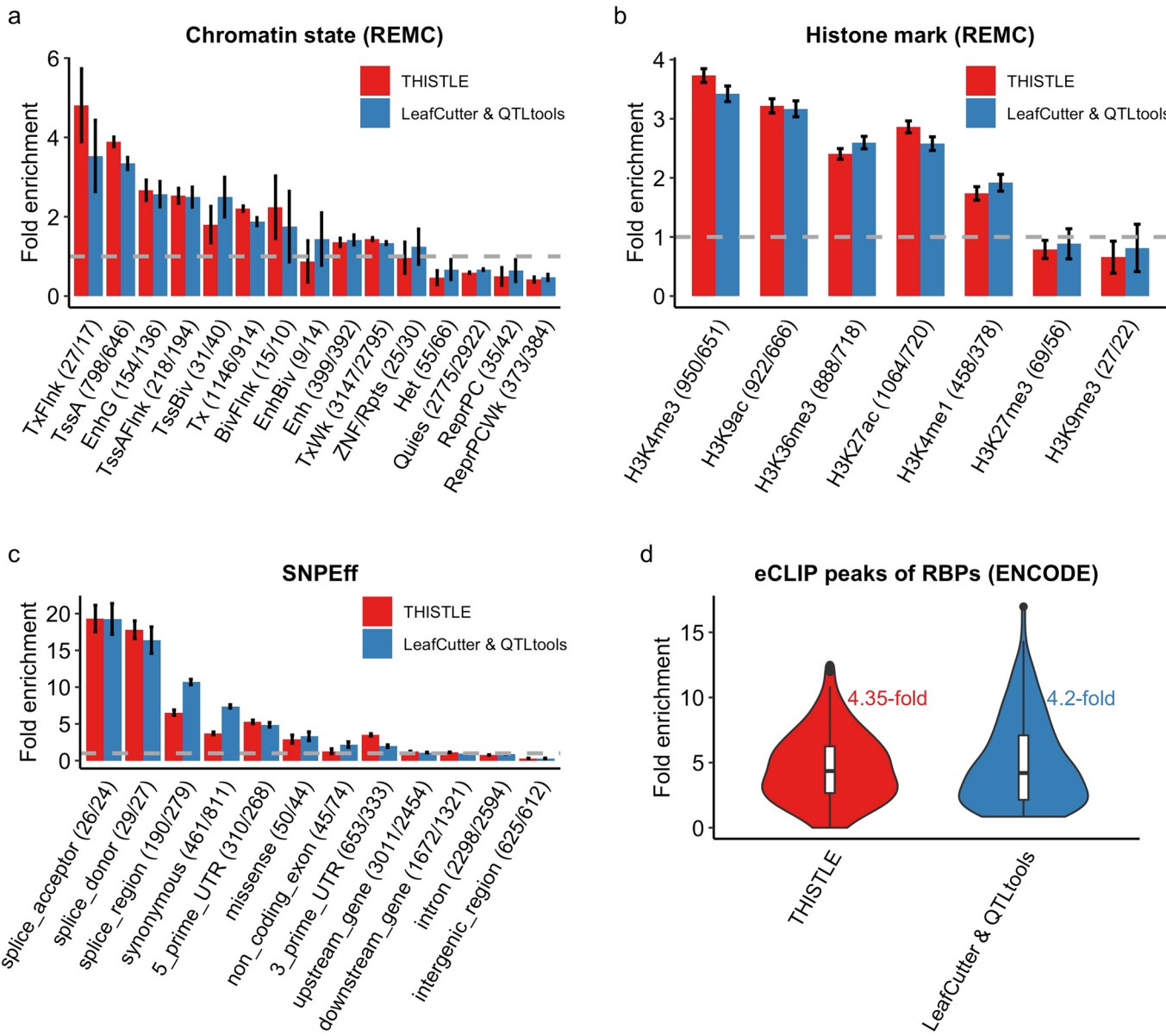

**Extended Data Fig. 8 | Enrichment of the lead THISTLE or LeafCutter & QTLtools cis-sQTL SNPs in functional annotation categories.** The annotation categories were defined by the chromatin state annotation data from REMC (**a**), histone marks from REMC (**b**), predicted variant functions by SNPEff (**c**), or eCLIP peaks of 113 RBPs binding sites from ENCODE (**d**). The fold enrichment was computed by dividing the percentage of lead cis-sQTL SNPs in a category by the mean percentage observed in 1,000 sets of control SNPs sampled repeatedly at random (Methods). Each column represents a point estimate with an error bar indicating the 95% CI of the estimate. The grey dashed line represents no enrichment. The numbers in the parentheses are the numbers of THISTLE/LeafCutter & QTLtools sQTL SNPs in each functional category. In panel **d**, a violin plot shows the distribution of fold enrichment estimates across 113 RBPs binding sites. The line inside each box indicates the median value, notches indicate the 95% CI, the central box indicates the IQR, whiskers indicate data up to 1.5 times the IQR, and outliers are shown as separate dots.

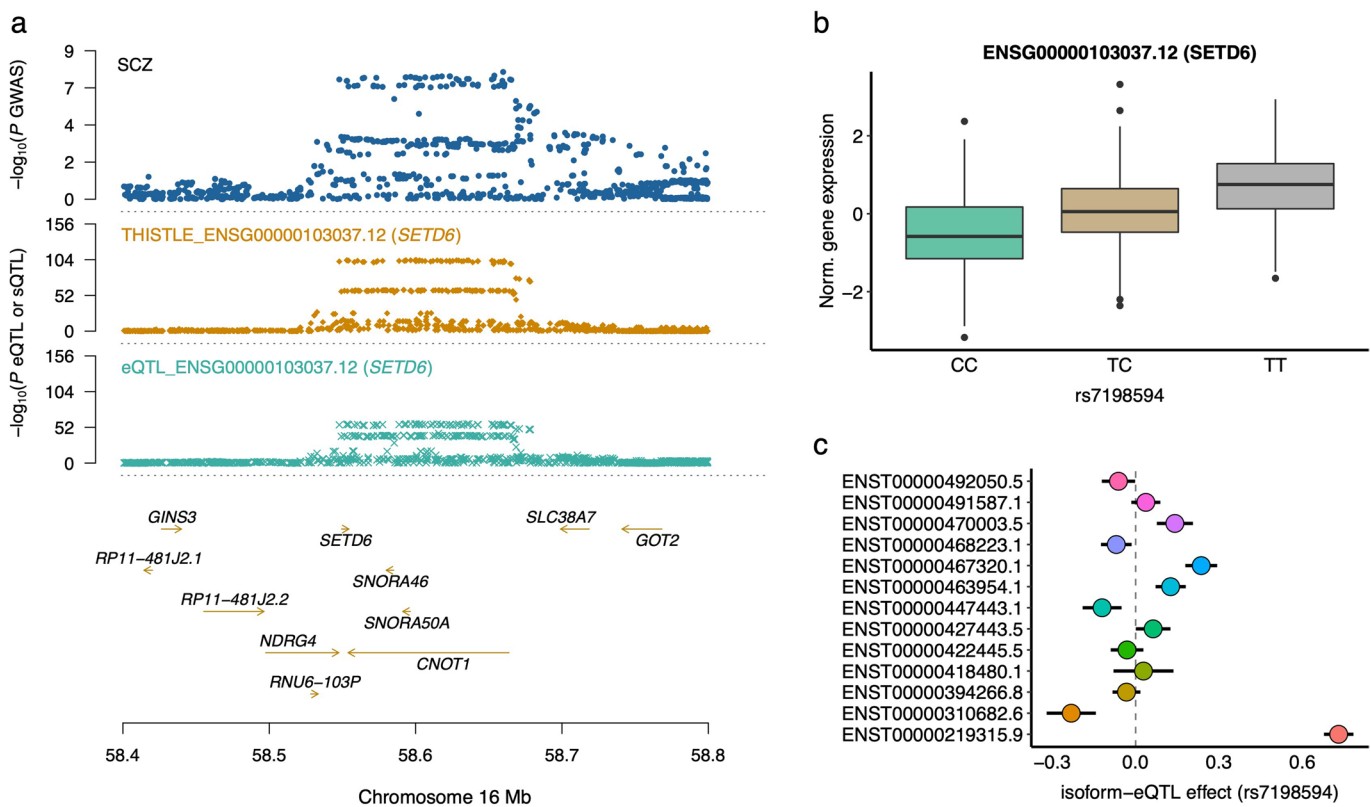

**Extended Data Fig. 9 | Association of *SETD6* with schizophrenia identified using either sQTL or eQTL data. a**) The top track shows −log10(p-values) of SNPs from the schizophrenia GWAS. The second and third tracks show −log10(p-values) from the THISTLE sQTL and eQTL analyses, respectively. The THISTLE sQTL and eQTL p-values were computed using a one-sided sum of chi-squared test and chi-squared test, respectively. **b**) Association of rs7198594 (the lead eQTL SNP) with the overall mRNA abundance of *SETD6* in the ROSMAP data (*n* = 832). Each boxplot shows the distribution of mRNA abundances in a genotype class, that is, CC (*n* = 288), CT (*n* = 392), or TT (*n* = 152). The line inside each box indicates the median value, notches indicate the 95% CI, the central box indicates the IQR, whiskers indicate data up to 1.5 times the IQR, and outliers are shown as separate dots. **c**) Isoform-eQTL effects for *SETD6* in the whole dataset (*n* = 2,865), with ENST00000310682.6 and ENST00000219315.9 at the two extremes with opposite isoform-eQTL effects. Each dot represents an estimate of isoform-eQTL effect with an error bar indicating the 95% CI of the estimate.

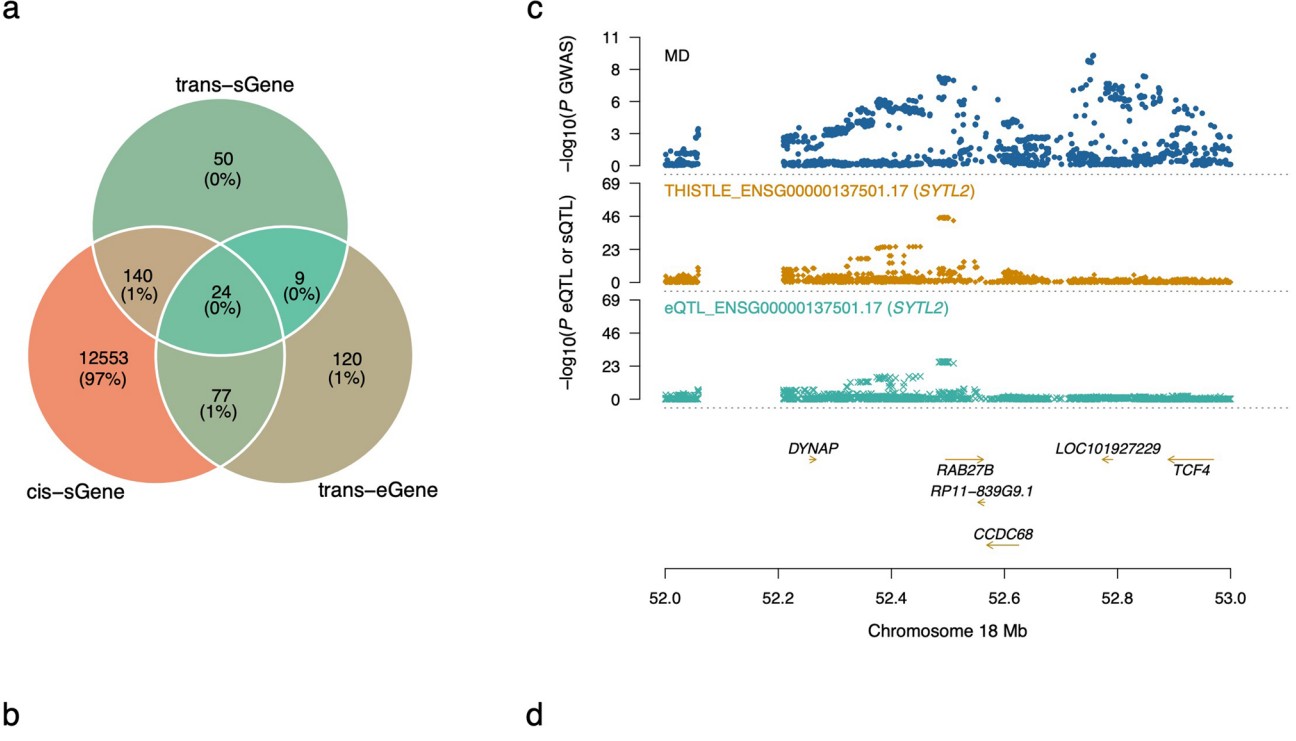

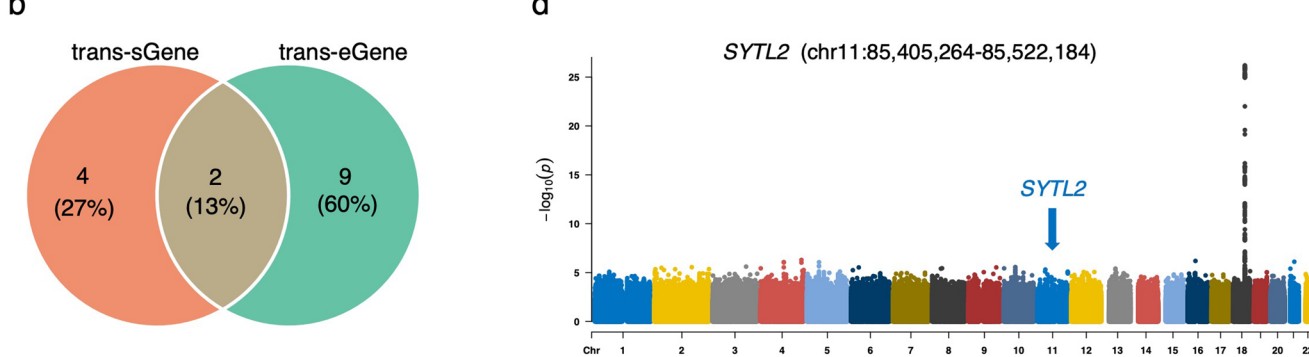

**Extended Data Fig. 10 | Trans-sQTL analysis. a)** Overlaps between trans-sGenes, cis-sGenes, and trans-eGenes. **b)** Overlap of the trait-associated trans-sGenes and trans-eGenes identified by SMR & COLOC PP4. **c)** Association of *SYTL2* with major depression (MD) through both the trans-sQTLs and trans-eQTLs. The top track shows −log10(p-values) of SNPs from the MD GWAS. The second and third tracks show −log10(p-values) from the THISTLE trans-sQTL and trans-eQTL analyses, respectively, for *SYTL2*. **d)** Manhattan plot of p-values from genome-wide THISTLE sQTL analysis for *SYTL2*. The blue arrow indicates the genomic position where *SYTL2* is located. The GWAS and eQTL p-values in panel **c** were computed using a one-sided chi-squared test, and the THISTLE sQTL p-values in panels **c** and **d** were computed using a one-sided sum of chi-squared test.

# Reporting Summary

Nature Research wishes to improve the reproducibility of the work that we publish. This form provides structure for consistency and transparency in reporting. For further information on Nature Research policies, see our Editorial Policies and the Editorial Policy Checklist.

## Statistics

For all statistical analyses, confirm that the following items are present in the figure legend, table legend, main text, or Methods section.

| n/a | Confirmed | |
|---|---|---|
| ☐ | ☒ | The exact sample size (*n*) for each experimental group/condition, given as a discrete number and unit of measurement |
| ☒ | ☐ | A statement on whether measurements were taken from distinct samples or whether the same sample was measured repeatedly |
| ☐ | ☒ | The statistical test(s) used AND whether they are one- or two-sided<br>*Only common tests should be described solely by name; describe more complex techniques in the Methods section.* |
| ☐ | ☒ | A description of all covariates tested |
| ☐ | ☒ | A description of any assumptions or corrections, such as tests of normality and adjustment for multiple comparisons |
| ☐ | ☒ | A full description of the statistical parameters including central tendency (e.g. means) or other basic estimates (e.g. regression coefficient) AND variation (e.g. standard deviation) or associated estimates of uncertainty (e.g. confidence intervals) |
| ☐ | ☒ | For null hypothesis testing, the test statistic (e.g. *F*, *t*, *r*) with confidence intervals, effect sizes, degrees of freedom and *P* value noted<br>*Give P values as exact values whenever suitable.* |
| ☒ | ☐ | For Bayesian analysis, information on the choice of priors and Markov chain Monte Carlo settings |
| ☒ | ☐ | For hierarchical and complex designs, identification of the appropriate level for tests and full reporting of outcomes |
| ☐ | ☒ | Estimates of effect sizes (e.g. Cohen's *d*, Pearson's *r*), indicating how they were calculated |

*Our web collection on statistics for biologists contains articles on many of the points above.*

## Software and code

Policy information about availability of computer code

| Data collection | We analyzed existing datasets. Thus, no software was used to collect data. |
|---|---|
| Data analysis | STAR v2.7.8a (https://github.com/alexdobin/STAR) was used for RNA-seq reads alignment. RNA-SeQC v2.3.5 (https://github.com/getzlab/rnaseqc) was used for quantifying gene-level abundance. RSEM v1.3.1 (https://github.com/deweylab/RSEM) was used for quantifying isoform-level abundance. OSCA v0.45 (https://yanglab.westlake.edu.cn/software/osca; https://github.com/jianyangqt/osca) was used for the THISTLE sQTL analysis. LeafCutter v0.2.9 (https://davidaknowles.github.io/leafcutter/articles/sQTL.html) and fastQTL v2 .184 (https://fastqtl.sourceforge.net/) were used for the LeafCutter sQTL analysis. S-LDSC v1.0.1 (https://github.com/bulik/ldsc) was used for the heritability enrichment analysis. MESC v1 (https://github.com/douglasyao/mesc) was used for the mediation heritability analysis. SMR v1.03 (https://yanglab.westlake.edu.cn/software/smr/#Overview) was used for the SMR analysis. COLOC v5.1.1 (https://cran.r-project.org/web/packages/coloc/index.html) and eCAVIAR v2.2 (https://github.com/fhormoz/caviar) were used for the COLOC analysis. TORUS v1 (https://github.com/xqwen/torus) was used for the functional enrichment analysis. FOCUS v0.6.10 (https://github.com/bogdanlab/focus) was used for fine-mapping causal genes. |

For manuscripts utilizing custom algorithms or software that are central to the research but not yet described in published literature, software must be made available to editors and reviewers. We strongly encourage code deposition in a community repository (e.g. GitHub). See the Nature Research guidelines for submitting code & software for further information.

## Data

PsychENCODE data: https://www.synapse.org/#!Synapse:syn4921369/files/. AMP-AD data: https://www.synapse.org/#!Synapse:syn5550382. Online tool for querying the sQTL and eQTL summary statistics: https://yanglab.westlake.edu.cn/data/brainmeta. The full summary statistics from the sQTL, eQTL, SMR and COLOC analyses are available at https://yanglab.westlake.edu.cn/pub_data.html. GRCh37 genome: https://www.ncbi.nlm.nih.gov/genome/guide/human/. GENCODE: https://www.gencodegenes.org/human/release_37lift37.html.

# Field-specific reporting

Please select the one below that is the best fit for your research. If you are not sure, read the appropriate sections before making your selection.

☒ Life sciences        ☐ Behavioural & social sciences        ☐ Ecological, evolutionary & environmental sciences

For a reference copy of the document with all sections, see nature.com/documents/nr-reporting-summary-flat.pdf

# Life sciences study design

All studies must disclose on these points even when the disclosure is negative.

| | |
|---|---|
| Sample size | We used 2,443 unrelated individuals of European ancestry from the Psych ENCODE and AMP-AD consortia for real data analysis. The sample size was determined by the maximum number of unrelated individuals of European ancestry with both SNP genotype and RNA-seq data in the PsychENCODE and AMP-AD consortia. |
| Data exclusions | We excluded individuals of non-European ancestry, with < 10 million total reads, or with RNA integrity number < 5.5, and one of each pair of individuals with genetic relatedness > 0.05. We excluded genetic variants with minor allele frequencies < 0.01, Hardy- Weinberg Equilibrium test P-value < le-6, imputation INFO score < 0.3, or missingness rate > 0.05. |
| Replication | We replicated the simulation 500 times in each scenario to investigate the performance of THISTLE in comparison with existing methods. All replications were successfully performed. In real data analysis, we used all the available data to maximize power for discovery, so replication was not performed. |
| Randomization | We analyzed existing data sets. Thus, no randomization was performed. In eQTL and sQTL analysis, covariates were adjusted for to account for potential confounding. |
| Blinding | Group allocation was not relevant to this study, so blinding was not necessary. |

# Reporting for specific materials, systems and methods

We require information from authors about some types of materials, experimental systems and methods used in many studies. Here, indicate whether each material, system or method listed is relevant to your study. If you are not sure if a list item applies to your research, read the appropriate section before selecting a response.

### Materials & experimental systems

| n/a | Involved in the study |
|---|---|
| ☒ | ☐ Antibodies |
| ☒ | ☐ Eukaryotic cell lines |
| ☒ | ☐ Palaeontology and archaeology |
| ☒ | ☐ Animals and other organisms |
| ☐ | ☒ Human research participants |
| ☒ | ☐ Clinical data |
| ☒ | ☐ Dual use research of concern |

### Methods

| n/a | Involved in the study |
|---|---|
| ☒ | ☐ ChIP-seq |
| ☒ | ☐ Flow cytometry |
| ☒ | ☐ MRI-based neuroimaging |

## Human research participants

Policy information about studies involving human research participants

| | |
|---|---|
| Population characteristics | Our study involved publicly available data sets (e.g. PsychENCODE, AMDAD, and existing summary statistics). We restricted our data to individuals of European ancestry. |

| Recruitment | We analyzed existing data sets. Thus, no recruitment was performed. |
| --- | --- |
| Ethics oversight | The Ethics Committee of Westlake University (approval number: 20200722YJ001) and the University of Queensland Human Research Ethics Committee B (approval number: 2011001173). |

Note that full information on the approval of the study protocol must also be provided in the manuscript.

