## [Peer Review File · Nature Genetics]

Peer Review Information

Manuscript Title: Genetic control of RNA splicing and its distinctive role in complex trait variation

Corresponding author name(s): Professor Jian Yang

Reviewer Comments & Decisions:

Decision Letter, initial version:
--

18th Mar 2021

Dear Jian,

Your Article entitled "Genetic control of RNA splicing and its distinctive role in complex trait variation" has now been seen by 3 referees, whose comments are attached. In the light of their advice we have decided that we cannot offer to publish your manuscript in Nature Genetics.

While the referees find your work of some interest, they raise concerns about the strength of the novel conclusions that can be drawn at this stage.

In brief, while the referees found some points of interest in your manuscript, all three also made overlapping comments regarding the biological novelty of your PsychENCODE sQTL catalogue, and a range of methodological/technical aspects of THISTLE and the analysis performed.

While there is clear guidance provided on how to improve the issues raised, we believe that this would require a significant revision involving a major expansion of the study; and we concluded that, even if this was successfully completed, that it remained unclear whether the referees would then be supportive of acceptance.

We feel that these reservations and the likely outcome are sufficiently important as to preclude publication of this study in Nature Genetics.

I am sorry that we cannot be more positive on this occasion but hope that you will find our referees' comments helpful when preparing your paper for submission elsewhere.

Sincerely,

Michael Fletcher, PhD
Associate Editor, Nature Genetics

ORCID: 0000-0003-1589-7087

Referee expertise:

Referee #1: neurogenetics

Referee #2: computational biology, genomics

Referee #3: QTL analysis, bioinformatics

Reviewers' Comments:

Reviewer #1:

Remarks to the Author:

The authors have created a new method for mapping splicing quantitative trait loci (sQTL) which are associations of genetic variation with changes in mRNA splicing, an important molecular phenotype. Their method, THISTLE, takes as input per-isoform QTL associations and performs a test for heterogeneity of effect sizes across isoforms for each gene. They compare their method to a previously published multivariate isoform-level method (sQTLseeker) and the current state of the art event-based method (LeafCutter). They find that THISTLE clearly outperforms sQTLseeker. However, they show that Leafcutter, due to differences in how splicing is modelled and estimated, leads to a partially overlapping but complementary set of associations.

They go on to combine sQTLs from THISTLE and Leafcutter into a large set of associations on the current largest human brain dataset created by the PsychENCODE consortium. Analyzing these results, they demonstrate that with increasing sample sizes, the gap between eQTL and sQTL discovery in terms of associated genes (eGenes and sGenes respectively) begins to shrink. However, they also demonstrate that in the majority of cases where a gene has both an eQTL and an sQTL, that the two associations arise from independent genetic variants.

They then use their sQTL catalogue to explore associations with a set of brain-derived diseases and traits. They find sQTLs and eQTLs to be largely comparable in their enrichment for heritability (using MESG) and of similar usefulness for prioritizing candidate genes, using COLOC and SMR-HEIDI. They present some examples of disease-relevant genes associated with sQTLs and/or eQTLs. Comparing their associated genes found by eQTLs and sQTLs, they find a modest overlap. Taking those overlapping genes, they find that there are a proportion of genes that can be explained by a single causal set of variants that act through both molecular phenotypes.

Splicing QTLs (presumably using Leafcutter) have already been generated by the PsychENCODE

2consortium (Wang et al, 2018, Science) in the same dataset used in this study, a fact which doesn't appear to be acknowledged in this manuscript, unless I missed it. Furthermore, multiple additional studies have generated splicing QTLs in the human brain and performed various heritability enrichment analyses and/or colocalization methods on the same diseases and traits applied in this study. Takata, 2017, Nat Comm mapped sQTLs (vast-tools, event based) in brain and found associations with Schizophrenia and other GWAS traits; Raj et al., 2018, Nat Gen mapped sQTL in brain (Leafcutter) and associated with Alzheimer's Disease GWAS; Li et al, 2019, Nat Comms mapped sQTL in brain (Leafcutter) and associated with Parkinson's disease; Guelfi et al, 2020, Nat Comms, mapped a range of splicing and expression QTLs (event-based) and colocalized with Schizophrenia and Parkinson's disease GWAS. It would strengthen this manuscript considerably to compare the current results with what has already been published, and would help to distinguish exactly how novel the current findings are.

A pressing need in studies of the genetic basis of splicing is the identification of which particular isoforms are affected by a particular variant. THISTLE tests for heterogeneity of isoform-level QTLs, but it does not appear to inform the user of which isoforms are driving such heterogeneity. Is some kind of post-hoc analysis within the scope of this work? It would make the interpretation of results for users much more informative. In addition, there is a pressing need for visualization of splicing QTLs. The manuscript would be strengthened if the authors made an attempt to visualize the isoform QTL results at least for the disease associated genes they highlight, as was done by the sQTLseeker authors.

The authors suggest that Leafcutter outperforms THISTLE because it identifies splicing changes in isoforms not quantified in their samples. However, the authors do not mention which reference transcriptome set was used to quantify isoforms with RSEM. Is it possible to compare THISTLE runs using isoform abundances created from a set of more conservative (RefSeq, GENCODE basic) and more comprehensive references (GENCODE full, CHES)? Does this show an increasing power for THISTLE? The authors could even create a de novo human brain transcript reference using stringtie or cufflinks to press this point. The authors should explore at least the first avenue as it would strengthen the performance of THISTLE compared to Leafcutter.

In most QTL studies, multiple testing correction is performed first at the feature level (the gene) with some sort of permutation procedure, followed by a second round of correction for the number of features tested (FDR, Storey's q-value). Can the authors comment on whether such a procedure is warranted with THISTLE, and if so, why didn't they perform the standard two-stage procedure?

Leafcutter does not capture differences in coverage across features such as retained introns and alternate UTRs, which presumably account for a number of sQTLs found only by THISTLE. Some discussion of the relative merits/limitations of these methods is needed. The lead SNP analysis on Fig S20 shows enrichment of SNPs within genomic features, but an enrichment with putative effected transcript within different types of transcript features.

Minor comments:

Fig S3: no key for colours of heatmap – is blue high or low? Please amend. Ideally put the replication

3proportions as labels in each cell.

Fig S8: “pre-adjust known covariates; RINT” presumably this is a typo and you mean “RIN”.

Reviewer #2:

Remarks to the Author:

The authors consider splicing QTL mapping. They: 1) propose a heterogeneity test (“THISTLE”) based on iso-eQTL summary stats 2) map thistle and LeafCutter sQTLs in PsychENCODE (PEC) 3) perform standard downstream e/sQTL analyses: overlap, functional annotation enrichment, heritability and colocalization analysis. They find thistle outperforms sQTLseeker in terms of sensitivity and that thistle sQTLs, LeafCutter sQTLs, and eQTLs are relatively distinct. They show all three are valuable for interpreting brain traits, in particular neurodegeneration.

The paper is clearly written with only a small number of typos. I would rather the methods were self contained rather than requiring referring to the SMR/HEIDI paper for the details of the multivariate test.

Thistle itself appears a reasonable approach to sQTL mapping with potential advantages for meta-analysis due to requiring only summary stats (although many eQTL studies do not release these at the isoform level). There are disadvantages somewhat acknowledged by the authors: it doesn't tell us which specific isoform(s) are regulated, and relatedly, does not provide effect sizes which are useful for many downstream statgen analyses.

The claim that this work is the largest sQTL study to date is a bit misleading. Wang et al 2018 (Science) mapped transcription ratio sQTLs in PEC (the same data used here) and those sQTLs are publicly available through the PEC website. At 0.001 claimed FDR, PEC found sQTLs for 37259 transcripts/9832 genes, contrasted with the present paper's 4,881 (5,629) sGenes using Thistle (LeafCutter) (7,491 sGenes combined). Of course the interpretation of these numbers is dependent on the multiple hypothesis correction and thresholds used. Perhaps the more significant contribution of the current paper is the SMR and coloc analyses comparing the contribution of sQTLs and eQTLs to various neuro traits. However, the authors should also run these analyses using the Wang et al transcript ratio QTLs as a baseline.

I have some major technical concerns with the SMR and coloc analysis. First, it has been shown that TWAS type analyses are prone to false positives through various mechanisms (see e.g. Wainberg et al, NatGen 2019), including co-regulation. SMR is in practice a form of TWAS so will suffer from the same problems. There have been attempts to address this, in particular FOCUS (Mancuso et al, NatGen 2019) which explicitly accounts for co-regulation. I would like to see the FOCUS (or equivalent) analysis alongside the SMR (I don't think that's too much to ask given that imo the TWAS/coloc analyses are the main contribution of this paper).

Second, I consider the authors' interpretation of the COLOC results to be invalid. They consider only PP4, the posterior probability that the variant for both phenotypes is shared. When this is >0.8 they

4say there is sharing, and no sharing otherwise. However, COLOC compares between 5 hypotheses: no true effects (H0), true effect only for phenotype 1 (H1), true effect only for trait 2 (H2), true effect for both but distinct variants (H3) and shared variants (H4). The authors are implicitly assuming PP3=1-PP4 but there are the first 3 hypotheses/models to consider. They should instead consider both PP3 and PP4: the number of genes with e.g. PP3>0.8 is a more meaningful lower bound on the number of genes with distinct effects. Incomplete power and false positives will contribute probability mass to H0, H1 and H2.

Third, the enrichment analyses for the sQTLs are quite naive. Afaik only the "top" (presumably you mean lead?) SNP is used. Various approaches for enrichment such as eQTNminer and William Wen's TORUS/DAP-G line of work have been proposed for properly assessing enrichment accounting for fine-mapping uncertainty and LD. These would give more meaningful enrichments. A bit more exploration of e.g. are the sQTLs typically exonic, intronic, or intergenic, and how far into the exon/intron do they tend to do would also be informative.

Overall this is an interesting paper but given a) that sQTLs have previously been mapped in PEC and b) the TWAS/COLOC analyses have major problems, at least a major revision is needed to understand whether there is a meaningful contribution to the field here.

Reviewer #3:

Remarks to the Author:

In this work, Qi and colleagues leverage 6 datasets to generate the largest brain cis sQTL collection to date. For sQTL mapping, they propose a method named THISTLE, the basis of which was previously published in the context of GWAS, in addition to LeafCutter. They also map eQTLs using the same data. They analyze sQTLs and eQTLs, together with summary statistics from brain-related complex traits. They found eQTLs and sQTLs to explain a similar fraction of heritability for these traits, while the corresponding causal variants tend to be distinct, and point to different genes associated with the traits. Although I find this submission interesting, I have substantial concerns regarding the methodology and the depth of certain downstream analyses, which need to be resolved:

1. The comparison between THISTLE and sQTLseeker (lines 110-130 and Supp. Note) is fairly limited. First, authors use an old version of sQTLseeker (sQTLseeker2 is available 10.1038/s41467-020-20578-2). In addition, simulations i) are restricted to very few scenarios, which does not allow to make such general claims about statistical power (e.g. lines 21, 77-80, 115-117), and ii) require a much more detailed description. Comparison with a third method such as DRIMseq (doi: 10.12688/f1000research.8900.2) would be also valuable.

1.1. It is unclear how authors study power (e.g. Supp. Fig. 2). Which is the H1 scenario selected (i.e. 3 or 4)? How many SNPs are under H1?. Please describe in detail. Power should be evaluated modifying effect sizes, rather than the P-value threshold (Supp. Fig. 2c-d). Power should be studied for various isoform numbers (e.g. typical protein coding genes with multiple isoforms have 2 - 25 isoforms), especially given that the number of terms in the THISTLE quadratic form increases dramatically with the number of isoforms ($\#terms \sim (isoforms)^{2/2}$). It would be also required to

5evaluate in depth the performance of both methods as a function of the degree of correlation between the responses and the sample size. MANOVA, DRIMseq or sQTLseeker model explicitly the covariances between different traits, but authors need to show how well THISTLE approximates various real covariance structures, especially with many isoforms.

1.2. Authors should explain better multivariate Poisson data generation: e.g. how can this distribution have mean 0 (or are they centering)?, how is it possible that a linear regression fit to the marginal poisson values results in meaningful beta coefficients? or are they either fitting a glm or log/rank-inverse normal transforming? Please detail. In any case, authors should simulate under a model that takes into account overdispersion of RNA-seq data, such as the Dirichlet-multinomial. Indeed, it may be meaningful to add DRIMseq (doi: 10.12688/f1000research.8900.2) to the comparison of methods. It would be also important to evaluate how robust is THISTLE to stronger violations of multivariate normality (given that sQTLseeker is non-parametric), and which is the impact of lowly expressed isoforms. Authors should compare/discuss the use of relative (as in sQTLseeker) vs absolute (THISTLE, DRIMseq) isoform abundances.

1.3. All comparisons between THISTLE, sQTLseeker and LeafCutter (e.g. lines 117-118, Supp. Fig 3, etc.) should be done on the same set of tested gene-variant pairs and uniformizing the analyses. Otherwise, the numbers provided are very difficult to compare, as the differences could be due to distinct filtering and/or parameter selection. Note that besides filtering, e.g. sQTLseeker does not report sQTLs with small effects, nor variance-sQTLs, and LeafCutter is able to find associations with novel introns.

1.4. Besides the technical comparison of the methods, authors should compare the sQTLs obtained via different methods in terms of the splicing events that they capture, the genes that they affect, and the enrichment of the sQTLs in functional annotations.

2. THISTLE computes p-values using the CDF of a quadratic form in normal variables. Although, there are exact methods to estimate it (e.g. Davies method), THISTLE relies on an approximation (lines 479-480). This approximation has a relative error that can be large, particularly in the case of low p-values (authors' ref 63, <https://www.jstor.org/stable/2673596?seq=1>, where they evaluate p-values only down to $1e-4$), and authors report p-values down to $1e-300$ (e.g. Supp. Fig 9). Beyond the discussion of whether such small p-values are meaningful, authors should assess the validity of the Saddlepoint approximation for small p-values, comparing with the exact method.

3. Authors use the Bonferroni method for multiple testing correction. Although common practice in GWAS, it has several pitfalls and is far from the gold standard in QTL mapping (i. e. permutations to correct for multiple variants tested per gene, accounting for LD, and FDR to correct for multiple genes tested genome-wide, 10.1093/bioinformatics/btv722). If permutations are too computationally intense, an alternative could be approximations such as eigenMT (10.1016/j.ajhg.2015.11.021).

4. All along the text, authors talk about 'event-based' and 'transcript-based' methods for sQTL mapping (lines 58-59, 164, discussion, etc.). However, it is important to distinguish between methods to quantify splicing (indeed, 'event-based' such as LeafCutter or 'transcript-based' such as RSEM), and methods to map sQTLs (i.e. find statistical associations with the genotype, e.g FastQTL, sQTLseeker

6or THISTLE). For instance, LeafCutter is not a method for sQTL mapping (line 164). LeafCutter intron-excision ratios can be used as input for sQTL mapping methods such as linear regression in FastQTL, but also as input for sQTLseeker2 or THISTLE itself. Please clarify in the text.

5. Authors employ a meta-analysis across isoform-level eQTLs to map expression QTLs (according to Methods lines 528 - 530). Instead, they should use a standard method for eQTL mapping (e.g. FastQTL, 10.1093/bioinformatics/btv722) to model gene expression values, provided that they are available.

6. Regarding the functional enrichment of sQTLs/eQTLs, authors perform a very limited study, restricting their claims to the fact that sQTLs are more enriched in splice sites than eQTLs, and eQTLs more enriched in the vicinity of TSSs (lines 240-245), something expected and pointed out often in the literature (10.1126/science.aad9417, 10.1126/science.aaz1776, 10.1038/s41467-020-20578-2). However they do not describe at all the results of the enrichments in the functional annotation by REMC, which seem to be very similar between sQTLs and eQTLs (Fig 3c, d), nor explore important aspects that would be relevant to distinguish between eQTLs/sQTLs: enrichment in RBPs, chromatin marks (e.g. H3k27ac/H3k36me3), etc, which may provide insight into the differential mechanisms of sQTLs and eQTLs. They also restrict the location analysis to the distance to the TSS, while the distribution of eQTLs and eQTLs along gene bodies of upstream/downstream regions is known to be important (10.1126/science.aad9417,10.1038/ncomms5698). Overall, further analyses are required to make this section compelling.

7. According to the authors (line 293), THISTLE does not provide an estimate of the effects. If so, which is the input that they provide to colocalization/fine mapping methods? If they provide P-values generated by the Saddlepoint approximation, these do not come from a normal distribution, which breaks a key assumption of most colocalization/fine mapping approaches, including COLOC and CAVIAR.

8. Authors claim that 32% of sGenes share causal variants with the eGenes. First, authors should only make statements on the set of genes that they can explicitly test for colocalization (that is, genes that are both sGenes and eGenes). Hence, this number is indeed 50% (2418/4893), lower than the $100 - 32 = 68\%$ reported. Moreover, that x% share a causal variant does not imply that 100-x% do not share it (if I understand correctly, that COLOC is not able to define whether they are the same signal - PP4 is low-, does not imply that they are different signals -PP3 is high-, what about the probabilities for the other hypotheses H0-H2?). To provide a better estimate of the fraction of shared/independent signals, authors should use a combination of different colocalization methods, including approaches that (in contrast to COLOC) are not restricted to only 1 causal variant per locus.

9. In their heritability analyses, authors show that i) sQTLs and eQTLs contribute at a similar degree to the selected traits and ii) lead to the identification of different genes via MR. Although i) has been shown before to some extent (10.1126/science.aad9417, 10.1038/s41467-020-20578-2), i believe that ii) is a novel and interesting result. However, this claim is based on very few genes (e.g. $68\% \sim 73/107$) and it is unclear how generalizable it is. In addition, when they explicitly test for sharing causal variants of the genes identified in common (34), they find that the vast majority (25) are due to the same causal variants, which contrasts with their previous claims.

7Minor points:

- As a general comment, I find that the methods section (and the supplementary note) should be more detailed.
- Summary statistics for GWAS traits and gene expression are routinely generated. However it is unclear whether this is also the case of isoform-level summary statistics, and therefore, how useful is this feature of THISTLE (lines 104, 371), which is presented as one of its advantages. Please justify.
- Regarding covariate correction across QTL analyses (lines 523, 539), if I understand correctly, variance partition is applied, and then the effect of known covariates is removed from original responses via an LMM. Later, residuals are regressed on genetic PCs and PEER factors. Is this the case? Are PEER factors learnt from these residuals? Why this procedure, instead of learning PEERs from the original matrix, providing known covariates as an argument (10.1038/nprot.2011.457)? How was the number of PEERs selected? Why different numbers for THISTLE and LeafCutter analyses? Please give more details and justify.
- Intron excision ratios are standardized and quantile-normalized (line 539). According to methods, Transcript expressions (TPMs) do not undergo any transformation. However in Supp. Fig. 8 they appear to be rank-inverse normalized. Please clarify. If so, how is this normalization affecting the correlation between isoforms? Authors should study this transformation in their simulations.
- Fig 3. (Supp. Fig 18) Enrichment in functional annotations should be done using a null distribution of matched variants regarding both MAF and distance to TSS (or better, their relative location: introns, exons, downstream, upstream + the actual distance to TSS/TTS). There is no need to do these two analyses separately.
- Fig 4a should be supplementary and leave just Fig 4b, which corrects for the ascertainment bias. In addition, authors should use as null a set of SNPs matched with respect to MAF, but also with respect to their location (see above) both here and in Supp. Figs 14a, 15a, 16a. I wonder also whether the main figure should use the fine mapped (instead of the top 1 QTLs). Please uniformize the y-axis label and the notation regarding heritability fractions in the figures and along the text (e.g. $\Pr(h^2_1)$ vs h^2_1).
- Figs 3, 4, and supp 12, 14, 15, 16, 20. Besides the ratios, please show the actual number of overlaps/the actual heritabilities in the plots.
- Lines 71-72. Please clarify what you mean by 'subtle splicing patterns'.
- Why including so small datasets (i.e. BipSeq $n = 27$, UCLA-ASDn = 64) in the analysis? This barely increases power while may reduce the number of variants analyzed. Please justify.
- Lines 125-126. Default p-values in sQTLseeker are not computed using permutations, but rather an asymptotic approximation. In sQTLseeker2, p-values are down to $1e-14$ via the Davies method (10.1038/s41467-020-20578-2).
- Line 154. Authors explore gene bodies +/- 2Mb. Why such a large window? How do you expect a variant e.g. 2Mb downstream to affect splicing? Please justify.
- Line 168. Please clarify in Methods how LeafCutter clusters are mapped to genes. Specify also which genes you focus on (e.g. only protein coding/lncRNAs/all).
- Line 195. Which fraction of sQTLs identified by LeafCutter corresponds to novel introns/isoforms? Please specify.
- Line 199. The R shiny app does not work, it does not return any result after entering data in the required fields (I just tried Linux+Firefox).

- Line 256-259. Please specify the actual traits studied in the non-case/control GWASs in Methods and in Supp. Table 1.
- Lines 278-279. Please define 'clumped SNPs' and give details on how clumping was done in Methods.
- Lines 283-284 and 585-588. Please state explicitly the difference between the meaning of tau, and how it differs from per-SNP heritability to help the reader. In addition, (fig 4d) discuss why tau is larger for sQTLs, while the fraction of heritability is larger for eQTLs.
- Lines 308-329. Some numbers seem not to coincide with the ones provided in Supp table 2. Please specify in the legend what the parentheses means in the LeafCutter/No. of coloc genes column.
- Line 315. As far as I am concerned, LeafCutter is not able to model intron retention events (<https://www.biorxiv.org/content/10.1101/463927v1.full.pdf>). Please clarify.
- Line 574. Why are non-top-1 sQTLs annotated the same as non-sQTLs (i.e. 0)? Why restricting to top 1? Please justify.
- Figs. 2a, 4e and Supp. 11a, 21. Please plot Venn diagrams instead. For two groups a simple Venn diagram is much more intuitive and easy to understand.
- Supp. Fig 11b. Please add violin plots as in 2b.
- Supp. Fig 13. Authors should show the actual QQplots too.
- Supp. Note. Please specify exactly how you simulate SNPs using the binomial model.
- Supp. Note. In the MVN scenario, how do authors restrict that isoform expression to be > 0 ? Please clarify.
- Please provide a supplementary "summary" table with the number of eQTLs and sQTLs identified using the different approaches.

Author Rebuttal to Initial comments

Responses to the Reviewers

We thank the reviewers for their constructive comments which have helped to improve our manuscript. We have addressed all the reviewers' comments point-by-point below (in blue) in this document and made the corresponding changes in the manuscript files (highlighted in yellow). Here is a summary of the main changes.

- 1) We have increased the sQTL study sample size from 1,073 to 2,865 and re-run all the analyses with the new data. While the number of discovered sQTLs and eQTLs and the number of GWAS signals shared with sQTL or eQTL signals increased owing to the larger sample size, our main conclusions that sQTLs and eQTLs are largely distinct and that sQTLs play a critical role in mediating complex trait genetic effects remained unchanged.
- 2) We have expanded the sQTL/eQTL analysis from cis-region only to genome-wide, aiming to identify trans-sQTLs/eQTLs, especially those colocalized with the GWAS signals. This expansion has led to the discovery of several trans-sQTLs/eQTL signals, some of which were shared with the GWAS signals.
- 3) We have compared THISTLE with more transcript-based sQTL mapping methods including the new version of sQTLseeker, under a broader spectrum of simulation settings and shown that THISTLE was still more powerful than the existing transcript-based methods when evaluated at the same level of false-positive rate. Nevertheless, we want to clarify that the primary goal of this study is not to develop a new sQTL method but to demonstrate, by using a combination of powerful sQTL mapping methods and large sample size, the importance of considering sQTLs to reveal biological mechanisms underlying the GWAS signals.
- 4) We have significantly improved the online tool to visualize and download the sQTLs/eQTL summary data using PheWeb (<https://yanglab.westlake.edu.cn/resources/qtl/>).
- 5) We have revised the text thoroughly to improve the readability of the manuscript.

Reviewer #1:

Remarks to the Author:

The authors have created a new method for mapping splicing quantitative trait loci (sQTL) which are associations of genetic variation with changes in mRNA splicing, an important molecular phenotype. Their method, THISTLE, takes as input per-isoform QTL associations and performs a test for heterogeneity of effect sizes across isoforms for each gene. They compare their method to a previously published multivariate isoform-level method (sQTLseeker) and the current state of the art event-based method (LeafCutter). They find that THISTLE clearly outperforms sQTLseeker. However, they show that Leafcutter, due to differences in how splicing is modelled and estimated, leads to a partially overlapping but complementary set of associations.

They go on to combine sQTLs from THISTLE and Leafcutter into a large set of associations on the current largest human brain dataset created by the PsychENCODE consortium. Analyzing these results, they demonstrate that with increasing sample sizes, the gap between eQTL and sQTL discovery in terms of associated genes (eGenes and sGenes respectively) begins to shrink.

However, they also demonstrate that in the majority of cases where a gene has both an eQTL and an sQTL, that the two associations arise from independent genetic variants.

They then use their sQTL catalogue to explore associations with a set of brain-derived diseases and traits. They find sQTLs and eQTLs to be largely comparable in their enrichment for heritability (using MESZ) and of similar usefulness for prioritizing candidate genes, using COLOC and SMR-HEIDI. They present some examples of disease-relevant genes associated with sQTLs and/or eQTLs. Comparing their associated genes found by eQTLs and sQTLs, they find a modest overlap. Taking those overlapping genes, they find that there are a proportion of genes that can be explained by a single causal set of variants that act through both molecular phenotypes.

Re: We thank the reviewer for the summary of our study.

1. Splicing QTLs (presumably using Leafcutter) have already been generated by the PsychENCODE consortium (Wang et al, 2018, Science) in the same dataset used in this study, a fact which doesn't appear to be acknowledged in this manuscript, unless I missed it. Furthermore, multiple additional studies have generated splicing QTLs in the human brain and performed various heritability enrichment analyses and/or colocalization methods on the same diseases and traits applied in this study. Takata, 2017, Nat Comm mapped sQTLs (vast-tools, event based) in brain and found associations with Schizophrenia and other GWAS traits; Raj et al., 2018, Nat Gen mapped sQTL in brain (Leafcutter) and associated with Alzheimer's Disease GWAS; Li et al, 2019, Nat Comms mapped sQTL in brain (Leafcutter) and associated with Parkinson's disease; Guelfi et al, 2020, Nat Comms, mapped a range of splicing and expression QTLs (event-based) and colocalized with Schizophrenia and Parkinson's disease GWAS. It would strengthen this manuscript considerably to compare the current results with what has already been published, and would help to distinguish exactly how novel the current findings are.

Re: We thank the reviewer for this comment, which have helped us to improve the manuscript considerably. During the revision process, we have increased our study sample size from 1,073 to 2,865 by adding RNA-seq and genotype data from three cohorts, Mayo, MSBB, and ROSMAP, from the AMP-AD consortium (**Supplementary Table 1**).

We have summarised the comparison of the sQTLs mapped in this study with those from previous studies in **Supplementary Table 3**. The differences in the numbers of sGenes and unique sQTL SNPs (without clumping) detected between this study and the previous studies are remarkable, reflecting the differences in study sample size (**Supplementary Table 3**). Among all the previous sQTL studies, the Wang et al. study has the largest sample size, which, however, is still more than two times smaller than our current sample size.

Apart from the difference in sample size, there is a difference in methodology (**Supplementary Table 3**). We have shown that THISTLE improved power over the state-of-the-art transcript-

based sQTL method, sQTLseeker, and is largely complementary to the state-of-the-art event-based sQTL method, LeafCutter. Hence, a combination of THISTLE and LeafCutter maximizes the sQTL detection power, in contrast to all the previous studies, including the Wang et al. study, in which only a single sQTL method was used.

There are two versions of the Wang et al. sQTL summary statistics available, with one version excluding transcripts whose expression levels were < 5 FPKM in all the individuals and the other version without such filtering. In the present study, following the GTEx analysis pipeline, we excluded transcripts whose expression levels were < 0.1 TPM in more than 80% of the individuals. So, there is a difference in QC filter. Under the same criterion without filtering transcripts, we identified 13,716 sGenes (with $P_{\text{sQTL}} < 5e-08$) with 1,875,864 unique sQTL SNPs (without clumping), compared to 7,296 sGenes and 462,722 unique sQTL SNPs in the Wang et al. study, 1.7- and 4-fold increase in the numbers of sGenes and sQTL SNPs, respectively (**Supplementary Fig. 14** and **Supplementary Table 3**). Linking the sQTLs to the GWAS signals for the 12 complex traits, the number of genes passing the SMR and COLOC PP4 thresholds was 255 using our data and 146 using the Wang et al. data, a 1.7-fold increase (**Supplementary Fig. 33**).

While previous studies have focused on identifying genes for which the sQTL signals are colocalized with the GWAS signals for a few specific diseases, our study was aimed at quantifying the role of sQTLs, in comparison to eQTLs, in mediating genetic effects for a host of complex traits using a powerful dataset.

One more difference is that we have expanded the sQTL analysis from cis-region only to genome-wide, resulting in the discovery of 225 trans-sGenes at 5% FDR (corresponding to $P_{\text{sQTL}} < 2.7e-11$; **Fig. 6**).

We have incorporated the discussion above in the revised manuscript (lines 147-148, 151-152, 199-201 and 507-511; **Supplementary Figs. 14** and **33**; **Supplementary Tables 1** and **3**; **Fig. 6**; section 5 of the **Supplementary Note**).

2. A pressing need in studies of the genetic basis of splicing is the identification of which particular isoforms are affected by a particular variant. THISTLE tests for heterogeneity of isoform-level QTLs, but it does not appear to inform the user of which isoforms are driving such heterogeneity. Is some kind of post-hoc analysis within the scope of this work? It would make the interpretation of results for users much more informative. In addition, there is a pressing need for visualization of splicing QTLs. The manuscript would be strengthened if the authors made an attempt to visualize the isoform QTL results at least for the disease associated genes they highlight, as was done by the sQTLseeker authors.

Re: We thank the reviewer for these suggestions. First, we have modified the THISTLE software to report the isoform-eQTL effects for each significant sQTL, which are helpful to identify the lead

isoforms, if there is, that drive the heterogeneity, and for post-hoc analyses that require isoform-eQTL summary statistics. Second, we have provided an R script at the THISTLE website (<https://yanglab.westlake.edu.cn/software/osca/#THISTLE>) to visualize the isoform-eQTL effects, as in **Fig. 5c** or **Supplementary Fig. 29c**.

3. The authors suggest that Leafcutter outperforms THISTLE because it identifies splicing changes in isoforms not quantified in their samples. However, the authors do not mention which reference transcriptome set was used to quantify isoforms with RSEM. Is it possible to compare THISTLE runs using isoform abundances created from a set of more conservative (RefSeq, GENCODE basic) and more comprehensive references (GENCODE full, CHES)? Does this show an increasing power for THISTLE? The authors could even create a de novo human brain transcript reference using stringtie or cufflinks to press this point. The authors should explore at least the first avenue as it would strengthen the performance of THISTLE compared to Leafcutter.

Re: We thank the reviewer for this helpful suggestion. In the original manuscript, the isoforms were annotated based on GENCODE v19 by the PsychENCODE Consortium. In the revised manuscript, to harmonize RNA-seq data from multiple cohorts, we annotated the isoforms based on GENCODE v37.

As per the reviewer's suggestion, we have compared the THISTLE results using four transcriptome reference sets, i.e., RefSeq, GENCODE v37 Basic, GENCODE v37, and de novo assembly (constructed from the RNA-seq data using StringTie based on GENCODE v37). For ease of computing, we limited the comparison in the MSBB cohort ($n = 183$) due to its small sample size. THISTLE with GENCODE identified 1,243 sGenes (with $P_{sQTL} < 5e-08$), compared to 681 with GENCODE Basic and 380 with RefSeq (**Supplementary Fig. 13**), reflecting the differences in the number of annotated isoforms between the three references (142,275 in GENCODE, 71,175 in GENCODE Basic, and 45,590 in RefSeq). The analysis with the de novo assembly identified 199 more sGenes than GENCODE (~16% increase). However, the proportion of additional THISTLE sGenes decreased to ~4% when applying the de novo assembly to the whole dataset ($n = 2,865$) largely because of the discrepancy in the de novo assembled isoforms among cohorts. Considering further a likely higher level of uncertainty in the de novo assembled isoforms than those annotated in GENCODE, we opted to use the results based on GENCODE as the main source of data for subsequent analyses.

One more thing worth mentioning is that we have updated the LeafCutter analysis with an additional flag "--wasp" in the alignment analysis to filter out allele-specific reads mapped with a bias. With all the analyses updated, the number of sGenes (with $P_{sQTL} < 5e-08$) identified from THISTLE (9,305) was larger than that from the LeafCutter-based sQTL analysis (8,602) in the whole dataset.

We have updated the manuscript accordingly (lines 171-177, 516-518, and 562-564; **Supplementary Fig. 13**).

4. In most QTL studies, multiple testing correction is performed first at the feature level (the gene) with some sort of permutation procedure, followed by a second round of correction for the number of features tested (FDR, Storey's q-value). Can the authors comment on whether such a procedure is warranted with THISTLE, and if so, why didn't they perform the standard two-stage procedure?

Re: We thank the reviewer for this suggestion. In the previous manuscript, we used a p-value threshold of $5e-8$ for two reasons. First, this threshold is recommended in the SMR software tool, one of the main approaches we've used to link the sQTL (or eQTL) signals to GWAS signals. Second, we focused our analysis only on the cis-regions so that a threshold of $5e-8$ is likely to be sufficient, considering the p-value thresholds calibrated from permutations in prior eQTL studies, e.g., an FDR threshold of 0.05 corresponds to a p-value threshold of $2.02e-05$ in eQTLGen (Vosa et al. 2021 Nat Genet).

In the revised manuscript, we have computed the p-value threshold corresponding to an overall FDR of 0.05 by a permutation-based procedure for both the cis-sQTLs and cis-eQTLs. For the eQTLs, we shuffled the gene-level transcriptional abundance phenotype across individuals and re-ran the eQTL analysis with QTLtools. For the LeafCutter & QTLtools sQTLs, we shuffled the splicing phenotype (using the --grp-best option in QTLtools) across individuals and re-ran the sQTL analysis with QTLtools. For the THISTLE sQTLs, we permuted the isoform-level transcriptional abundance phenotype across individuals and re-ran the isoform-eQTL mapping and the THISTLE sQTL analysis using OSCA. In each of the three analyses above, we ran 1,000 permutations and fitted a beta distribution to the minimum p-values obtained from the permutations to compute the effective number of independent variants for each gene. We adjusted the original sQTL or eQTL p-values by this effective number of independent variants and employed the Storey's q-value approach to the lead sQTL or eQTL SNPs to account for multiple testing across genes. This procedure generated a p-value threshold of $3.4e-05$ for cis-eQTLs, $3.3e-05$ for THISTLE cis-sQTLs, and $5.9e-07$ for LeafCutter & QTLtools cis-sQTLs at an overall FDR of 0.05. All the thresholds were less stringent than $5e-8$ used in our earlier manuscript.

We have updated the manuscript accordingly (lines 167-168 and 192-194; section 4 of the **Supplementary Note**).

5. Leafcutter does not capture differences in coverage across features such as retained introns and alternate UTRs, which presumably account for a number of sQTLs found only by THISTLE. Some discussion of the relative merits/limitations of these methods is needed. The lead SNP analysis on Fig S20 shows enrichment of SNPs within genomic features, but an enrichment with putative effected transcript within different types of transcript features.

Re: We thank the reviewer for these suggestions. First, we have discussed the advantages and disadvantages of the two methods, including the point made by the reviewer, in the revised manuscript (lines 64-68 and 375-388). Second, we have performed a functional enrichment analysis of the splicing events associated with the lead THISTLE sQTL SNPs. The result shows that most sGenes identified by THISTLE undergo complex splicing events and that THISTLE can identify intron retention and alternative UTR. We have added these results in the revised manuscript (**Supplementary Fig. 6**; lines 381-383). We could not characterize the splicing events associated with LeafCutter sQTLs because LeafCutter does not report the flanking exons of the excised introns.

Minor comments:

Fig S3: no key for colours of heatmap – is blue high or low? Please amend. Ideally put the replication proportions as labels in each cell.

Re: We have updated the figure in the revised manuscript (now **Supplementary Fig. 5**).

Fig S8: “pre-adjust known covariates; RINT” presumably this is a typo and you mean “RIN”.

Re: RINT means rank-based inverse normal transformation (RINT). We have clarified it in the revised manuscript (line 556; now **Supplementary Fig. 10**).

Reviewer #2:

Remarks to the Author:

The authors consider splicing QTL mapping. They: 1) propose a heterogeneity test ("THISTLE") based on iso-eQTL summary stats 2) map thistle and LeafCutter sQTLs in PsychENCODE (PEC) 3) perform standard downstream e/sQTL analyses: overlap, functional annotation enrichment, heritability and colocalization analysis. They find thistle outperforms sQTLseeker in terms of sensitivity and that thistle sQTLs, LeafCutter sQTLs, and eQTLs are relatively distinct. They show all three are valuable for interpreting brain traits, in particular neurodegeneration.

1. The paper is clearly written with only a small number of typos. I would rather the methods were self contained rather than requiring referring to the SMR/HEIDI paper for the details of the multivariate test.

Re: We have included more details of the multivariate test in the Methods section (lines 497-499).

2. Thistle itself appears a reasonable approach to sQTL mapping with potential advantages for meta-analysis due to requiring only summary stats (although many eQTL studies do not release these at the isoform level). There are disadvantages somewhat acknowledged by the authors: it doesn't tell us which specific isoform(s) are regulated, and relatedly, does not provide effect sizes which are useful for many downstream statgen analyses.

Re: We thank the reviewer for this comment. As in response to comment #2 from Reviewer #1 above, this limitation can, to some extent, be alleviated by reporting the isoform-eQTL effects for the significant sQTLs, which are helpful for downstream analyses and to identify the lead isoforms, if there is, that drive the heterogeneity in the isoform-eQTL effects. We have modified the THISTLE software to do so and have provided an R script at the THISTLE website (<https://yanglab.westlake.edu.cn/software/osca/#THISTLE>) to make a forest plot of the isoform-eQTL effects, as in **Fig. 5c** or **Supplementary Fig. 29c**.

3. The claim that this work is the largest sQTL study to date is a bit misleading. Wang et al 2018 (Science) mapped transcription ratio sQTLs in PEC (the same data used here) and those sQTLs are publicly available through the PEC website. At 0.001 claimed FDR, PEC found sQTLs for 37259 transcripts/9832 genes, contrasted with the present paper's 4,881 (5,629) sGenes using Thistle (Leacutter) (7,491 sGenes combined). Of course the interpretation of these numbers is dependent on the multiple hypothesis correction and thresholds used. Perhaps the more significant contribution of the current paper is the SMR and coloc analyses comparing the contribution of sQTLs and eQTLs to various neuro traits. However, the authors should also run these analyses using the Wang et al transcript ratio QTLs as a baseline.

Re: This comment relates to comment #1 from Reviewer #1. During the revision process, we have increased our study sample size from 1,073 to 2,865 by adding RNA-seq and genotype data from three cohorts, Mayo, MSBB, and ROSMAP, from the AMP-AD consortium (**Supplementary Table 1**). The current sample size is the largest among all the published sQTL studies in any tissue.

The number of sGenes or sQTLs identified depends on sample size, quality control (QC) filters, methods used, and p-value threshold applied. We have shown that THISTLE improved power over the state-of-the-art transcript-based sQTL method, sQTLseeker, and is largely complementary to the state-of-the-art event-based sQTL method, LeafCutter. Hence, a combination of THISTLE and LeafCutter would maximize the sQTL detection power. We have now summarized the comparison of the sQTLs identified in this study with those from the previous studies in **Supplementary Table 3**. Regarding the comparison with the Wang et al. study, there are two versions of the Wang et al. sQTL summary statistics available, with one version excluding transcripts whose expression levels were < 5 FPKM in all the individuals and the other version without such filtering. In the present study, following the GTEx analysis pipeline, we excluded transcripts whose expression levels were < 0.1 TPM in more than 80% of the individuals. So, there is a difference in QC filter. Under the same criterion without filtering transcripts, we identified 13,716 sGenes (with $P_{\text{sQTL}} < 5e-08$) with 1,875,864 unique sQTL SNPs (without clumping), compared to 7,296 sGenes and 462,722 unique sQTL SNPs in the Wang et al. study, 1.7- and 4-fold increase in the numbers of sGenes and sQTL SNPs, respectively (**Supplementary Fig. 14** and **Supplementary Table 3**). Linking the sQTLs to the GWAS signals for the 12 complex traits, the number of genes passing the SMR and COLOC PP4 thresholds (see below for the response to the comment regarding the use of

a hybrid approach that combines the SMR test and the COLOC PP4) was 255 using our data and 146 using the Wang et al. data, a 1.7-fold increase (**Supplementary Fig. 33**).

We have updated the manuscript accordingly (lines 147-148, 151-152, 199-201 and 507-511; **Supplementary Figs. 14** and **33**; **Supplementary Tables 1** and **3**; section 5 of the **Supplementary Note**).

4. I have some major technical concerns with the SMR and coloc analysis.

First, it has been shown that TWAS type analyses are prone to false positives through various mechanisms (see e.g. Wainberg et al, NatGen 2019), including co-regulation. SMR is in practice a form of TWAS so will suffer from the same problems. There have been attempts to address this, in particular FOCUS (Mancuso et al, NatGen 2019) which explicitly accounts for co-regulation. I would like to see the FOCUS (or equivalent) analysis alongside the SMR (I don't think that's too much to ask given that imo the TWAS/coloc analyses are the main contribution of this paper).

Re: The reviewer is correct that co-regulation is one of the major issues for the TWAS type of analyses, including SMR. This is the reason why we followed all the SMR results by the COLOC PP4 test. In a typical SMR analysis, the SMR test is followed by the HEIDI test that utilizes LD pattern in the focal region to distinguish whether a molecular QTL (xQTL) signal is colocalised with a GWAS signal because of shared causal variant(s) (i.e., pleiotropic or causal model) or LD between distinct xQTL and GWAS causal variant(s) (i.e., co-regulation as defined in Wainberg et al. 2019 Nat Genet, or linkage model as defined in Zhu et al. 2016 Nat Genet). The HEIDI test has been proved effective in rejecting SMR associations due to co-regulation (Zhu et al. 2016 Nat Genet; Wu et al. 2018 Nat Commun). However, the HEIDI test requires the effects of a sufficient number of xQTL SNPs in the focal region, which are not available for the THISTLE sQTLs. As pointed out by the reviewer below, COLOC PP4 is a posterior probability to support the hypothesis that the GWAS signal is colocalised with the xQTL signal due to shared causal variant(s), which is conceptually similar to the HEIDI test (see **Supplementary Fig. 26** for a contrast of COLOC PP4 between the SMR associations accepted and rejected by the HEIDI test). Because the COLOC analysis requires only GWAS and xQTL p-values, we used COLOC PP4 as a replacement of HEIDI, to reject SMR associations owing to co-regulation in the analysis to link the sQTLs (or eQTLs) to GWAS signals. We have clarified this in the revised manuscript (lines 305-307 and 309-313; **Supplementary Fig. 26**).

As per the reviewer's suggestion, we have also performed an FOCUS analysis for the 12 brain-related traits using the eQTL and LeafCutter & QTLtools sQTL data. FOCUS prioritized 298 trait-associated sGenes and 567 trait-associated eGenes (**Supplementary Table 6**), both of which were larger than the corresponding number of genes identified by SMR & COLOC PP4 using the same data (114 trait-associated sGenes and 226 trait-associated eGenes), likely because FOCUS could not fully resolve the large number of genes prioritized by FUSION due to co-regulation (**Supplementary Table 6**). Of the 298 trait-associated sGenes, ~79.8% of which were not

identified through eQTLs (**Supplementary Fig. 27**), in line with our conclusions that sQTLs are largely independent of eQTLs and play a comparable role as eQTLs in mediating complex trait genetic effects. We have updated the manuscript accordingly (lines 328-330; **Supplementary Fig. 27; Supplementary Table 6**).

Second, I consider the authors' interpretation of the COLOC results to be invalid. They consider only PP4, the posterior probability that the variant for both phenotypes is shared. When this is >0.8 they say there is sharing, and no sharing otherwise. However, COLOC compares between 5 hypotheses: no true effects (H0), true effect only for phenotype 1 (H1), true effect only for trait 2 (H2), true effect for both but distinct variants (H3) and shared variants (H4). The authors are implicitly assuming $PP3=1-PP4$ but there are the first 3 hypotheses/models to consider. They should instead consider both PP3 and PP4: the number of genes with e.g. $PP3>0.8$ is a more meaningful lower bound on the number of genes with distinct effects. Incomplete power and false positives will contribute probability mass to H0, H1 and H2.

Re: We thank the reviewer for this comment, which is related to comment #8 from Reviewer #3 below. First, we acknowledge that we understand the five hypotheses of COLOC and the corresponding posterior probabilities. In the sGene-eGene colocalization analysis, we focused only on genes with both sQTL and eQTL signals. Consequently, the posterior probabilities of H0, H1, and H2 were negligible (median $PP0 = 3.0e-40$, $PP1 = 3.0e-100$, and $PP2 = 3.0e-125$). Nevertheless, we agree with the reviewer that it is more intuitive and direct to use PP3 to quantify the proportion of sQTLs distinct from eQTLs and have done so with the updated sQTL and eQTL data in the revised manuscript. There were 9,035 genes with both cis-sQTLs and cis-eQTLs, among which 4,377 genes had $PP3 > 0.8$ (**Fig. 2b**). Together with 3,405 unique sGenes without eQTLs, $\sim 61\%$ of the sGenes, $(4,377 + 3,405)/12,794$, showed sQTL signals distinct from eQTL signals, a proportion not too dissimilar to that quantified in our previous manuscript (i.e., 68%). We have updated the manuscript accordingly (lines 204, 225-226, and 342-343; **Fig. 2b**).

Third, the enrichment analyses for the sQTLs are quite naive. Afaik only the "top" (presumably you mean lead?) SNP is used. Various approaches for enrichment such as eQTNminer and William Wen's TORUS/DAP-G line of work have been proposed for properly assessing enrichment accounting for fine-mapping uncertainty and LD. These would give more meaningful enrichments. A bit more exploration of e.g. are the sQTLs typically exonic, intronic, or intergenic, and how far into the exon/intron do they tend to do would also be informative.

Re: Firstly, we have shown by several sensitivity analyses that the functional or heritability enrichment results remained essentially unchanged regardless of whether considering only the lead cis-sQTL/eQTL SNP or more associated cis-SNPs with or without accounting for LD. For example, we have performed the functional enrichment analysis with TORUS using the full cis-sQTL/eQTL summary statistics without SNP selection and observed similar functional enrichment patterns (**Supplementary Fig. 19**) as those based only on the lead sQTL/eQTL SNPs

(Fig. 3). We have also performed the heritability enrichment analysis with S-LDSC using all the significant, clumped, or fine-mapped cis-sQTL/eQTL SNPs. The results consistently showed comparable levels of heritability enrichment between the sQTLs and eQTLs (**Supplementary Figs. 23b, 24b and 25b**).

Additionally, we have expanded our functional enrichment analysis by including more annotation data such as 113 RNA-binding proteins (RBPs) binding sites and 7 histone marks. Note that the fold enrichment was computed as the proportion of sQTLs/eQTLs in a functional category divided by the mean of a null distribution generated by resampling “control” SNPs with MAF and distance to transcription start site (TSS) matched to the SNPs in query. The results showed that the sQTLs were more enriched in splicing regions (e.g., splice acceptors and splice donors) and RBPs binding sites than the eQTLs (**Fig. 3b & d**). The eQTLs were more enriched in TSS than the sQTLs (e.g., active TSS and bivalent TSS; **Fig. 3a**), consistent with our observation that the eQTLs were located much closer to TSS than the sQTLs (**Fig. 3e**). Both the sQTLs and eQTLs tended to be enriched at both ends of gene bodies, exons, and introns (**Supplementary Fig. 17**) and were significantly enriched in histone marks H3K4me3, H3K9ac, H3K27ac, H3K36me3, and H3K4me1 (**Fig. 3c**) but deflated in non-coding regions (**Fig. 3b**). Compared to that of the eQTLs, the enrichment of the sQTLs was higher in H3K36me3 but lower in H3K4me3, H3K9ac, or H3K27ac (**Fig. 3c**). We have added these results in the revised manuscript (lines 237-238, 243, 245-253 and 283-286; **Fig. 3; Supplementary Figs. 17, 23b, 24b and 25b**).

Overall this is an interesting paper but given a) that sQTLs have previously been mapped in PEC and b) the TWAS/COLOC analyses have major problems, at least a major revision is needed to understand whether there is a meaningful contribution to the field here.

Re: We thank the reviewer for all the constructive comments. We have addressed the two major concerns in the responses above.

Reviewer #3:

Remarks to the Author:

In this work, Qi and colleagues leverage 6 datasets to generate the largest brain cis sQTL collection to date. For sQTL mapping, they propose a method named THISTLE, the basis of which was previously published in the context of GWAS, in addition to LeafCutter. They also map eQTLs using the same data. They analyze sQTLs and eQTLs, together with summary statistics from brain-related complex traits. They found eQTLs and sQTLs to explain a similar fraction of heritability for these traits, while the corresponding causal variants tend to be distinct, and point to different genes associated with the traits. Although I find this submission interesting, I have substantial concerns regarding the methodology and the depth of certain downstream analyses, which need to be resolved:

1. The comparison between THISTLE and sQTLseeker (lines 110-130 and Supp. Note) is fairly

limited. First, authors use an old version of sQTLseeker (sQTLseeker2 is available 10.1038/s41467-020-20578-2). In addition, simulations i) are restricted to very few scenarios, which does not allow to make such general claims about statistical power (e.g. lines 21, 77-80, 115-117), and ii) require a much more detailed description. Comparison with a third method such as DRIMseq (doi: 10.12688/f1000research.8900.2) would be also valuable.

Re: We thank the reviewer for these comments. We have made the following improvements of the simulation study. First, we have used much broader ranges of the simulation parameters, including sample size (e.g., 300, 600, 900, and 1200), sQTL effect sizes (i.e., differences in isoform-eQTL effect between isoforms), degree of overdispersion of transcript abundance, and number of isoforms per gene (e.g., 2, 5, 10, 15, and 20). Second, apart from simulating transcript abundances numerically from a hypothetical distribution, we have added simulation scenarios in which transcript abundances were generated by sampling RNA-seq reads to mimic real RNA-seq data using Polyester (Frazee et al. 2015 Bioinformatics). This simulation setting retains the correlations of transcriptional abundance among isoforms and facilitates the comparison between methods that require different types of RNA-seq input data (e.g., TPM or read count). Third, we have added sQTLseeker2, DRIMSeq, and MANOVA into the method comparison. The updated simulation results showed that THISTLE was still well-calibrated under the null (**Supplementary Fig. 3**) and more powerful than sQTLseeker, DRIMSeq, and MANOVA, when evaluated at the same level of false-positive rate, in most scenarios under the alternative (**Supplementary Fig. 4**). We have provided all the details of the simulation study in section 1 of the **Supplementary Note** and updated the main text accordingly (lines 113-122; **Supplementary Figs. 3 and 4**).

1.1. It is unclear how authors study power (e.g. Supp. Fig. 2). Which is the H1 scenario selected (i.e. 3 or 4)? How many SNPs are under H1?. Please describe in detail. Power should be evaluated modifying effect sizes, rather than the P-value threshold (Supp. Fig. 2c-d). Power should be studied for various isoform numbers (e.g. typical protein coding genes with multiple isoforms have 2 - 25 isoforms), especially given that the number of terms in the THISTLE quadratic form increases dramatically with the number of isoforms ($\#terms \sim (isoforms)^2/2$). It would be also required to evaluate in depth the performance of both methods as a function of the degree of correlation between the responses and the sample size. MANOVA, DRIMseq or sQTLseeker model explicitly the covariances between different traits, but authors need to show how well THISTLE approximates various real covariance structures, especially with many isoforms.

Re: Shown in **Supplementary Fig. 2** were the simulation results from both scenarios 3 and 4 described in section 1 of the **Supplementary Note**. In brief, we simulated genotype data of 1000 unlinked SNPs in 500 individuals from binomial distributions and randomly selected one SNP as causal to generate transcriptional abundances of three isoforms for a hypothetical gene. The causal SNP has an additive effect on the transcriptional abundance of each isoform, and an sQTL effect was realised by creating differences between the SNP effects on the three isoforms (i.e.,

scenarios 3 and 4 described in section 1 of the **Supplementary Note**). Each simulation scenario was repeated 500 times, and power was calculated as the proportion of simulation replicates in which the sQTL effect was detected at a given significance level (i.e., true-positive rate or TPR). We evaluated the power at a range of significance levels to demonstrate that the difference in power between THISTLE and sQTLseekeR increased with increasing significance level. We thank the reviewer for pointing out this issue and have made the simulation setting clear in all the related figures in the revised manuscript.

To mimic the correlations of transcriptional abundance among isoforms, we have added simulation scenarios in which transcriptional abundances were generated by sampling RNA-seq reads using Polyester to mimic real RNA-seq data. We have also extended the ranges of the simulation parameters substantially, including sample size (300, 600, 900, or 1200), sQTL effects, degree of overdispersion of transcriptional abundance ($s = 600, 900, 1200, \text{ or } 1500$), and number of isoforms per gene (2, 5, 10, 15, or 20), and included more sQTL methods (i.e., MANOVA and DRIMSeq) into the comparison. In the revised manuscript, apart from TPR, we have also used AUC to compare statistical power between the methods because of the inflated FPR observed for DRIMSeq under the null (**Supplementary Fig. 3**). The updated simulation results showed that THISTLE was still well-calibrated under the null (**Supplementary Fig. 3**) and more powerful than sQTLseekeR, DRIMSeq, and MANOVA in most simulation scenarios (**Supplementary Fig. 4**), even for genes with a relatively large number of isoforms (**Supplementary Fig. 4d**). We have updated the main text accordingly (lines 113-122).

1.2. Authors should explain better multivariate Poisson data generation: e.g. how can this distribution have mean 0 (or are they centering)?, how is it possible that a linear regression fit to the marginal poisson values results in meaningful beta coefficients? or are they either fitting a glm or log/rank-inverse normal transforming? Please detail. In any case, authors should simulate under a model that takes into account overdispersion of RNA-seq data, such as the Dirichlet-multinomial. Indeed, it may be meaningful to add DRIMseq (doi: 10.12688/f1000research.8900.2) to the comparison of methods. It would be also important to evaluate how robust is THISTLE to stronger violations of multivariate normality (given that sQTLseekeR is non-parametric), and which is the impact of lowly expressed isoforms. Authors should compare/discuss the use of relative (as in sQTLseekeR) vs absolute (THISTLE, DRIMseq) isoform abundances.

Re: In our original simulation study, RNA-seq data generated from a multivariate Poisson distribution were normalized using the rank-based inverse normal transformation (RINT), and the normalized expression data were fitted in a linear regression model to estimate isoform-eQTL effects. We have added these details in the revised manuscript (section 1 of the **Supplementary Note**). To simulate a scenario that takes overdispersion into account, we have applied the function `simulate_experiment()` in Polyester to generate RNA-seq reads. The built-in transcript read count model assumes that the number of reads for each transcript is drawn from a negative binomial distribution across biological replicates. The parameter “size” in `simulate_experiment()` can be

used to control the negative binomial variance, i.e., $mean + mean^2/size$, where $size = readspertx * fold_changes / s$, with $readspertx$ being the read count per transcript and $fold_changes$ being the fold change of transcription abundance between groups. Thus, large s leads to small $size$ and large negative binomial variance. In this case, the RNA-seq data (as measured by TPM) are distributed with overdispersion and deviated strongly from multivariate normality (see **Fig. R1** for the distribution of the simulated RNA-seq data of an isoform given $s = 1500$). As mentioned above, we have added DRIMSeq and MANOVA into the comparison. The updated simulation results show that THISTLE was still well-calibrated under the null (**Supplementary Fig. 3**) and more powerful than sQTLseeker2, DRIMSeq and MANOVA under the alternative in most simulation scenarios (**Supplementary Fig. 4**). These results are consistent with that THISTLE identified more sGenes than sQTLseeker in real data analysis (**Supplementary Fig. 5**). We did not observe a large difference in the proportion of lowly expressed genes among the sGenes identified by different sQTL methods, with 2.4%, 2.9%, and 2.1% for THISTLE, sQTLseeker, LeafCutter & QTLtools, respectively (**Supplementary Fig. 12**). We also did not observe a pattern of differences between the two classes of method (whether using the relative or absolute abundances) according to the FPR and power quantified from our simulations. We have revised the manuscript accordingly to address the issues above (lines 113-122; **Supplementary Figs. 3, 4, 5 and 12**; section 1 of the **Supplementary Note**).

Fig. R1 Distribution of TPM of one simulated isoform

1.3. All comparisons between THISTLE, sQTLseeker and LeafCutter (e.g. lines 117-118, Supp. Fig 3, etc.) should be done on the same set of tested gene-variant pairs and uniformizing the analyses. Otherwise, the numbers provided are very difficult to compare, as the differences could be due to distinct filtering and/or parameter selection. Note that besides filtering, e.g. sQTLseeker does not report sQTLs with small effects, nor variance-sQTLs, and LeafCutter is able to find associations with novel introns.

Re: We thank the reviewer for this reminder and have done so in both simulations and real data analyses. In the simulations, the methods were compared under the same conditions. In the analyses of the ROSMAP data, we have also reported the scenario in which the comparison between THISTLE, sQTLseeker, and LeafCutter & QTLtools was limited to the same set of gene-

SNP pairs (**Supplementary Fig. 5b-d**). Even in this case (**Supplementary Fig. 5b**), the number of sGenes identified by THISTLE (5,362) was still much larger than that for sQTLseekeR (1,243), even larger than that for LeafCutter & QTLtools (4,275). Note that without such restriction, the number of sGenes identified by THISTLE was 6,289, compared to 1,761 for sQTLseekeR and 6,551 for LeafCutter & QTLtools (**Supplementary Fig. 5a**). We have commented on this in the revised manuscript (lines 129-137).

1.4. Besides the technical comparison of the methods, authors should compare the sQTLs obtained via different methods in terms of the splicing events that they capture, the genes that they affect, and the enrichment of the sQTLs in functional annotations.

Re: We have quantified the splicing events for the sGenes identified by THISTLE and sQTLseekeR based on the isoforms with the most positive and negative isoform-eQTL effects. The results showed that most sGenes identified by THISTLE undergo complex splicing events and that THISTLE can identify intron retention and alternative UTRs. The splicing events captured by THISTLE were similar to that by sQTLseekeR (**Supplementary Fig. 6**), which is not unexpected given that both of them are transcript-based methods and that 95% of the sQTLseekeR sQTLs were a subset of the THISTLE sQTLs (**Supplementary Fig. 5**). Since LeafCutter does not provide information about the flanking exons of the excised introns, we could not characterize the splicing events for the LeafCutter & QTLtools sGenes.

Regarding the splicing events associated with the sQTLs, given the strong overlap between the sQTLseekeR and THISTLE sQTLs (**Supplementary Fig. 5**), we could expect the functional enrichment of the sQTLseekeR sQTLs to be similar to that of the THISTLE sQTLs. We therefore only compared the functional enrichment of the THISTLE sQTLs to that of the LeafCutter & QTLtools sQTLs. The functional enrichment patterns (e.g., sQTLs were more enriched in splicing sites and RBPs binding sites than eQTLs) remained largely unchanged whether we stratified the sQTLs by mapping method (THISTLE vs. LeafCutter & QTLtools; **Supplementary Fig. 18**) or performed the enrichment analysis with TORUS using the full summary statistics without SNP selection (**Supplementary Fig. 19**). We have added these enrichment results in the revised manuscript (lines 134-138 and 250-253; **Supplementary Figs. 6 and 19**).

2. THISTLE computes p-values using the CDF of a quadratic form in normal variables. Although, there are exact methods to estimate it (e.g. Davies method), THISTLE relies on an approximation (lines 479-480). This approximation has a relative error that can be large, particularly in the case of low p-values (authors' ref 63, <https://www.jstor.org/stable/2673596?seq=1>, where they evaluate p-values only down to $1e-4$), and authors report p-values down to $1e-300$ (e.g. Supp. Fig 9). Beyond the discussion of whether such small p-values are meaningful, authors should assess the validity of the Saddlepoint approximation for small p-values, comparing with the exact method.

Re: We thank the reviewer for this suggestion and have compared THISTLE p-values computed from the Saddlepoint approximation to those from the Davies method in simulations. Our benchmarking result shows that the Saddlepoint approximation was still highly accurate for p-values down to $1e-14$ (i.e, a precision limit for the Davies algorithm suggested by the authors of sQTLseekeR2) and that there was no evidence that the approximation became less accurate for smaller p-values (**Supplementary Fig. 7**). Truncating p-values is problematic for downstream analyses that require p-values as the input, such as COLOC. We have revised the manuscript accordingly to address this concern (lines 138-140; **Supplementary Fig. 7**).

3. Authors use the Bonferroni method for multiple testing correction. Although common practice in GWAS, it has several pitfalls and is far from the gold standard in QTL mapping (i. e. permutations to correct for multiple variants tested per gene, accounting for LD, and FDR to correct for multiple genes tested genome-wide, 10.1093/bioinformatics/btv722). If permutations are too computationally intense, an alternative could be approximations such as eigenMT (10.1016/j.ajhg.2015.11.021).

Re: This comment relates to comment #4 from Reviewer #1. In the revised manuscript, we have employed a permutation-based approach to correct for multiple testing for both the cis-eQTLs and cis-sQTLs. For the eQTLs, we shuffled the gene-level transcriptional abundance phenotype across individuals and re-ran the eQTL mapping analysis with QTLtools. For the LeafCutter & QTLtools sQTLs, we shuffled the splicing phenotype (using the --grp-best option in QTLtools) across individuals and re-ran the sQTL analysis with QTLtools. For the THISTLE sQTLs, we permuted the isoform-level transcriptional abundance phenotype across individuals and re-ran the isoform-eQTL mapping and the THISTLE sQTL analysis using OSCA. In each of the three analyses above, we ran 1,000 permutations and fitted a beta distribution to the minimum p-values obtained from the permutations to compute the effective number of independent variants for each gene. We adjusted the original sQTL or eQTL p-values by this effective number of independent variants and employed the Storey's q-value approach to the lead sQTL or eQTL SNPs to account for multiple testing across genes. This procedure generated a p-value threshold of $3.4e-05$ for cis-eQTLs, $3.3e-05$ for THISTLE cis-sQTLs, and $5.9e-07$ for LeafCutter & QTLtools cis-sQTLs at an overall FDR of 0.05. All the thresholds were less stringent than $5e-8$ used in our earlier manuscript.

We have updated the manuscript accordingly (lines 167-168 and 192-194; section 4 of the **Supplementary Note**).

4. All along the text, authors talk about 'event-based' and 'transcript-based' methods for sQTL mapping (lines 58-59, 164, discussion, etc.). However, it is important to distinguish between methods to quantify splicing (indeed, 'event-based' such as LeafCutter or 'transcript-based' such as RSEM), and methods to map sQTLs (i.e. find statistical associations with the genotype, e.g FastQTL, sQTLseekeR or THISTLE). For instance, LeafCutter is not a method for sQTL mapping (line 164). LeafCutter intron-excision ratios can be used as input for sQTL mapping methods such

as linear regression in FastQTL, but also as input for sQTLseeker2 or THISTLE itself. Please clarify in the text.

Re: In the revised manuscript, we have made it clear that LeafCutter was the tool used to generate intron-excision ratio (i.e., a splicing phenotype) and call the corresponding sQTLs “the LeafCutter & QTLtools sQTLs” (lines 198, 202, 204, 315, 355, 387, and 389; **Figs. 4 and 5**).

5. Authors employ a meta-analysis across isoform-level eQTLs to map expression QTLs (according to Methods lines 528 - 530). Instead, they should use a standard method for eQTL mapping (e.g. FastQTL, 10.1093/bioinformatics/btv722) to model gene expression values, provided that they are available.

Re: We indeed used a standard method implemented in QTLtools for eQTL mapping (**Supplementary Fig. 10**). We included the statement about meta-analysis in our original manuscript for the purpose of explaining the relationship between isoform-eQTL and eQTL. To avoid confusion, we have removed this sentence in the revised manuscript (lines 557-560).

6. Regarding the functional enrichment of sQTLs/eQTLs, authors perform a very limited study, restricting their claims to the fact that sQTLs are more enriched in splice sites than eQTLs, and eQTLs more enriched in the vicinity of TSSs (lines 240-245), something expected and pointed out often in the literature (10.1126/science.aad9417, 10.1126/science.aaz1776, 10.1038/s41467-020-20578-2). However they do not describe at all the results of the enrichments in the functional annotation by REMC, which seem to be very similar between sQTLs and eQTLs (Fig 3c, d), nor explore important aspects that would be relevant to distinguish between eQTLs/sQTLs: enrichment in RBPs, chromatin marks (e.g. H3k27ac/H3k36me3), etc, which may provide insight into the differential mechanisms of sQTLs and eQTLs. They also restrict the location analysis to the distance to the TSS, while the distribution of eQTLs and sQTLs along gene bodies of upstream/downstream regions is known to be important (10.1126/science.aad9417,10.1038/ncomms5698). Overall, further analyses are required to make this section compelling.

Re: We thank the reviewer for these suggestions. We have quantified the enrichment of the sQTLs and eQTLs in eCLIP peaks of RBPs binding sites from ENCODE and seven histone marks from REMC. Together with the results from the previous enrichment analyses, we found that the sQTLs were more enriched in splicing regions (e.g., splice acceptors and splice donors) and RBPs binding sites than the eQTLs (**Fig. 3b & d**). The eQTLs were more enriched in TSS than the sQTLs (e.g., active TSS and bivalent TSS; **Fig. 3a**), consistent with our observation that the eQTLs were located much closer to TSS than the sQTLs (**Fig. 3e**). Both the sQTLs and eQTLs were significantly enriched in histone marks H3K4me1, H3K4me3, H3K9ac, H3K27ac, and H3K36me3 (**Fig. 3c**) but deflated in non-coding regions (**Fig. 3b**). Compared to that of the eQTLs, the enrichment of the sQTLs was higher in H3K36me3 but lower in H3K4me3, H3K9ac, or H3K27ac (**Fig. 3c**). We have

also quantified the distance of each lead sQTL or eQTL SNP to the intron, exon, upstream region, or downstream region of the corresponding gene and found that both the sQTLs and eQTLs tended to be enriched at both ends of gene bodies, exons, and introns (**Supplementary Fig. 17**). Compared to the eQTLs, the sQTLs tended to be more enriched at the ends of the introns and exons, where the splicing sites are located (**Supplementary Fig. 17**); the differences, however, were small. These additional analyses have provided more insights into the differences in potential mechanisms between the cis-sQTLs and cis-eQTLs. We have added these results in the revised manuscript (lines 234, 237-238, 243, and 245-253; **Fig. 3**; **Supplementary Fig. 17**).

7. According to the authors (line 293), THISTLE does not provide an estimate of the effects. If so, which is the input that they provide to colocalization/fine mapping methods? If they provide P-values generated by the Saddlepoint approximation, these do not come from a normal distribution, which breaks a key assumption of most colocalization/fine mapping approaches, including COLOC and CAVIAR.

Re: We have clarified in the revised manuscript that we used the sQTL p-values from both THISTLE and LeafCutter & QTLtools as input for the colocalization analysis (lines 312-313) and the signed sQTL z-statistics from only LeafCutter & QTLtools as input for the SuSiE fine-mapping analysis (**Supplementary Fig. 25**). We have demonstrated by additional simulations that the COLOC posterior probabilities (PP) computed from THISTLE sQTL p-values were well calibrated under different models (**Fig. R2**), suggesting that p-values generated by the Saddlepoint approximation can be applied to COLOC.

Fig. R2 Distributions of posterior probabilities from COLOC analysis with THISTLE sQTL p-values in simulations. We simulated genotype data for unlinked SNPs from binomial distributions using the method described in section 1 of the **Supplementary Note**. Two phenotypes were simulated under 5 scenarios fulfilling the five COLOC hypotheses (H0-H4). H0: no sQTL for either phenotype. H1: one sQTL for phenotype 1 but not for phenotype 2. H2: one sQTL for phenotype 2 but not for phenotype 1. H3: one sQTL for phenotype 1 and another distinct sQTL for phenotype 2. H4: one shared sQTL for both phenotypes. Each scenario was replicated 500 times. We also ran 500 eQTL simulations based on the same setting. The sQTL p-values were generated from THISTLE using the Saddlepoint approximation, and the eQTL p-values were generated from linear regression. Shown on each panel are the box plots of the five COLOC posterior probabilities (PP0-PP4). The result demonstrates that there is no apparent difference in performance between COLOC with THISTLE sQTLs (top five panels) and eQTLs (bottom five panels).

8. Authors claim that 32% of sGenes share causal variants with the eGenes. First, authors should only make statements on the set of genes that they can explicitly test for colocalization (that is, genes that are both sGenes and eGenes). Hence, this number is indeed 50% (2418/4893), lower than the $100 - 32 = 68\%$ reported. Moreover, that x% share a causal variant does not imply that $100-x\%$ do not share it (if I understand correctly, that COLOC is not able to define whether they are the same signal -PP4 is low-, does not imply that they are different signals -PP3 is high-, what about the probabilities for the other hypotheses H0-H2?). To provide a better estimate of the fraction of shared/independent signals, authors should use a combination of different

colocalization methods, including approaches that (in contrast to COLOC) are not restricted to only 1 causal variant per locus.

Re: This comment relates to comment #4 from Reviewer #2. First, the reviewer is correct that if we focus only on the sGenes in common with the eGenes, the percentage of sGenes that share causal variants with the eGenes is ~50%. However, to quantify the number of sQTL signals distinct from the eQTL signals, we need to take the unique sGenes into account, which explains why the reported proportion of distinct sQTLs (i.e., 68%) is higher than 50%. Second, we agree with the reviewer that x% of the sGenes showing evidence of sharing causal variants with sGenes does not necessarily mean that 100-x% of the sGenes that do not share.

According to the comments from both Reviewers #2 and #3, we have opted to use PP3 to quantify the proportion of sQTL signals distinct from eQTL signals and have done so with the updated sQTL and eQTL data in the revised manuscript. There were 9,035 genes with both cis-sQTLs and cis-eQTLs, among which 4,377 genes had PP3 > 0.8 (**Fig. 2b**). Together with 3,405 unique sGenes without eQTLs, ~61% of the sGenes, $(4,377 + 3,405)/12,794$, showed sQTL signals distinct from eQTL signals, a proportion not too dissimilar to that quantified in our previous manuscript (i.e., 68%). The result remained largely unchanged when we performed the colocalization analysis with eCAVIAR (Hormozdiari et al. 2016 AJHG) that accounts for multiple causal variants in a locus (**Supplementary Fig. 16**). We have updated the manuscript accordingly (lines 204, 225-226, 328-330, and 342-343; **Fig. 2b**; **Supplementary Fig. 16**).

9. In their heritability analyses, authors show that i) sQTLs and eQTLs contribute at a similar degree to the selected traits and ii) lead to the identification of different genes via MR. Although i) has been shown before to some extent (10.1126/science.aad9417, 10.1038/s41467-020-20578-2), i believe that ii) is a novel and interesting result. However, this claim is based on very few genes (e.g. 68% ~ 73/107) and it is unclear how generalizable it is. In addition, when they explicitly test for sharing causal variants of the genes identified in common (34), they find that the vast majority (25) are due to the same causal variants, which contrasts with their previous claims.

Re: Regarding i), the relative contributions of sQTLs and eQTLs to trait variation have been demonstrated in previous studies largely by the genomic inflation factor, which is sample size dependent. Heritability enrichment analysis in Hormozdiari et al. shows that the contribution of the sQTL category is not statistically significant when fitted jointly with the eQTL category in an LD score regression model. So, the results in the literature are not consistent. We attempted to collect the largest available data for sQTL mapping and quantified the relevance of sQTLs to complex trait variation by several distinct analyses, including the heritability enrichment and mediation analyses to quantify the contribution of sQTLs to the overall polygenic effects and the colocalization analyses to pinpoint specific complex trait genes through sQTLs. All the analyses were benchmarked with eQTLs identified from the same data. To the best of our knowledge, this

is the most powerful and comprehensive study to date to quantify the role of sQTLs in complex trait variation.

Regarding the generalizability of our conclusion, we have expanded the colocalization analysis (using SMR & COLOC PP4) to 7 additional traits (not necessarily brain related) for which summary statistics from large GWAS ($n > 600,000$) are available. The number of GWAS signals shared with sQTLs (568) was still comparable to that for eQTLs (480), and the proportion of trait-associated sGenes not detected using eQTL data remained consistent (~62%) (**Supplementary Fig. 28**). We have updated the manuscript accordingly (lines 333-335; **Supplementary Fig. 28**).

Regarding the question about the genes with both sQTL and eQTL signals, we have checked whether genes with shared sQTL and eQTL signals were more likely to be mapped to the complex traits. We found that 2.9% of the genes with distinct sQTL and eQTL signals were mapped to complex traits, and such proportion was 3.1% for genes with shared sQTL and eQTL signals. Thus, there was no substantial difference in probability to be mapped to complex traits between the two sets of genes. We believe that the problem was caused by that when counting the complex trait genes with both sQTL and eQTL signals, genes with distinct sQTL and eQTL signals were less likely to be included in the 96 overlapping gene set because it required the distinct signals to be mapped to the traits consistently. We have revised the manuscript accordingly to avoid the confusion (lines 344-346).

Minor points:

- As a general comment, I find that the methods section (and the supplementary note) should be more detailed.

Re: We have added more method details in the revised manuscript (lines 497-499, 514-518, 536-541, 545-546, 549-554, 557-564, and 569-571; **Supplementary Note**).

- Summary statistics for GWAS traits and gene expression are routinely generated. However it is unclear whether this is also the case of isoform-level summary statistics, and therefore, how useful is this feature of THISTLE (lines 104, 371), which is presented as one of its advantages. Please justify.

Re: We agree and have toned down the statements about this attribute of THISTLE in the revised manuscript (lines 140-142).

- Regarding covariate correction across QTL analyses (lines 523, 539), if I understand correctly, variance partition is applied, and then the effect of known covariates is removed from original responses via an LMM. Later, residuals are regressed on genetic PCs and PEER factors. Is this the

case? Are PEER factors learnt from these residuals? Why this procedure, instead of learning PEERs from the original matrix, providing known covariates as an argument (10.1038/nprot.2011.457)? How was the number of PEERs selected? Why different numbers for THISTLE and LeafCutter analyses? Please give more details and justify.

Re: The short answer to the reviewer's questions is that we largely followed the GTEx pipeline (GTEx consortium, 2020) for quality controls, LeafCutter & QTLtools sQTL mapping, and eQTL mapping. Regarding the question about whether the PEER factors should be computed before or after covariate adjustment, we believe that the difference are likely to be very small, as evidenced by the strong correlation between the two residuals (e.g., median correlation of 0.91 for either all genes or genes with sQTLs or eQTLs in the MSBB dataset). Regarding the question about the number of PEER factors included in the eQTL or sQTL mapping, again we followed the GTEx pipeline. For eQTL mapping or isoform-eQTL mapping as part of the THISTLE analysis, the number of PEER factors used was determined by sample size (N): 15 factors for $N < 150$, 30 factors for $150 \leq N < 250$, 45 factors for $250 \leq N < 350$, and 60 factors for $N \geq 350$. For the LeafCutter & QTLtools sQTL mapping analysis, since the proportion of variance in transcriptional abundance explained by PEER factors was small, only 15 PEER factors were included in the sQTL mapping analysis in all the datasets. We have clarified this in the revised manuscript (lines 549-552).

- Intron excision ratios are standardized and quantile-normalized (line 539). According to methods, Transcript expressions (TPMs) do not undergo any transformation. However in Supp. Fig. 8 they appear to be rank-inverse normalized. Please clarify. If so, how is this normalization affecting the correlation between isoforms? Authors should study this transformation in their simulations.

Re: We have clarified in the Methods section that the TPM were processed by quantile normalization followed by rank-based inverse normal transformation (lines 545-546 and 554). In the simulation study, we have simulated RNA-seq reads to mimic real RNA-seq data and used the same normalization procedure as we did in the real data analysis. The simulation result shows that the THISTLE statistics remained well-calibrated with such transformation (**Supplementary Fig. 3**).

- Fig 3. (Supp. Fig 18) Enrichment in functional annotations should be done using a null distribution of matched variants regarding both MAF and distance to TSS (or better, their relative location: introns, exons, downstream, upstream + the actual distance to TSS/TTS). There is no need to do these two analyses separately.

Re: We thank the reviewer for this suggestion. The null distributions in the revised manuscript were generated based on "control variants" randomly sampled with MAF and distance to TSS matched to the variants in query (lines 241 and 589).

- Fig 4a should be supplementary and leave just Fig 4b, which corrects for the ascertainment bias. In addition, authors should use as null a set of SNPs matched with respect to MAF, but also with respect to their location (see above) both here and in Supp. Figs 14a, 15a, 16a. I wonder also whether the main figure should use the fine mapped (instead of the top 1 QTLs). Please uniformize the y-axis label and the notation regarding heritability fractions in the figures and along the text (e.g. $\Pr(h^2_1)$ vs h^2_1).

Re: We have moved original **Fig. 4a** to the supplementary (now **Supplementary Fig. 22**) and updated the heritability enrichment analyses with the control variants sampled at random with MAF and distance to TSS matched to the variants in query in all the figures mentioned by the reviewer. Given that the heritability enrichment results based on the lead cis-sQTL/eQTL SNPs were similar to those based on the fine-mapped cis-sQTL/eQTL SNPs, we keep the results based on the lead SNPs in the main text (to be consistent with the functional enrichment analysis) and include the results with the fine-mapped SNPs in the supplementary. We have also uniformized the y-axis labels regarding the proportion of heritability explained in the relevant figures and text (**Fig. 4a; Supplementary Figs. 23, 24 and 25**)

- Figs 3, 4, and supp 12, 14, 15, 16, 20. Besides the ratios, please show the actual number of overlaps/the actual heritabilities in the plots.

Re: Done (**Figs. 3 and 4a; Supplementary Figs. 22a, 23a, 24a, and 32**).

- Lines 71-72. Please clarify what you mean by 'subtle splicing patterns'.

Re: We have change it to "subtle splicing events" which refers to the formations of RNA transcripts by removing or including a small number of nucleotides (e.g., one exon), such as exon skipping. We have clarified this in the revised manuscript (lines 66-67).

- Why including so small datasets (i.e. BipSeq n = 27, UCLA-ASDn = 64) in the analysis? This barely increases power while may reduce the number of variants analyzed. Please justify.

Re: We have excluded these two small datasets and added additional six datasets in our new analysis. The smallest sample size has now been 128 (**Supplementary Table 1**).

- Lines 125-126. Default p-values in sQTLseeker are not computed using permutations, but rather an asymptotic approximation. In sQTLseeker2, p-values are down to $1e-14$ via the Davies method (10.1038/s41467-020-20578-2).

Re: We have compared THISTLE with sQTLseeker2 and made the corresponding changes regarding the statements about sQTLseeker in the revised manuscript (lines 138-140).

- Line 154. Authors explore gene bodies +/- 2Mb. Why such a large window? How do you expect a variant e.g. 2Mb downstream to affect splicing? Please justify.

Re: We believe that the genetic regulatory mechanisms of splicing are largely unknown so that using a larger window size would introduce less ascertainment biases with respect to the genomic locations of the sQTL SNPs. In fact, we have observed in our real data analyses sQTL SNPs located more than 1 Mb apart of the corresponding genes (**Fig. 3e**) or even on different chromosomes (**Fig. 6**).

- Line 168. Please clarify in Methods how LeafCutter clusters are mapped to genes. Specify also which genes you focus on (e.g. only protein coding/lncRNAs/all).

Re: We used the R function `map_clusters_to_genes()` provided by LeafCutter and exon annotation from GENCODE v37 to map clusters to genes. All the transcripts available in the RNA-seq data, including those from protein-coding genes and lncRNAs, were included in our analysis. We have clarified this in the Methods section (lines 514-515 and 569-571).

- Line 195. Which fraction of sQTLs identified by LeafCutter corresponds to novel introns/isoforms? Please specify.

Re: We have specified in the revised manuscript that “203,889 unique cis-sQTL SNPs for 1,148 clusters with unknown associated genes (7% of the significant intron clusters)” (lines 191-192).

- Line 199. The R shiny app does not work, it does not return any result after entering data in the required fields (I just tried Linux+Firefox).

Re: We have significantly improved the online tool to query the sQTLs/eQTL summary data using PheWeb: <https://yanglab.westlake.edu.cn/resources/ql/>. We have asked multiple interactional collaborators to access it from different locations of the globe. It should work now.

- Line 256-259. Please specify the actual traits studied in the non-case/control GWASs in Methods and in Supp. Table 1.

Re: Done (lines 536-541 and **Supplementary Table 4**).

- Lines 278-279. Please define ‘clumped SNPs’ and give details on how clumping was done in Methods.

Re: Done (**Supplementary Fig. 24**). The text now reads: “To select independently associated cis-sQTL/eQTL SNPs, we performed PLINK clumping analyses with the cis-sQTL or cis-eQTL

summary statistics using an LD r^2 threshold of 0.10, a window size of 2 Mb, and a p-value threshold of 5×10^{-8} .

- Lines 283-284 and 585-588. Please state explicitly the difference between the meaning of tau, and how it differs from per-SNP heritability to help the reader. In addition, (fig 4d) discuss why tau is larger for sQTLs, while the fraction of heritability is larger for eQTLs.

Re: τ is the partial regression coefficient for an annotation category when fitted jointly with the other annotation categories in an LD score regression model. If all the annotation categories are disjoint (no overlapping SNP among categories), τ can be interpreted as the per-SNP heritability of the corresponding annotation category. In the presence of overlaps among the annotation categories, the interpretation of τ is complicated. However, it can still be used to quantify the contribution of an annotation category to the overall SNP-based heritability (h_G^2), conditioning on the other categories. The SNP-based heritability for an annotation category (h_C^2) is computed based on the estimate of τ for the focal category as well as the estimates of τ for other categories overlapping with the focal category. Hence, h_C^2 is a marginal term, meaning that the sum of h_C^2 across all annotation categories is large than h_G^2 in the presence of overlaps among the annotation categories. The fold enrichment of heritability in an annotation category is computed as $\frac{h_C^2 M}{h_G^2 C}$ with C being the number of SNPs in the focal category and M being the overall number of SNPs. In addition, through the MESC analysis we attempted to estimate the proportion of h_C^2 mediated by the cis-sQTLs or cis-eQTLs discovered in this study. We have made these definitions clear in the revised manuscript (lines 286-291 and 615-621).

Regarding the comment on the difference in τ , heritability enrichment, or mediated heritability between the sQTLs and eQTLs. With the updated sQTL and eQTL data, the differences remained, i.e., the sQTLs tended to have a larger estimate of τ and higher level of heritability enrichment but lower level of mediated heritability than the eQTLs. Nevertheless, the discrepancy needs to be interpreted with caution given the large standard errors of the estimates. One possible source of the discrepancy is that τ and heritability enrichment were estimated using the sQTLs from both THISTLE and LeafCutter & QTLtools, whereas the mediated heritability was estimated using only the LeafCutter & QTLtools sQTLs. For instance, there was nearly no difference between the sQTLs and eQTLs if τ was estimated using only the LeafCutter & QTLtools sQTLs (mean $\hat{\tau} = 5.64e-7$ for sQTLs and $5.85e-7$ for eQTLs across the 12 traits). We have discussed this in the revised manuscript (lines 275-277 and 283-286).

- Lines 308-329. Some numbers seem not to coincide with the ones provided in Supp table 2. Please specify in the legend what the parentheses means in the LeafCutter/No. of coloc genes column.

Re: Done.

- Line 315. As far as I am concerned, LeafCutter is not able to model intron retention events (<https://www.biorxiv.org/content/10.1101/463927v1.full.pdf>). Please clarify.

Re: What we meant is that the genetic variant is in the retained intron. We have removed this example to avoid confusion.

- Line 574. Why are non-top-1 sQTLs annotated the same as non-sQTLs (i.e. 0)? Why restricting to top 1? Please justify.

Re: We focused on the lead sQTL SNP because it is the most informative one for an sQTL signal. Nevertheless, we have used several other annotation strategies, including annotating all the significant sQTL SNPs as 1 and non-significant SNPs as 0, annotating the clumped sQTL SNPs as 1 and the other SNPs as 0, or annotating the fine-mapped sQTL SNPs as 1 and the other SNPs as 0. The results remained largely unchanged (lines 283-286).

- Figs. 2a, 4e and Supp. 11a, 21. Please plot Venn diagrams instead. For two groups a simple Venn diagram is much more intuitive and easy to understand.

Re: Done.

- Supp. Fig 11b. Please add violin plots as in 2b.

Re: Done (Now **Supplementary Fig. 15**).

- Supp. Fig 13. Authors should show the actual QQplots too.

Re: We have shown QQ plots for the 12 traits in the revised manuscript (**Supplementary Fig. 20**).

- Supp. Note. Please specify exactly how you simulate SNPs using the binomial model.

Re: We simulated the genotype data of an SNP using the R function `rbinom(n, 2, p)` with n being the sample size and p being the minor allele frequency of the SNP, generated from a uniform distribution using the R function `runif(1, 0.05, 0.5)`. We have added these details in section 1 of the **Supplementary Note**.

- Supp. Note. In the MVN scenario, how do authors restrict that isoform expression to be > 0 ? Please clarify.

Re: We did not restrict the isoform expression to be > 0 .

- Please provide a supplementary “summary” table with the number of eQTLs and sQTLs identified using the different approaches.

Re: Done (Supplementary Table 2).

Decision Letter, first revision:

15th Mar 2022

Dear Jian,

Your Article, "Genetic control of RNA splicing and its distinctive role in complex trait variation" has now been seen by 3 referees. You will see from their comments below that while they find your work of interest, some important points are raised. We are interested in the possibility of publishing your study in Nature Genetics, but would like to consider your response to these concerns in the form of a revised manuscript before we make a final decision on publication.

Briefly, the three referees are all very impressed with the revision, acknowledging the major improvement and sounding supportive of an eventual publication.

Reviewer #1 is fully satisfied. Reviewer #2 makes one new request: that as much of the QTL summary statistics be shared as possible.

Reviewer #3, on the other hand, raises a few more questions. We think these are thoughtful suggestions that, if addressed, would usefully improve the impact of your study; pleasingly, they do not seem to require extensive further analysis.

To guide the scope of the revisions, the editors discuss the referee reports in detail within the team, including with the chief editor, with a view to identifying key priorities that should be addressed in revision and sometimes overruling referee requests that are deemed beyond the scope of the current study. We hope that you will find the prioritized set of referee points to be useful when revising your study. Please do not hesitate to get in touch if you would like to discuss these issues further.

We therefore invite you to revise your manuscript taking into account all reviewer and editor comments. Please highlight all changes in the manuscript text file. At this stage we will need you to upload a copy of the manuscript in MS Word .docx or similar editable format.

10*2) If you have not done so already please begin to revise your manuscript so that it conforms to our Article format instructions, available

[here](http://www.nature.com/ng/authors/article_types/index.html).

*3) Include a revised version of any required Reporting Summary:

[REDACTED]

We hope to receive your revised manuscript within four to eight weeks. If you cannot send it within this time, please let us know.

Sincerely,

Michael Fletcher, PhD
Associate Editor, Nature Genetics

ORCID: 0000-0003-1589-7087

Referee expertise:

Referee #1: neurogenetics

Referee #2: computational biology, genomics

Referee #3: QTL analysis, bioinformatics

Reviewers' Comments:

Reviewer #1:

Remarks to the Author:

The authors have provided a very comprehensive response to the review. They have satisfactorily addressed all of my previous comments.

Reviewer #2:

Remarks to the Author:

The authors have done an extremely thorough job of addressing all the reviewers concerns.

My only criterion for publication is that full summary statistics should be made available, including the COLOC/SMR results. It would also be great to share the full trans QTLs although I realize this may be challenging given the scale: maybe clumped or pruned variants could be shared instead.

Reviewer #3:

Remarks to the Author:

Reviewer #3

The authors have resolved the majority of the concerns that I expressed in my previous revision, particularly from the methodological standpoint. I believe that the improvement of the manuscript has been remarkable, and would like to congratulate them for their effort. I think, however, that a couple of additional points should to be addressed:

1. The original simulation framework has been substantially expanded by considering a more realistic "RNA-seq-like" scenario, modifying different parameters and including additional tools in the comparison. In such scenario, all methods (except for DRIMSeq) display controlled FPR. Regarding TPR, although THISTLE displays an improvement over the other methods in different scenarios, this improvement is small (particularly wrt. sQTLseeker2, from the plots I would say that the difference in

12power -AUC- is on average ~5%). I believe that authors should acknowledge this in the text (e.g. showing the actual average % of improvement in power -AUC-). Nevertheless, I find that these results are consistent with the statistics behind each of the methods and my prior experience with them. What strikes me is how this is compatible with the huge difference observed when analyzing ROSMAP data (~4 fold). In my previous review, I suggested that this could be due to additional filters in sQTLseeker2, as authors now claim in the text. However, as I proposed back then (point 1.3): why not directly compare the same set of tests performed? That is easy and would give a clearer picture. I mean i) finding the common set of N tests (actual SNP-gene pairs TESTED, not just inputted to both THISTLE and sQTLseeker), ii) subsetting these N tests for each method, iii) performing multiple testing correction on the P-values of the subsetted N tests for each method and iv) determining the actual overlap (this affects the main text and Supp. Fig. 5). Also, to make a fair comparison, the same covariates should be included (I suggest dropping PEERs in both analyses just for comparison, as they are not matched to sQTLseeker's phenotypes (sqrt-relative abundances)). Authors also suggest that a second possible aspect that affects the power of both methods is the stringency of the correction (as shown in Supp. Fig. 2c-d). Related to this, I still think that several aspects of this initial simulation are quite unclear (this involves also Supp. Fig. 1): how is *re* obtained? why does the variance of each isoform depend on the effect of the causal variant? The drop in power the same for any other structure for S? (I have the impression that this result corresponds to a very specific scenario and the behaviour may vary substantially, e.g. what if S is the identity matrix or a structured matrix of 0.2's with 1's in the diagonal?). How can you ensure that *y* takes only positive values (sQTLseeker requires that, as it performs a sqrt transformation)? I agree that the primary goal of the authors is not to benchmark methods, but I consider that all this information is relevant for potential users.

2. In reply to my previous review (point 7), authors performed some simulations, showing that COLOC can work well with THISTLE's P-values, which come from a distribution that deviates substantially from a normal distribution. They should show this also for CAVIAR, provided that they use it too, and include both simulation results and plots in the manuscript (currently these are only attached in the response letter). Showing that these two widely used co-localization methods are robust to the use of P-values coming from non-normal distributions will be really useful for many researchers in the field and adds even more value to the paper.

3. As suggested by reviewers, including myself (point 3 in previous review), authors performed a permutation-based multiple testing correction. However, they still use for all downstream analyses the 5e-8 threshold results. Authors should justify this well, showing at least that their choice is more stringent and that the overlap between both sets of results (permutations and 5e-8) is large enough (for all the sets of xQTLs generated).

4. Finally, it would be of high value for the method development community and the final users if authors made the source code of OSCA/THISTLE available in a platform such as GitHub/GitLab/Bitbucket (if this is done already it is not clear from the manuscript), and provide some information on the running time (even better if this is in comparison with other tools).

Minor comments:

-lines 54-68, 74-77. This relates to point 4 from my previous review. As they did in other sections of

13the paper, authors should make clear in the introduction that 'event-based' or 'transcript-based' is not a feature of the sQTL mapping statistical method but rather of the splicing phenotype definition/quantification. Again, LeafCutter is not a method for sQTL mapping (lines 60-61), and THISTLE/sQTLseekeR are not transcript-based methods (only), as they can work with both transcript-based and event-based quantifications (indeed: <https://doi.org/10.1038/s41467-020-20578-2>, Supp. Note. 2) .

- Please use the same color codes along the manuscript (both main and supplementary figures) to refer to the same things (e.g. in Fig 4 you have different colors for eQTLs and sQTLs on each panel).
- Fig. 3d. typo: "fol" instead of "fold".
- Fig. 3e. probably more informative to represent $\log_{10}(\text{distances})$
- lines 50-52: "with a rich collection of sQTLs and a broader spectrum of splicing events". This cannot be known before performing any analysis.
- line 232. Is 3,405 the number of genes that are only sGenes? Please clarify in the text.
- line 295. hg2 is not defined before
- end of line 496. You mean identity matrix?
- Supp. Fig. 3 and other places: "when one simulation parameter was fixed, all the other parameters varied". I do not understand this. Do you mean "when one simulation parameter varied, all the other parameters were fixed"? If so, fixed to which values?
- lines 556-560. Authors replied to my previous review (point 5) with "we indeed used a standard method implemented in QTLtools for eQTL mapping", but this contrasts to what is stated here. Please clarify.
- Supp. Figs. 22, 23,24 "to the proportion of the SNPs" you mean "to the proportion of heritability explained by the control SNPs"?
- Overall, I think figure quality could be improved

Author Rebuttal, first revision:

Reviewer #1:

Remarks to the Author:

The authors have provided a very comprehensive response to the review. They have satisfactorily addressed all of my previous comments.

Re: We thank the reviewer for the positive remark and the previous comments that have significantly improved our manuscript.

Reviewer #2:

Remarks to the Author:

The authors have done an extremely thorough job of addressing all the reviewers concerns.

Re: We thank the reviewer for the positive remark.

My only criterion for publication is that full summary statistics should be made available, including the COLOC/SMR results. It would also be great to share the full trans QTLs although I realize this may be challenging given the scale: maybe clumped or pruned variants could be shared instead.

Re: The full summary statistics for all the cis-sGenes, cis-eGenes, trans-sGenes, and trans-eGenes (265.8 GB in total) have been available through a PheWeb application at <https://yanglab.westlake.edu.cn/data/brainmeta>. There is a “Download summary statistics” button on the top right corner of each association p-value plot, which can be brought up by clicking the name of a gene or intron on the list.

To facilitate data download and genome-wide analyses for users, we have merged the data files of individual genes or introns by chromosome, stored them in SMR BESD format (or sparse BESD format for trans-sQTLs and trans-eQTLs to save storage), and made the BESD files available in a tarball for each of the four datasets above at https://yanglab.westlake.edu.cn/pub_data.html.

The full summary statistics from the SMR and COLOC analyses are available at https://yanglab.westlake.edu.cn/pub_data.html.

Reviewer #3:

Remarks to the Author:

The authors have resolved the majority of the concerns that I expressed in my previous revision, particularly from the methodological standpoint. I believe that the improvement of the manuscript has been remarkable, and would like to congratulate them for their effort.

Re: We thank for the reviewer for the positive remarks.

I think, however, that a couple of additional points should to be addressed:

1. The original simulation framework has been substantially expanded by considering a more realistic “RNA-seq-like” scenario, modifying different parameters and including additional tools

in the comparison. In such scenario, all methods (except for DRIMSeq) display controlled FPR. Regarding TPR, although THISTLE displays an improvement over the other methods in different scenarios, this improvement is small (particularly wrt. sQTLseeker2, from the plots I would say that the difference in power -AUC- is on average ~5%). I believe that authors should acknowledge this in the text (e.g. showing the actual average % of improvement in power - AUC-). Nevertheless, I find that these results are consistent with the statistics behind each of the methods and my prior experience with them.

Re: As per the reviewer's suggestion, we have included the difference in AUC between the methods in the revised manuscript (lines 113-114). However, as pointed out by the reviewer below, the small difference in AUC does not seem consistent with the large differences between the numbers of sGenes identified by the two methods in real data analysis. We reason that such discrepancy is because the difference in AUC does not reflect the difference in true-positive rate (TPR) at a specific level of false-positive rate (FPR). AUC is a metric to measure the predictive power of a classifier, and the overall AUC measures the predictive power for the whole range of FPR. In an association analysis, however, we are often only interested in the TPR of a method when the FPR is controlled at a low level. We had observed from the first set of simulations (i.e., simulating mRNA abundances from a multivariate distribution) the difference in TPR between THISTLE and sQTLseeker became larger when a smaller p-value threshold was used. This observation has now been replicated in the second set of simulations (i.e., simulating mRNA abundances by sampling RNA-seq reads from reference sequence) and further in the analysis of the ROSMAP data (see our response to the comment below for more details). Hence, the results are not inconsistent but reflect the difference in power between methods quantified at different significance levels.

What strikes me is how this is compatible with the huge difference observed when analyzing ROSMAP data (~4 fold). In my previous review, I suggested that this could be due to additional filters in sQTLseeker2, as authors now claim in the text. However, as I proposed back then (point 1.3): why not directly compare the same set of tests performed? That is easy and would give a clearer picture. I mean i) finding the common set of N tests (actual SNP-gene pairs TESTED, not just inputted to both THISTLE and sQTLseeker), ii) subsetting these N tests for each method, iii) performing multiple testing correction on the P-values of the subsetting N tests for each method and iv) determining the actual overlap (this affects the main text and Supp. Fig. 5). Also, to make a fair comparison, the same covariates should be included (I suggest dropping PEERs in both analyses just for comparison, as they are not matched to sQTLseeker phenotypes (sqrt-relative abundances)). Authors also suggest that a second possible aspect that affects the power of both methods is the stringency of the correction (as shown in Supp. Fig. 2c-d). Related to this, I still think that several aspects of this initial simulation are quite unclear (this involves also Supp. Fig. 1): how is *re* obtained? why does the variance of each isoform depend on the effect of the causal variant? The drop in power the same for any other structure for *S*? (I have the impression that this result corresponds to a very specific scenario and the behaviour may vary substantially, e.g. what if *S* is the identity matrix or a structured matrix of 0.2's with 1's in the diagonal?). How can you ensure that *y* takes only positive values (sQTLseeker requires that, as it performs a sqrt transformation)? I agree that the primary goal of the authors is not to benchmark methods, but I consider that all this information is relevant

for potential users.

Re: We confirm that all the comparisons between THISTLE and sQTLseeker (including that in the analysis of the ROSMAP data) were based on the same variant-gene pairs. Following the discussion in our response to the reviewer comment above, the small difference in overall AUC does not necessarily mean that the difference in TPR was small at any specific significance level. As mentioned above, we had observed in the first set of simulations (i.e., mRNA abundance generated from a multivariate distribution) that the difference in TPR between THISTLE and sQTLseeker increased when a more stringent p-value threshold was used (**Supplementary Fig. 2**). We have now replicated this observation in the second set of simulations (i.e., mRNA abundance generated by sampling RNA-seq reads from reference sequence) (**Supplementary Fig. 4e**), and further in the analysis of the ROSMAP data (**Supplementary Fig. 5d**). These results suggest that the TPR of THISTLE can be substantially higher than that of sQTLseeker when a stringent p-value threshold is used (e.g., $5e-8$ in real data analysis).

Another factor that was not discussed previously is covariate adjustment. We have double-checked and realized that the previous sQTLseeker2 analysis was performed without fitting covariates. We apologize for the oversight. With the enhanced version of sQTLseeker2 (v1.1.0), we have now been able to include in the sQTLseeker analysis the same set of covariates (including PEER factors) as used in the THISTLE analysis and discovered more sGenes by sQTLseeker, suggesting a gain of power due to covariate adjustment. However, even under this circumstance, the number of sGenes detected by THISTLE was still 2.1-fold larger than that by sQTLseeker in the ROSMAP data (3.3-fold in the CMC data). We have further tested the scenario where covariate adjustment was not performed for both methods and found that the number of sGenes decreased for both methods, but the ratio of THISTLE to sQTLseeker remained to be 1.7 in the ROSMAP data (1.6-fold in the CMC data).

In summary, our simulation results show that, although the overall AUC for THISTLE was only slightly (4.7% on average) larger than that for sQTLseeker, the difference in TPR between the methods increased with the increased $-\log_{10}(\text{p-value})$ threshold (**Supplementary Figs. 2 and 4**). In the analysis of the ROSMAP data with covariate adjustment, the number of sGenes for THISTLE was 2.1-fold larger than that for sQTLseeker, and the ratio decreased to 1.9 at $p < 1e-6$, to 1.2 at $p < 1e-4$, and eventually to nearly 1 at $p < 0.001$ (**Supplementary Fig. 5d**), consistent with the simulation results. We have clarified this in the revised manuscript (lines 123-125, 127-128, 130-132, and 134-135).

Here are our answers to the reviewer's questions regarding the simulation setting.

1) r_e was randomly sampled from the correlations of mRNA abundance between isoforms observed in the ROSMAP data.

2) We used a classical quantitative genetic model, $y = xb + e$, to generate isoform abundance (y), where x is the SNP genotype variable, b is the causal effect, and e is the residual. Given this model, we have $\text{var}(y) = \text{var}(x)b^2 + \text{var}(e)$. So, per definition, $\text{var}(y)$ depends on b , or more precisely on b^2 . We used the equation $\text{var}(e) = 2p(1-p)b^2(\frac{1}{q^2} - 1)$, with p being the allele frequency and q^2 being the proportion of variance in isoform abundance explained by the causal effect, to

generate residual variance, $var(e)$, to ensure the proportion of variance in y explained by x to be q^2 .

3) We have explained above that the difference in power between the methods is primarily driven by the choice of p-value threshold, and the observation can be replicated in real data. So, we do not think the setting of the variance-covariance matrix (**S**) would cause a bias, especially given that the elements in **S** were randomly sampled from correlation values observed in real data.

4) We added a sufficiently large positive mean value to the simulated transcription abundance of each isoform across individuals to ensure all the isoform abundances to be positive.

We have improved the corresponding texts in the **Supplementary Note** to avoid confusion.

2. In reply to my previous review (point 7), authors performed some simulations, showing that COLOC can work well with THISTLE's P-values, which come from a distribution that deviates substantially from a normal distribution. They should show this also for CAVIAR, provided that they use it too, and include both simulation results and plots in the manuscript (currently these are only attached in the response letter). Showing that these two widely used co-localization methods are robust to the use of P-values coming from non-normal distributions will be really useful for many researchers in the field and adds even more value to the paper.

Re: We thank the reviewer for this suggestion and have included the result from the COLOC analysis with THISTLE p-values in the revised manuscript (**Supplementary Fig. 17**). For clarification, we did not run the eCAVIAR analysis with the THISTLE sQTLs because of the lack of signed z-statistics. The eCAVIAR analysis presented in **Supplementary Fig. 18** was performed with the LeafCutter & QTLtools sQTLs. We have clarified this in the revised manuscript (legend of **Supplementary Fig. 18**).

3. As suggested by reviewers, including myself (point 3 in previous review), authors performed a permutation-based multiple testing correction. However, they still use for all downstream analyses the $5e-8$ threshold results. Authors should justify this well, showing at least that their choice is more stringent and that the overlap between both sets of results (permutations and $5e-8$) is large enough (for all the sets of xQTLs generated).

Re: As mentioned in our manuscript, we used a p-value threshold of $5e-8$ because it is the most commonly used genome-wide significance threshold in GWAS and the default threshold used in SMR to select instrument SNPs (lines 169-174). As per the reviewer's suggestion, we have now compared this threshold with the permutation-based p-value threshold corresponding to an overall FDR of 0.05 in each of the 10 datasets. The result shows that $5e-8$ is more stringent than the permutation-based p-value threshold in all datasets, and the xQTLs with $P_{xQTL} < 5 \times 10^{-8}$ represent a large proportion (mean = $\sim 68\%$ across datasets) of the xQTLs with $FDR < 0.05$ (**Supplementary Table 3**). We have revised the manuscript accordingly (lines 169-174, and 199-201).

4. Finally, it would be of high value for the method development community and the final users if authors made the source code of OSCA/THISTLE available in a platform such as

GitHub/GitLab/Bitbucket (if this is done already it is not clear from the manuscript), and provide some information on the running time (even better if this is in comparison with other tools).

Re: We have made the source code of OSCA-THISTLE available at <https://github.com/jianyangqt/osca> (lines 658-659). We have benchmarked the speed of OSCA-THISTLE on a computing platform with 16 GB memory and 16 CPU cores, in comparison with sQTLseeker2-nf. The overall runtime of THISTLE (including the time for estimating isoform-eQTL effects) in the analysis of the ROSMAP data ($n = 832$) with 382 genes and 109,853 SNPs on chromosome 22 was 1.05 minutes (averaged from 10 repeats), approximately 7.6 times faster than sQTLseeker. We have updated the manuscript accordingly (lines 141-145; **Supplementary Fig. 8**).

Minor comments:

-lines 54-68, 74-77. This relates to point 4 from my previous review. As they did in other sections of the paper, authors should make clear in the introduction that ‘event-based’ or ‘transcript-based’ is not a feature of the sQTL mapping statistical method but rather of the splicing phenotype definition/quantification. Again, LeafCutter is not a method for sQTL mapping (lines 60-61), and THISTLE/sQTLseeker are not transcript-based methods (only), as they can work with both transcript-based and event-based quantifications (indeed: <https://doi.org/10.1038/s41467-020-20578-2>, Supp. Note. 2) .

Re: We have made a number of changes throughout the manuscript to make the distinction between alternative splicing variation quantification and sQTL mapping and ensure that we neither refer to LeafCutter as an sQTL mapping method nor refer to THISTLE or sQTLseeker as a transcript-based method only (lines 18-21, 53-57, 60-61, 64-65, 74-79, 213-214, 256, 378-385, 389, 444, and 466).

-Please use the same color codes along the manuscript (both main and supplementary figures) to refer to the same things (e.g. in Fig 4 you have different colors for eQTLs and sQTLs on each panel).

Re: We thank the reviewer for the suggestion and have regenerated all the relevant figures in the revised manuscript (**Figs. 2, 3, and 4; Supplementary Figs. 20, 21, 22, 23, 29, and 30**).

-Fig. 3d. typo: “fol” instead of “fold”.

Re: Fixed.

-Fig. 3e. probably more informative to represent $\log_{10}(\text{distances})$

Re: We thank the reviewer for this suggestion and have changed distance to $\log_{10}(\text{distance})$ in the revised **Fig 3e**.

-lines 50-52: “with a rich collection of sQTLs and a broader spectrum of splicing events”. This cannot be known before performing any analysis.

Re: We agree and have removed this part of statement in the revised manuscript.

-line 232. Is 3,405 the number of genes that are only sGenes? Please clarify in the text.

Re: Yes, we have clarified it in the revised manuscript (line 237).

-line 295. hg2 is not defined before

Re: It is a typo. It should be h_{SNP}^2 . We have fixed it in the revised manuscript (line 299).

-end of line 496. You mean identity matrix?

Re: Yes, it is a typo. We have fixed it in the revised manuscript (line 501).

-Supp. Fig. 3 and other places: “when one simulation parameter was fixed, all the other parameters varied”. I do not understand this. Do you mean “when one simulation parameter varied, all the other parameters were fixed”? If so, fixed to which values?

Re: It means when a parameter is fixed at a certain value, the other parameters are allowed to vary (sampled at random from the specified categories). For example, in **Supplementary Fig. 3a**, when n was fixed at 300, we randomly sampled the sQTL effect size from {small, median, or large}, the number of isoforms per gene from {2, 5, 10, 15, or 20}, and the degree of over-dispersion of transcriptional abundance from {600, 900, 1200, or 1500} (**Supplementary Note**). We have clarified this in the revised supplementary file (**Supplementary Figs. 3 and 4**).

-lines 556-560. Authors replied to my previous review (point 5) with “we indeed used a standard method implemented in QTLtools for eQTL mapping”, but this contrasts to what is stated here. Please clarify.

Re: We confirm that the eQTL analysis was performed by applying the standard linear regression method (implemented in OSCA or QTLtools) to the gene-level expression data rather than a meta-analysis of the isoform-eQTL data (**Supplementary Fig. 11**). We have revised the text to improve the clarity, and the text now reads “To identify eQTLs, we applied a similar quality control and covariate adjustment pipeline as above to gene-level expression data, i.e., excluding genes with TPM < 0.1 or read count < 6 in more than 80% of the samples, TMM normalization, pre-adjusting for covariates identified by VariancePartition and PEER factors, and RINT (**Supplementary Fig. 11**). A linear regression model was applied to the standardized gene-level expression data to test for eQTLs, with the first 5 genetic PCs fitted as covariates, followed by a meta-analysis of the eQTL summary statistics across the 10 RNA-seq datasets by MeCS.” (lines 564-570).

-Supp. Figs. 22, 23,24 “to the proportion of the SNPs” you mean “to the proportion of heritability explained by the control SNPs”?

Re: “Proportion of the SNPs” means the ratio of the number of SNPs in query to the overall number of SNPs. We have clarified this in the figure legends in the revised manuscript (**Supplementary Figs. 24, 25, 26, and 27**).

- Overall, I think figure quality could be improved

Re: We have regenerated most figures (**Figs. 1-6 and Supplementary Fig. 1, 3-7, 13-16, and 18-35**). We appreciated the reviewer for all the thoughtful comments and suggestions, which have helped us improve the quality of our manuscript.

Decision Letter, second revision:

Our ref: NG-A56596R3

29th Apr 2022

Dear Jian,

Thank you for submitting your revised manuscript "Genetic control of RNA splicing and its distinctive role in complex trait variation" (NG-A56596R3). It has now been seen by the original referees and their comments are below. The reviewers find that the paper has improved in revision, and therefore we'll be happy in principle to publish it in Nature Genetics, pending minor revisions to satisfy the referees' final requests and to comply with our editorial and formatting guidelines.

Sincerely,

Michael Fletcher, PhD
Senior Editor, Nature Genetics

ORCID: 0000-0003-1589-7087

Reviewer #3 (Remarks to the Author):

I thank the authors for the detailed response letter and the time devoted to answer my questions. They have resolved all my concerns. Let me add just a couple of final remarks:

- It is intriguing how the drop in power with the significance threshold is more marked for sQTLseeker. However, after all the analyses performed, it seems to be there, although to a lesser extent than what was initially claimed. On the real-data scenario, I believe authors should report the numbers corresponding to the comparison without PEER factors, as this is likely the fairest comparison possible, given that PEERs are computed on the phenotypes that THISTLE uses (RINT absolute

15abundances), but not on those used by sQTLseeker (sqrt of relative abundances). The 1.7-fold actually seems also to match well the simulation results.

- As regards the running time benchmark, please just check that the comparison is THISTLE vs sQTLseeker “without permutations” (i.e. only nominal pass, again this would be the fair comparison as THISTLE does not run permutation-based multiple testing correction).

- In Supp. Fig 17, to help the reader, please clarify that “one phenotype” for sQTLs is the abundance of multiple isoforms (specify how many were simulated) and for eQTLs is the abundance of a single isoform.

Author Rebuttal, second revision:

Reviewer #3 (Remarks to the Author):

I thank the authors for the detailed response letter and the time devoted to answer my questions. They have resolved all my concerns. Let me add just a couple of final remarks:

Re: We thank the reviewer for all the three rounds of comments, which have greatly helped to improve our manuscript.

- It is intriguing how the drop in power with the significance threshold is more marked for sQTLseeker. However, after all the analyses performed, it seems to be there, although to a lesser extent than what was initially claimed. On the real-data scenario, I believe authors should report the numbers corresponding to the comparison without PEER factors, as this is likely the fairest comparison possible, given that PEERs are computed on the phenotypes that THISTLE uses (RINT absolute abundances), but not on those used by sQTLseeker (sqrt of relative abundances). The 1.7-fold actually seems also to match well the simulation results.

Re: As per the reviewer’s suggestion, we have included a statement in the main text (lines 120-122).

The text now reads: “Further analysis in the ROSMAP data without covariate adjustment showed that the number of sGenes decreased for both methods, but the ratio of THISTLE to sQTLseeker remained to be 1.7 at $P < 5 \times 10^{-8}$ ”.

- As regards the running time benchmark, please just check that the comparison is THISTLE vs sQTLseeker “without permutations” (i.e. only nominal pass, again this would be the fair

comparison as THISTLE does not run permutation-based multiple testing correction).

Re: We have confirmed and noted it in **Extended Data Figure 4** that sQTLseekeR was performed without permutations in the runtime comparison.

- In Supp. Fig 17, to help the reader, please clarify that “one phenotype” for sQTLs is the abundance of multiple isoforms (specify how many were simulated) and for eQTLs is the abundance of a single isoform.

Re: We thank the reviewer for this suggestion and have clarified the phenotypes for both the sQTL and eQTL analyses in the figure legend (now **Supplementary Figure 10**).

Final Decision Letter:

In reply please quote: NG-A56596R4 Yang

8th Jul 2022

Dear Jian,

I am delighted to say that your manuscript "Genetic control of RNA splicing and its distinct role in complex trait variation" has been accepted for publication in an upcoming issue of Nature Genetics.

Your paper will be published online after we receive your corrections and will appear in print in the next available issue. You can find out your date of online publication by contacting the Nature Press Office (press@nature.com) after sending your e-proof corrections. Now is the time to inform your Public Relations or Press Office about your paper, as they might be interested in promoting its publication. This will allow them time to prepare an accurate and satisfactory press release. Include your manuscript tracking number (NG-A56596R4) and the name of the journal, which they will need when they contact our Press Office.

Please note that *Nature Genetics* is a Transformative Journal (TJ). Authors may publish their research with us through the traditional subscription access route or make their paper immediately open access through payment of an article-processing charge (APC). Authors will not be required to make a final decision about access to their article until it has been accepted. [Find out more about Transformative Journals](https://www.springernature.com/gp/open-research/transformative-journals)

Authors may need to take specific actions to achieve [compliance](https://www.springernature.com/gp/open-research/funding/policy-compliance-faqs) with funder and institutional open access mandates. If your research

is supported by a funder that requires immediate open access (e.g. according to [Plan S principles](https://www.springernature.com/gp/open-research/plan-s-compliance)) then you should select the gold OA route, and we will direct you to the compliant route where possible. For authors selecting the subscription publication route, the journal's standard licensing terms will need to be accepted, including <https://www.nature.com/nature-portfolio/editorial-policies/self-archiving-and-license-to-publish>. Those licensing terms will supersede any other terms that the author or any third party may assert apply to any version of the manuscript.

Please note that Nature Portfolio offers an immediate open access option only for papers that were first submitted after 1 January, 2021.

If you have not already done so, we invite you to upload the step-by-step protocols used in this manuscript to the Protocols Exchange, part of our on-line web resource, natureprotocols.com. If you complete the upload by the time you receive your manuscript proofs, we can insert links in your article that lead directly to the protocol details. Your protocol will be made freely available upon publication of your paper. By participating in natureprotocols.com, you are enabling researchers to more readily reproduce or adapt the methodology you use. [Natureprotocols.com](https://natureprotocols.com) is fully searchable, providing your protocols and paper with increased utility and visibility. Please submit your protocol to <https://protocolexchange.researchsquare.com/>. After entering your nature.com username and password you will need to enter your manuscript number (NG-A56596R4). Further information can be found at <https://www.nature.com/nature-portfolio/editorial-policies/reporting-standards#protocols>

Sincerely,

Michael Fletcher, PhD
Senior Editor, Nature Genetics

ORCID: 0000-0003-1589-7087